# PROBABILISTIC ROBUSTNESS FOR FREE?
# REVISITING TRAINING VIA A BENCHMARK

## ABSTRACT

Deep learning models are notoriously vulnerable to imperceptible perturbations. Most existing research centers on adversarial robustness (AR), which evaluates models under worst-case scenarios by examining the existence of deterministic adversarial examples (AEs). In contrast, probabilistic robustness (PR) adopts a statistical perspective, measuring the probability that predictions remain correct under stochastic perturbations. While PR is widely regarded as a practical complement to AR, dedicated training methods for improving PR are still relatively underexplored, albeit with emerging progress. Among the few PR-targeted training methods, we identify three limitations: *i)* non-comparable evaluation protocols; *ii)* limited comparisons to strong AT baselines despite anecdotal PR gains from AT, and; *iii)* no unified framework to compare the generalization of these methods. Thus, we introduce `PRBench`, the first benchmark dedicated to evaluating improvements in PR achieved by different robustness training methods. `PRBench` empirically compares most common AT and PR-targeted training methods using a comprehensive set of metrics, including clean accuracy, PR and AR performance, training efficiency, and generalization error (GE). We also provide theoretical analysis on the GE of PR performance across different training methods. Main findings revealed by `PRBench` include: AT methods are more versatile than PR-targeted training methods in terms of improving both AR and PR performance across diverse hyperparameter settings, while PR-targeted training methods consistently yield lower GE and higher clean accuracy. A leaderboard comprising 222 trained models across 7 datasets and 10 model architectures is publicly available at https://tmpspace.github.io/PRBenchLeaderboard/.

## 1 INTRODUCTION

Deep learning (DL) has demonstrated remarkable potential to drive transformative advancements across a wide range of industries, such as autonomous driving and medical diagnostics. In such safety-critical domains, robustness is a fundamental prerequisite for DL models' pervasive deployment. Numerous studies have investigated DL robustness, leading to a range of benchmarks that systematically track the progress in the topic Ling et al. (2019); Guo et al. (2023); Croce et al. (2021); Tang et al. (2021); Dong et al. (2020); Liu et al. (2025). These efforts have predominantly focused on *adversarial robustness* (AR), which emphasizes the extreme *worst-case* scenarios by evaluating local robustness based on the existence of *deterministic* adversarial examples (AEs): subtle perturbations to inputs that alter model predictions Szegedy et al. (2014); Goodfellow (2015); Papernot et al. (2016b).

A more recent and practical perspective, *probabilistic robustness* (PR) Webb et al. (2019); Weng et al. (2019); Couellan (2021); Baluta et al. (2021); Zhao et al. (2021); Tit et al. (2021); Robey et al. (2022); Pautov et al. (2022); Zhang et al. (2023b); Huang et al. (2023); Dong et al. (2023); TIT et al. (2023); Zhang et al. (2024a;b; 2025); Zhang & Sun (2025); Zhang et al. (2023a), employs statistical approaches to answer the question: "What is the probability that predictions remain correct under stochastic perturbations". This probabilistic view is arguably more relevant to real-world applications than AR, as it provides an *overall* assessment of a model's local robustness, accounting for scenarios where AEs may exist Webb et al. (2019); Huang et al. (2023); Zhao (2025) and acknowledging *residual risks* Zhang et al. (2023b; 2024a; 2025) that are more realistic to manage in practice.

Despite its potential, most existing work on PR is restricted to *evaluation*, with only a few studies Wang et al. (2021); Robey et al. (2022); Zhang et al. (2024a; 2025) exploring *training* methods specifically designed to *improve* PR Zhao (2025). Among them, we identify three limitations. First, they use different sets of evaluation metrics, with none adopting a comprehensive set to assess the training methods holistically, making it difficult to understand which methods are truly effective/efficient and under what conditions. Second, an interesting observation is that adversarial training (AT) that primarily designed to improve AR often enhances PR as a "free by-product". However, existing studies assess only a limited selection of AT methods, limiting the understanding of AT's potential as a more efficient PR improvement tool. Third, there is no *unified theoretical framework* to compare the generalisability of PR-targeted training, making it unclear of their broader applicability in various context. These gaps and the rapid growth of PR research highlight the need for a systematic benchmark for PR training methods, which is currently missing from the literature.

To bridge the gaps, we introduce `PRBench`, the first benchmark dedicated to evaluating training methods for improving PR. Key features of `PRBench` include: *i)* A comprehensive set of metrics, covering clean accuracy, PR and AR performance, training efficiency, and generalisation error (GE); *ii)* A large set of training methods including most common AT methods and all PR-targeted training methods; *iii)* Theoretical bounds of GE are derived under a unified framework of Uniform Stability Analysis Xiao et al. (2022b); Cheng et al. to conclude comparative insights. Specifically, `PRBench` includes 222 trained models based on 7 widely adopted datasets and 10 model architectures. It uses 4 common adversarial attacks to measure AR performance, 2 types of PR metrics (with various hyperparameters, e.g., distributions of perturbations) for PR, GE metrics for both AR and PR, the clean accuracy and the training time. While the total number of training methods included is 13, `PRBench` code-base is designed to be extendable for future inclusion and comparison of new methods.

Our analysis shows that, in most cases, AT surprisingly outperforms PR-targeted training in improving both AR and PR. This suggests that PR *does* generally come "for free"[1] when applying AT for AR; *not vice versa*. That said, PR-targeted methods offer advantages in, e.g., better generalisability and clean accuracy. The very recent "hybrid" training method combines the advantages of both, but at the price of training efficiency. Key contributions of this paper include:

- **Benchmarking:** After formalizing a general formulation of the PR-targeted training methods, we develop the first benchmark dedicated for PR, evaluating a broad set of training methods with metrics of AR, PR, accuracy, generalisability, and efficiency.

- **Analysis:** Analysis is provided based on both empirical and theoretical studies, highlighting the strengths, limitations, and trade-offs among training methods in the context of improving PR.

- **Open-source repository and Leaderboard:** All experimental details are included in the supplementary materials. A public repository will be released after the review process. A public leaderboard is also released at `https://tmpspace.github.io/PRBenchLeaderboard/`.

## 2 PRELIMINARIES AND RELATED WORKS

### 2.1 GENERALISATION ERRORS AND ADVERSARIAL ROBUSTNESS

Consider a classification task where $\boldsymbol{x} \in \mathcal{X} \subseteq \mathbb{R}^d$ denotes the inputs, and $y \in \mathcal{Y} \subseteq \{1, 2, \ldots, \kappa\}$ represents the labels. Let $\mathcal{D}$ be an unknown probability measure over $\mathcal{X} \times \mathcal{Y}$. We define $f : \Theta \times \mathcal{X} \to \mathbb{R}^\kappa$ as a DL model, parameterised by $\boldsymbol{\theta} \in \Theta$, and $\mathbf{p}$ denote the *softmax* function. Given i.i.d. samples $S = \{(\boldsymbol{x}_i, y_i)\}_{i=1}^n$ drawn from $\mathcal{D}$ and a loss function[2] $\mathcal{L} : [0, 1]^\kappa \times \mathcal{Y} \to \mathbb{R}^+$, the *natural* and *empirical risks* can be represented as:

$$R(\boldsymbol{\theta}) = \mathbb{E}_{(\boldsymbol{x}, y) \sim \mathcal{D}} \left[ \mathcal{L}(\mathbf{p}(f(\boldsymbol{x}, \boldsymbol{\theta})), y) \right] \quad \text{and} \quad R_S(\boldsymbol{\theta}) = \frac{1}{n} \sum_{i=1}^n \mathcal{L}(\mathbf{p}(f(\boldsymbol{x}_i, \boldsymbol{\theta})), y_i). \tag{1}$$

---

[1]"For free" is used figuratively (not literally) to indicate that PR gains emerge naturally under AT. Trade-offs such as reduced clean accuracy still apply; see later sections for details.

[2]For simplicity, we also denote the composed loss function in Eq. 1 and 3 as $\mathcal{L}(\boldsymbol{x} + \boldsymbol{\delta}, y; \boldsymbol{\theta})$ in the following sections, whenever it is unambiguous.

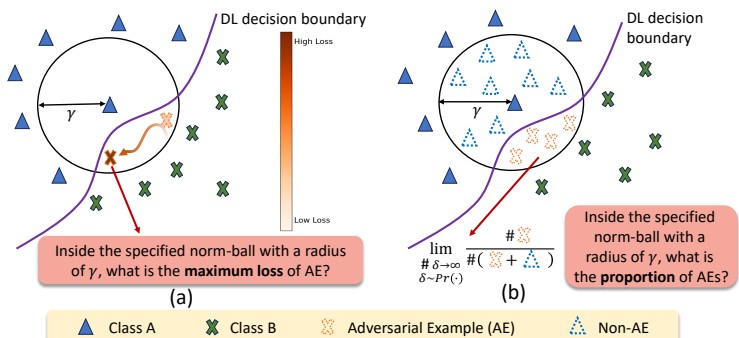

Figure 1: Comparison of Adversarial (a) and Probabilistic Robustness (b)

**Definition 1 (Generalization Error)** *The GE of $f$ on $S$ is then defined as the difference between the natural and empirical risk:*

$$\mathrm{GE}(\theta) = R(\theta) - R_S(\theta). \tag{2}$$

Although definitions of robustness vary across different DL tasks and model types, it generally refers to a DL model's ability to maintain consistent predictions despite small input perturbations. Typically it is defined as all inputs in a region $\eta$ have the same prediction, where $\eta$ is a small norm ball (in a $L_p$-norm distance) of radius $\gamma$ around an input $\boldsymbol{x}$. A perturbed input (e.g., by adding noise on $\boldsymbol{x}$) $\boldsymbol{x}'$ within $\eta$ is an AE if its prediction label differs from ground truth label $y$.

To evaluate AR, we normally formulate it as a question of maximizing the prediction loss, cf. Fig. 1 (a), where *adversarial attacks* are introduced to find such a worst-case AE through perturbation:

$$\boldsymbol{\delta}^\star = \arg \max_{\|\boldsymbol{\delta}\| \leq \gamma} \mathcal{L}(\boldsymbol{x} + \boldsymbol{\delta}, y; \boldsymbol{\theta}). \tag{3}$$

AT Goodfellow (2015); Tramèr et al. (2018); Gowal et al. (2019); Madry et al. (2018); Balunović & Vechev (2020) is the most common and effective empirical approach for enhancing AR. It is typically formulated as a min-max optimisation problem, where the inner max aims at finding the worst-case AE in Eq. 3, e.g., by FGSM Goodfellow (2015) and PGD Madry et al. (2018). Then the model is trained to minimise the loss over these AEs derived from a training dataset $\mathcal{D}$:

$$\min_{\boldsymbol{\theta}} \mathbb{E}_{(\boldsymbol{x},y)\sim\mathcal{D}} \left[ \max_{\|\boldsymbol{\delta}\| \leq \gamma} \mathcal{L}(\boldsymbol{x} + \boldsymbol{\delta}, y; \boldsymbol{\theta}) \right]. \tag{4}$$

### 2.2 PROBABILISTIC ROBUSTNESS

PR adopts a *probabilistic* view, evaluating the *overall* local robustness in the presence of AEs, cf. Fig. 1 (b). This probabilistic notion of robustness is acknowledged to be more practical than the worst-case AR, as many applications only require the risk of AEs to stay below an acceptable threshold, rather than eliminating them entirely Webb et al. (2019); Zhang et al. (2023b; 2024b); Zhao (2025).

**Definition 2 (Probabilistic Robustness)** *For a DL classifier $f_{\boldsymbol{\theta}}$ that takes input $\boldsymbol{x}$ and returns a prediction label, the PR of an input $\boldsymbol{x}$ in a norm ball of radius $\gamma$ is:*

$$PR(\boldsymbol{x}, \gamma) = \mathbb{E}_{\substack{\boldsymbol{\delta}\sim Pr(\cdot|\boldsymbol{x}) \\ \|\boldsymbol{\delta}\| \leq \gamma}} [I_{\{f_{\boldsymbol{\theta}}(\boldsymbol{x}+\boldsymbol{\delta})=y\}}(\boldsymbol{x} + \boldsymbol{\delta})], \tag{5}$$

*where $I_{\mathcal{S}}(\boldsymbol{x})$ is an indicator function that equals 1 when $\mathcal{S}$ is true and 0 otherwise; $Pr(\cdot)$ is the local distribution of inputs representing how perturbations $\boldsymbol{\delta}$ are generated, which is precisely the "input model" used by Webb et al. (2019); Weng et al. (2019); Zhang et al. (2024b).*

Def. 2 suggests that PR is the probability that the model prediction remains unchanged from a random perturbation $\boldsymbol{x}'$. A "frequentist" interpretation of this expected probability is—it is the *limiting relative frequency* of perturbations where the output label is preserved, in an infinite sequence of independently generated perturbations Zhang et al. (2024b). In other words, the "proportion" of

non-AEs in the infinite set of perturbed inputs. To evaluate PR, one of the earliest works Webb et al. (2019) introduced a useful black-box statistical estimator, especially for cases where PR is very high. Later, more efficient white-box statistical estimators were proposed in TIT et al. (2023). Additionally, PR has been extended to applications such as explainable AI Huang et al. (2023) and text-to-image models Zhang et al. (2024b). For an overview of PR estimators, we refer readers to Zhao (2025).

In contrast to the relatively extensive studies on PR estimators, research on training methods for PR remains scarce. To the best of our knowledge, four studies Wang et al. (2021); Robey et al. (2022); Zhang et al. (2024a; 2025) explicitly motivated to develop training methods for improving PR. The first three can be categorized as Risk-based Training (RT) methods, which shift away from the worst-case perspective towards a training paradigm based on statistical risks induced by distributional perturbations, incorporating *perturbation risk functions* into the objective. We formally introduce its general formulation as:

**Definition 3 (Risk-based Training (RT))** *Reusing the notations above, RT is to*

$$\min_{\boldsymbol{\theta}} \mathbb{E}_{(\boldsymbol{x},y)\sim\mathcal{D}} \left[ \mathcal{R}_\gamma \left( \mathcal{L}(\boldsymbol{x}+\boldsymbol{\delta}, y; \boldsymbol{\theta}), \boldsymbol{\delta} \sim Pr(\cdot \mid \boldsymbol{x}) \right) \right], \tag{6}$$

*where $\mathcal{R}_\gamma$ is the perturbation risk function that defines some statistical quantities of the loss over a distribution of perturbations $Pr(\cdot \mid \boldsymbol{x})$.*

The specific choice of $\mathcal{R}_\gamma$ in RT methods varies: Wang et al. (2021) uses the identity function, while Robey et al. (2022); Zhang et al. (2024a) adopt functions calculating the Conditional Value-at-Risk (CVaR) and Entropic Value-at-Risk (EVaR), respectively. Essentially, rather than training on optimized AEs like AT methods for AR, RT methods train on statistical risks quantified by sampling stochastic perturbations from the perturbation distribution.

A very recent idea of adapting AT for PR was proposed in Zhang et al. (2025), which does not follow RT paradigm of Def. 3. Instead, their approach aligns with the traditional AT formulation in Eq. 4. Through multi-start PGD attacks and decision boundary exploration, AT-PR identifies an optimal AE whose neighborhood constitutes the largest all-AE region. Intuitively, training over such an optimal AE would reduce the overall "proportion" of AEs in the local norm ball and thus improve PR. We classify it as "hybrid" given it is targeting PR but adapting AT by solving a new min-max problem. Both RT methods and the "hybrid" method AT-PR constitute the current class of PR-targeted method. Further details for all four training methods are provided in Appendix E.

Table 1: Summary of abbreviations used in the paper.

| Acronyms | Meaning | Acronyms | Meaning |
|---|---|---|---|
| AR | Adversarial Robustness | AT | Adversarial Training |
| ERM | Empirical Risk Minimization | GE | Generalization Error |
| PR | Probabilistic Robustness | RT | Risk-based Training |

## 3 DESCRIPTION OF PRBENCH

All existing robustness benchmarks focus exclusively on AR Guo et al. (2023); Croce et al. (2021); Tang et al. (2021); Dong et al. (2020); Liu et al. (2025), with more related benchmarks summarized in Appendix B.5. In contrast, our PRBench is the first benchmark dedicated to PR.

The models under evaluation spans 3 types of architectures: (1) plain CNNs; (2) residual CNNs; and (3) transformer-based models. We train a diverse set of models, including VGG-19, SimpleCNN, ResNet-18, ResNet-34, WRN-28-10, ViT (Small/Base/Large), and DeiT (Tiny/Small). In total, 222 models are trained on 7 datasets: MNIST, SVHN, CIFAR-10, CIFAR-100, CINIC-10, TinyImageNet, and ImageNet-50. Model selection and dataset specifications are provided in Appendix B.4.

### 3.1 TRAINING METHODS

As shown in Table 2, PRBench categorizes all training methods into 4 groups: standard training (i.e., empirical risk minimization (ERM)), 6 AT methods, 4 RT methods (with corruption training using uniform, Gaussian, and Laplace noise) and the "hybrid" method AT-PR.

For AT, we consider the following representative methods. The PGD attack is widely regarded as the standard approach, which minimizes the adversarial cross-entropy loss $\mathcal{L}_{CE}$. Building on this, logit pairing methods Kannan et al. (2018), including adversarial logit pairing (ALP) and clean logit pairing (CLP), augment the objective with a regularization term coupling natural and AEs. TRADES further introduces a KL-based regularization, defining its objective as a combination of the natural loss and the KL divergence $\mathcal{L}_{KL}$ between predictions on clean and adversarial inputs. MART follows the same KL-based regularization but augments it with the adversarial cross-entropy loss $\mathcal{L}_{CE}$, while additionally emphasizing misclassified examples through larger penalty weights. Overall, existing AT methods can be characterized by different uses of $\mathcal{L}_{CE}$ and $\mathcal{L}_{KL}$, either individually or in combination, for both AE generation and training. We identify a missing configuration and introduce KL-PGD, a combination that generates AEs using $\mathcal{L}_{KL}$ (as in TRADES) and trains the model with $\mathcal{L}_{CE}$ (as in PGD). KL-PGD is designed as a diagnostic variant to disentangle the effects of loss functions from perturbation generation strategies, thereby providing a controlled comparison point for systematically evaluating the relative contributions of different optimization objectives.

Table 2: Loss functions and AE generation strategies for different training methods.

| Type | Method | Loss Function | AE Generation |
|---|---|---|---|
| Standard | ERM | $\mathcal{L}_{CE}(\mathbf{p}(\boldsymbol{x},\boldsymbol{\theta}),y)$ | – |
| AT | PGD | $\mathcal{L}_{CE}(\mathbf{p}(\boldsymbol{x}+\boldsymbol{\delta},\boldsymbol{\theta}),y)$ | |
| | MART | $\mathcal{L}_{CE}(\mathbf{p}(\boldsymbol{x}+\boldsymbol{\delta},\boldsymbol{\theta}),y)+\lambda(1-\mathbf{p}(y|\boldsymbol{x}))\cdot\mathcal{L}_{KL}(\mathbf{p}(\boldsymbol{x},\boldsymbol{\theta})\|\mathbf{p}(\boldsymbol{x}+\boldsymbol{\delta},\boldsymbol{\theta}))$ | $\boldsymbol{\delta}_t=\alpha\cdot\mathrm{sign}\left(\nabla_{\boldsymbol{x}}\mathcal{L}_{CE}(\mathbf{p}(\boldsymbol{x}_{t-1}+\boldsymbol{\delta}_{t-1},\boldsymbol{\theta}),y)\right)$ |
| | ALP | $\mathcal{L}_{CE}(\mathbf{p}(\boldsymbol{x}+\boldsymbol{\delta},\boldsymbol{\theta}),y)+\lambda\cdot\|\mathbf{p}(\boldsymbol{x}+\boldsymbol{\delta},\boldsymbol{\theta})-\mathbf{p}(\boldsymbol{x},\boldsymbol{\theta})\|_2^2$ | |
| | CLP | $\mathcal{L}_{CE}(\mathbf{p}(\boldsymbol{x},\boldsymbol{\theta}),y)+\lambda\cdot\|\mathbf{p}(\boldsymbol{x}+\boldsymbol{\delta},\boldsymbol{\theta})-\mathbf{p}(\boldsymbol{x},\boldsymbol{\theta})\|_2^2$ | |
| | TRADES | $\mathcal{L}_{CE}(\mathbf{p}(\boldsymbol{x},\boldsymbol{\theta}),y)+\lambda\cdot\mathcal{L}_{KL}(\mathbf{p}(\boldsymbol{x},\boldsymbol{\theta})\|\mathbf{p}(\boldsymbol{x}+\boldsymbol{\delta},\boldsymbol{\theta}))$ | $\boldsymbol{\delta}_t=\alpha\cdot\mathrm{sign}\left(\nabla_{\boldsymbol{x}}\mathcal{L}_{KL}(\mathbf{p}(\boldsymbol{x}_{t-1},\boldsymbol{\theta})\,\|\,\mathbf{p}(\boldsymbol{x}_{t-1}+\boldsymbol{\delta}_{t-1},\boldsymbol{\theta}))\right)$ |
| | KL-PGD | $\mathcal{L}_{CE}(\mathbf{p}(\boldsymbol{x}+\boldsymbol{\delta},\boldsymbol{\theta}),y)$ | |
| RT | Corruption | $\mathcal{L}_{CE}(\mathbf{p}(\boldsymbol{x}+\boldsymbol{\delta},\boldsymbol{\theta}),y)$ | $\boldsymbol{\delta}\sim Pr(\cdot\mid\boldsymbol{x})$ |
| | CVaR | $\mathrm{CVaR}_{1-\rho}\left(\mathcal{L}_{CE}(\mathbf{p}(\boldsymbol{x}+\boldsymbol{\delta},\boldsymbol{\theta}),y)\right)$ | |
| Hybrid | AT-PR | $\mathcal{L}_{CE}(\mathbf{p}(\boldsymbol{x},\boldsymbol{\theta}),y)$ | $\boldsymbol{\delta}=\arg\max_{\boldsymbol{\delta}\in\{\boldsymbol{\delta}^{PGD}\}}k\ \text{s.t. } PR(\boldsymbol{x}+\boldsymbol{\delta},k)=0$ |

By instantiating the general formulation of RT training (Def. 3), three existing works have been proposed, corresponding to essentially two distinct approaches depending on how the perturbation risk function $\mathcal{R}_\gamma$ is realized. A simple augmentation-based strategy is introduced by Wang et al. (2021), referred to as *corruption training*, where each input $\boldsymbol{x}_i$ is perturbed by a sample drawn from a distribution (uniform by default) within an $\ell_\infty$ ball of radius $\gamma$. Robey et al. (2022) proposed a CVaR-based risk objective that emphasizes the tail of the loss distribution, thereby prioritizing samples with larger loss values. A similar work by Zhang et al. (2024a) replaces CVaR with EVaR, which leverages the entire distribution while following the same training strategy via sampling[3]. The "hybrid" method AT-PR targets PR while following the spirit of AT, as summarized in Table 2. It introduces a new min–max optimization objective (Eq. 21). Through multi-start PGD attacks and decision boundary exploration, AT-PR selects a local optimal AE in the norm-ball ($\boldsymbol{\delta}\in\{\boldsymbol{\delta}^{PGD}\}$) whose neighborhood constitutes the largest all-AE region $k$, i.e., $PR(x+\delta,k)=0$.

## 3.2 EVALUATION METRICS

Table 3: Evaluation metrics of AR, PR, and GE.

| Type | AR | PR | GE |
|---|---|---|---|
| **Metric** | $AR=\dfrac{1}{|\mathcal{D}|}\sum_{\boldsymbol{x}_i\in\mathcal{D}}I_{\{f_{\boldsymbol{\theta}}(\boldsymbol{x}_i')=y_i\}}(\boldsymbol{x}_i)$ | $PR_{\mathcal{D}}(\gamma)=\dfrac{1}{|\mathcal{D}|}\sum_{\boldsymbol{x}_i\in\mathcal{D}}PR(\boldsymbol{x}_i,\gamma)$ $ProbAcc(\rho)=\dfrac{1}{|\mathcal{D}|}\sum_{\boldsymbol{x}_i\in\mathcal{D}}I_{\{PR(\boldsymbol{x}_i,\gamma)\geq1-\rho\}}(\boldsymbol{x}_i)$ | $GE_{PR_{\mathcal{D}}(\gamma)}=PR_{\mathcal{D}_{\text{train}}}(\gamma)-PR_{\mathcal{D}_{\text{test}}}(\gamma)$ $GE_{AR}=AR_{\mathcal{D}_{\text{train}}}-AR_{\mathcal{D}_{\text{test}}}$ |

In addition to clean accuracy, PRBench evaluates the models across three core aspects: AR, PR, and GE, cf. Table 3. AR performance is assessed by classification accuracy under adversarial attacks (Eq. 3). PR is quantified using two metrics from the literature: $PR_{\mathcal{D}}(\gamma)$ Webb et al. (2019), the average probability that correctly classified inputs $\boldsymbol{x}_i$ remain correct under perturbations of radius $\gamma$ (cf. Def. 2), and $ProbAcc(\rho)$ Robey et al. (2022), the fraction of inputs $\boldsymbol{x}$ whose $PR(\boldsymbol{x},\gamma)$ exceeds a given threshold $1-\rho$. For the two PR metrics, $|\mathcal{D}|$ denotes the number of test samples that are correctly classified by the model, whereas for AR, $|\mathcal{D}|$ refers to the total number of test samples. GE captures robustness gap between the training and test sets. More details are deferred to Appendix F.

---

[3]Given the conceptual similarity, categorize both EVaR and CVaR under the CVaR category in Table 2.

## 4 THEORETICAL FOUNDATIONS OF ROBUST OVERFITTING

While AT is regarded as one of the most promising methods for enhancing AR, empirical studies Rice et al. (2020); Gowal et al. (2020) have shown that it suffers significantly from overfitting. To understand this phenomenon, several theoretical studies have been conducted under different frameworks, including VC-dimension Montasser et al. (2019); Attias et al. (2022), Rademacher complexity Khim & Loh (2018); Yin et al. (2019); Awasthi et al. (2020); Xiao et al. (2022a), and Uniform Algorithmic Stability Farnia & Ozdaglar (2021); Xing et al. (2021); Xiao et al. (2022c). Moreover, Jiang et al. (2020) study generalization in standard DL models, focusing on the empirical correlation between complexity measures such as VC-dimension and norm-based measures. Kim et al. (2023) further conduct a large-scale empirical study of robust generalization under AT, analyzing how margin-based, smoothness-based, flatness-based, and gradient-based measures correlate with the generalization. A more detailed discussion of related work on generalization can be found in Appendix C. Among all these theoretical perspectives, one of the most notable contributions is by Xiao et al. (2022b), which introduces the concept of $\eta$-approximate $\beta$-smoothness, as defined in Eq. 7. This work Xiao et al. (2022b) analyses the surrogate loss of the max function and show that generalization is affected by an additional term proportional to $\eta$.

**Definition 4 (Approximate Smoothness Xiao et al. (2022b))** *Let $f : \mathbb{R}^d \times \mathbb{R}^m \to \mathbb{R}$, then $f$ is $\eta$-approximate $\beta$-smooth, if for all $\boldsymbol{z} \in \mathbb{R}^d$, and $\forall \boldsymbol{\theta}_1, \boldsymbol{\theta}_2$, we have:*

$$\|\nabla_{\boldsymbol{\theta}} f(\boldsymbol{z}, \boldsymbol{\theta}_2) - \nabla_{\boldsymbol{\theta}} f(\boldsymbol{z}, \boldsymbol{\theta}_1)\| \le \beta \|\boldsymbol{\theta}_2 - \boldsymbol{\theta}_1\| + \eta. \tag{7}$$

To facilitate comparison between different AT schemes, we make the following assumptions.

**Assumption 1** *Assume that the machine learning model $f(\boldsymbol{x}, \boldsymbol{\theta})$ is $L_{\boldsymbol{\theta}}$-Lipschitz w.r.t. $\boldsymbol{\theta}$ and $L$-Lipschitz w.r.t. $\boldsymbol{x}$ such that:*

$$L_{\boldsymbol{\theta}} \triangleq \sup_{\boldsymbol{\theta}_2 \neq \boldsymbol{\theta}_1} \frac{\|f(\boldsymbol{x}, \boldsymbol{\theta}_2) - f(\boldsymbol{x}, \boldsymbol{\theta}_1)\|}{\|\boldsymbol{\theta}_2 - \boldsymbol{\theta}_1\|} \quad and \quad L \triangleq \sup_{\boldsymbol{x}_2 \neq \boldsymbol{x}_1} \frac{\|f(\boldsymbol{x}_2, \boldsymbol{\theta}) - f(\boldsymbol{x}_1, \boldsymbol{\theta})\|}{\|\boldsymbol{x}_2 - \boldsymbol{x}_1\|}. \tag{8}$$

*Similarly, we assume the smoothness condition for the gradient of model $f$ w.r.t. $\boldsymbol{\theta}$ as:*

$$\forall \boldsymbol{x} \in \mathcal{X}, \quad \|\nabla_{\boldsymbol{\theta}} f(\boldsymbol{x}, \boldsymbol{\theta}_2) - \nabla_{\boldsymbol{\theta}} f(\boldsymbol{x}, \boldsymbol{\theta}_1)\| \le \beta_{\boldsymbol{\theta}} \|\boldsymbol{\theta}_2 - \boldsymbol{\theta}_1\|, \tag{9}$$

$$\forall \boldsymbol{\theta} \in \Theta, \quad \|\nabla_{\boldsymbol{\theta}} f(\boldsymbol{x}_2, \boldsymbol{\theta}) - \nabla_{\boldsymbol{\theta}} f(\boldsymbol{x}_1, \boldsymbol{\theta})\| \le \beta \|\boldsymbol{x}_2 - \boldsymbol{x}_1\|. \tag{10}$$

This assumption has been widely adopted and validated in prior work Farnia & Ozdaglar (2021); Xing et al. (2021); Xiao et al. (2022c;b), where it serves as a foundational premise for analyzing algorithmic stability and generalisation under AT. We show that the GE is bounded in Thm. 1.

**Theorem 1** *Given the Lipschitz and smoothness assumption in Assumption 1 for classifier $f$, we show that the surrogate loss $\max_{\|\boldsymbol{\delta}\| \le \gamma} \mathcal{L}_{CE}(\mathbf{p}(f(\boldsymbol{x} + \boldsymbol{\delta}, \boldsymbol{\theta})), y)$ is $\varphi$-Lipschitz and $\phi$-approximate $\psi$-smooth, such that*

$$\varphi = 2L_{\boldsymbol{\theta}} \quad and \quad \phi = (4\beta\gamma + 2L_{\boldsymbol{\theta}}) \quad and \quad \psi = \left(2\beta_{\boldsymbol{\theta}} + L_{\boldsymbol{\theta}}^2\right). \tag{11}$$

*We run SGD with learning rate $\alpha_t \le c/t$ for $T$ steps with a constant $c$ such that $1/c \ge \psi$. Then, the GE is bounded as*

$$|R(\boldsymbol{\theta}) - R_S(\boldsymbol{\theta})| \le \frac{1}{n} + \frac{2\varphi^2 + n\varphi\phi}{\psi(n-1)} T. \tag{12}$$

Thm. 1 follows from the results in Xiao et al. (2022b), where the authors conclude that the additional term $n\varphi\phi$ contributes to robust overfitting. The complete proof corresponds to the special case $\lambda = 0$ of Thm. 3 in Appendix G.2. And the proof for uniform stability is shown in Appendix G.5.

**Theorem 2** *Follow the same condition for Thm. 1, consider the objective function for contained AT*

$$\max_{\|\boldsymbol{\delta}\|_2 \le \gamma} \mathcal{L}_{CE}(f(\mathbf{p}(\boldsymbol{x} + \boldsymbol{\delta}, \boldsymbol{\theta})), y) + \lambda \|\mathbf{p}(f(\boldsymbol{x} + \boldsymbol{\delta}, \boldsymbol{\theta})) - \mathbf{p}(f(\boldsymbol{x}, \boldsymbol{\theta}))\|_2^2. \tag{13}$$

*We show that the Lipschitz constant, $\widetilde{\varphi}$, for the objective function is*

$$\widetilde{\varphi} = 2L_{\boldsymbol{\theta}} + 2\lambda(\nu\beta\gamma + 3\nu^2 L_{\boldsymbol{\theta}}) = \varphi + 2\lambda(\nu\beta\gamma + 3\nu^2 L_{\boldsymbol{\theta}}), \tag{14}$$

*and it is $\widetilde{\phi}$-approxmiate $\widetilde{\psi}$-smooth, such that*

$$\widetilde{\phi} = (4\beta\gamma + 2\nu L_{\boldsymbol{\theta}}) + 2\lambda\gamma\left(\nu^2\beta + 2\nu LL_{\boldsymbol{\theta}}\right) = \phi + 2\lambda\gamma\left(\nu^2\beta + 2\nu LL_{\boldsymbol{\theta}}\right) - 2L_{\boldsymbol{\theta}}(1-\nu) \tag{15}$$

$$\widetilde{\psi} = \left(2\beta_{\boldsymbol{\theta}} + L_{\boldsymbol{\theta}}^2\right) + 6\lambda\left(\nu^2\beta_{\boldsymbol{\theta}} + 4\nu L_{\boldsymbol{\theta}}^2\right) = \psi + 6\lambda\left(\nu^2\beta_{\boldsymbol{\theta}} + 4\nu L_{\boldsymbol{\theta}}^2\right) \tag{16}$$

*where $\varphi$, $\phi$ and $\psi$ are corresponding variables for AT in Thm. 1 without constraints and*

$$\nu \triangleq \max_{\|\boldsymbol{\delta}\|_2 \leq \gamma, \boldsymbol{\theta} \in \Theta} \|\mathbf{p}(f(\boldsymbol{x} + \boldsymbol{\delta}, \boldsymbol{\theta})) - \mathbf{p}(f(\boldsymbol{x}, \boldsymbol{\theta}))\|_2, \tag{17}$$

*denotes the upper bound of the penalty function.*

Thm 1 aims to establish the fundamental theoretical basis for our GE analysis: it formalizes the Lipschitzness and smoothness assumption of the adversarial surrogate loss follows from the results in Xiao et al. (2022b), which is the key prerequisite for analyzing robust overfitting. Thm. 2 builds directly on Thm 1 by extending these properties to the AT objective with regularization, quantifying how the added regularizer changes the Lipschitz and smoothness constants and thereby affects the GE behavior observed in our subsequent empirical results.

## 5 ANALYSIS AND DISCUSSION

### 5.1 EMPIRICAL ANALYSIS OF EVALUATION RESULTS

While the complete evaluation results across all models and datasets are provided in Appendix D, we present *representative* example in Tables 4 and 5 and Fig. 2 for brevity.

**AT methods improve PR "for free".** All AT methods consistently improve PR alongside AR. Within the training perturbation norm-ball, models trained by AT achieve over 99% of $PR_{\mathcal{D}}(\gamma)$, cf. Table 4. The $ProbAcc(\rho = 0.01)$ metric further indicates that more than 97% of test samples exhibit $PR_{\mathcal{D}}(\gamma) > 99\%$. Moreover, as shown in Fig. 2 (c) (e), model trained with AT methods remains stable PR performance as the perturbation radius $\gamma$ increases and across different robustness tolerance levels $\rho$, consistently achieving over 93% robustness under both $PR_{\mathcal{D}}(\gamma)$ and $ProbAcc(\rho)$. Our results show that AT, although originally designed to improve AR, consistently yields improved PR as a by-product, without any modification to its training objective. Here, we use the phrase "for free" to indicate that PR improvements arise naturally under standard AT, without requiring any modification to its original training paradigm. It does not imply there is no cost in other metrics, indeed, AT still suffers from a drop in clean accuracy. For example, the clean accuracy decreases from 94.85% under standard training to 83.83% after PGD training (cf. Table 4). In this sense, the phenomenon can also be interpreted as a multi-objective trade-off.

**RT methods underperform AT methods on PR performance.** While RT methods effectively improve PR compared with ERM, they still lag behind AT methods, as shown in Table 4. Moreover, their PR performance drops sharply as the perturbation radius increases, cf. Fig. 2 (c) (d), showing limited generalization beyond the norm-ball setting in training. Notably, our analysis of the CVaR codebase from Robey et al. (2022) suggests a potential evaluation inconsistency: it includes misclassified original clean samples when computing $ProbAcc(\rho)$ rather than focusing on the robustness of correctly classified original samples. We have corrected this in our experiments. This view is also supported by prior studies, e.g., Li et al. (2024), which states: "When considering the adversarial robustness of a wrongly classified sample $\boldsymbol{x}$, the robustness should be 0". Similarly, Chen & Lee (2024) emphasizes that "AEs must satisfy the constraints: the original examples are classified correctly while the predictions of the AEs are wrong". In other words, if a sample is misclassified by the model initially, its ability to resist perturbations and maintain the wrong prediction (i.e., being "robustly wrong") does not constitute meaningful robustness.

**AT methods ensure stable and high PR under diverse perturbation distributions.** We evaluate the PR performance of PGD-trained models under three common noise distributions: Uniform, Gaussian, and Laplace, and compare them with models trained by Corruption method using the same noise

Table 4: Performance of different training methods on CIFAR-10 (ResNet-18), evaluated by clean accuracy (Acc.), AR, PR, GE performance, and training time (sec./epoch).

| Type | Method | Acc. % | AR % | | | | $PR_{\mathcal{D}}^{Uniform}(\gamma)$ % | | | | ProbAcc($\rho, \gamma=0.03$) % | | | $GE_{AR}$ % | $GE_{PR_{\mathcal{D}}^{Uni}(\gamma)}$ % | | | | Time |
|---|---|---|---|---|---|---|---|---|---|---|---|---|---|---|---|---|---|---|---|
| | | | $PGD^{10}$ | $PGD^{20}$ | $CW^{20}$ | AA | 0.03 | 0.08 | 0.1 | 0.12 | 0.1 | 0.05 | 0.01 | $PGD^{20}$ | 0.03 | 0.08 | 0.1 | 0.12 | s/ep. |
| Std. | ERM | **94.85** | 0.01 | 0.0 | 0.0 | 0.0 | 97.64 | 76.19 | 61.65 | 47.07 | 95.04 | 93.48 | 89.82 | **0.0** | 6.24 | 3.98 | **2.94** | **2.54** | 3 |
| RT | Corr_Uniform | 94.17 | 0.28 | 0.05 | 0.02 | 0.0 | 99.12 | 90.92 | 82.32 | 70.48 | 97.78 | 97.02 | 94.9 | 0.04 | 4.83 | 6.24 | 4.79 | 2.31 | 3 |
| | CVaR | 89.91 | 41.79 | 33.45 | 0.0 | 0.0 | 98.67 | 87.75 | 78.62 | 68.27 | 96.63 | 95.49 | 92.56 | 1.13 | **3.9** | 4.56 | 3.9 | 2.72 | 61 |
| Hybrid | AT-PR | 86.35 | 48.22 | 46.53 | 47.39 | 44.01 | **99.68** | **98.13** | **97.01** | **95.46** | **99.13** | **98.82** | **98.24** | 27.43 | 11.69 | 11.81 | 11.86 | 12.57 | 160 |
| AT | PGD | 83.83 | 50.86 | 49.46 | 49.18 | 46.34 | 99.63 | 97.89 | 96.59 | 94.85 | 99.05 | 98.73 | 98.01 | 25.02 | 11.4 | 11.01 | 11.2 | 12.03 | 30 |
| | TRADES | 83.34 | 54.09 | 53.15 | 50.84 | 48.99 | 99.55 | 97.74 | 96.33 | 94.55 | 98.88 | 98.68 | 98.22 | 16.12 | 10.29 | 10.22 | 10.08 | 10.27 | 25 |
| | MART | 82.26 | **54.83** | 53.84 | 49.63 | 46.94 | 99.56 | 97.4 | 95.92 | 93.93 | 98.88 | 98.58 | 98.04 | 17.4 | 6.84 | 7.23 | 7.73 | 8.07 | 22 |
| | ALP | 73.98 | 54.41 | **54.08** | 50.72 | 48.41 | 99.27 | 96.73 | 95.16 | 93.24 | 98.55 | 98.29 | 97.85 | 8.44 | 5.26 | 5.93 | 5.52 | 4.44 | 36 |
| | CLP | 81.47 | 54.12 | 53.34 | **51.05** | **49.05** | 99.53 | 97.61 | 96.29 | 94.54 | 98.73 | 98.52 | 97.88 | 15.32 | 8.05 | 7.94 | 8.29 | 8.7 | 36 |
| | KL-PGD | 87.55 | 49.4 | 48.43 | 47.06 | 44.77 | 99.63 | 97.86 | 96.54 | 94.72 | 99.07 | 98.73 | 98.14 | 14.57 | 7.28 | 7.06 | 7.46 | 8.15 | 27 |

Table 5: Comparison of ERM, PGD, and Corruption-trained models under Uniform, Gaussian, and Laplace noise. Reports Acc., PR, and GE at various perturbation radii $\gamma$ on CIFAR-10 (ResNet-18).

| Type | Method | Acc. % | $PR_{\mathcal{D}}^{Uniform}(\gamma)$ % | | | | $PR_{\mathcal{D}}^{Gaussian}(\gamma)$ % | | | | $PR_{\mathcal{D}}^{Laplace}(\gamma)$ % | | | | $GE_{PR_{\mathcal{D}}(\gamma=0.03)}$ % | | |
|---|---|---|---|---|---|---|---|---|---|---|---|---|---|---|---|---|---|
| | | | 0.03 | 0.08 | 0.1 | 0.12 | 0.03 | 0.08 | 0.1 | 0.12 | 0.03 | 0.08 | 0.1 | 0.12 | Uni. | Gau. | Lap. |
| Std | ERM | **94.85** | 97.64 | 76.19 | 61.65 | 47.07 | 96.16 | 61.88 | 44.2 | 30.6 | 96.04 | 60.98 | 43.24 | 29.79 | 6.24 | 6.36 | 6.3 |
| RT | Corr_Uniform | 94.17 | 99.12 | 90.92 | 82.32 | 70.48 | 98.69 | 82.46 | 67.4 | 49.32 | 98.67 | 81.85 | 66.29 | 48.09 | **4.83** | **5.03** | **5.05** |
| | Corr_Gaussian | 93.32 | 99.41 | 96.22 | 92.14 | 85.32 | 99.23 | 92.23 | 83.42 | 70.46 | 99.22 | 91.91 | 82.72 | 69.42 | 5.69 | 5.67 | 5.6 |
| | Corr_Laplace | 93.61 | 99.45 | 96.32 | 91.97 | 84.4 | 99.3 | 92.09 | 82.09 | 66.42 | 99.29 | 91.78 | 81.3 | 65.16 | 5.72 | 5.67 | 5.63 |
| AT | PGD | 83.83 | 99.63 | 97.89 | 96.59 | 94.85 | 99.48 | 96.68 | 94.5 | 91.54 | 99.47 | 96.59 | 94.34 | 91.29 | 11.4 | 11.24 | 11.21 |

distribution. As shown in Table 5 and Fig. 2 (d), PGD-based AT consistently achieves the highest PR across all distributions, outperforming Corruption in all settings.

**RT methods achieve higher clean accuracy but has near-zero improvement on AR** As shown in Table 4, models trained with RT methods exhibit smaller reductions in clean accuracy. However, their AR degrades significantly, with performance dropping to nearly zero under strong attacks such as C&W and Auto-Attack. Only the CVaR method achieves non-zero AR performance against PGD attacks, though it still lags behind models trained with AT methods. A similar observation is evident in Fig. 2 (a), where RT methods consistently appear along the leftmost vertical axis, indicating near-zero AR performance across all datasets and models. More results are shown in Table 10.

**AT-PR achieves a better trade-off among Accuracy, AR, and PR.** As shown in Table 4: *i)* it achieves comparable PR performance to AT methods; *ii)* it achieves AR slightly weaker than AT, yet still substantially higher than that of RT methods; *iii)* it reduces clean accuracy less severely than AT. Overall, although AT-PR is designed as a PR-targeted method, its AT-inspired formulation (min-max optimization) delivers a more balanced performance across the three metrics.

**Training efficiency varies widely.** Table 4 shows that Corruption method is highly efficient, generating only a single random sample per training example, whereas CVaR is computationally expensive due to the need for multiple samples to approximate the perturbation distribution. Standard AT methods exhibit intermediate and more consistent training times. AT-PR is the most costly, as it generates a set of PGD-based AEs and selects the one maximizing the all-AE region, leading to a substantial increase in training time (as noted by the complexity analysis in Zhang et al. (2025)). To further investigate efficiency-oriented approaches, we additionally integrate the filtering-based method of Chen & Lee (2024) into PRBench as a case study (Table 12). This aligns with ongoing work on efficient robustness, and our extensible design enables inclusion of future methods.

**AT methods lead in composite robustness evaluation.** To holistically assess training methods across all key aspects (AR, PR, GE, clean accuracy, and training efficiency), we introduce a *composite robustness score*. This score is computed as a (weighted) sum of Min–Max normalized metric values, with "lower-is-better" metrics (e.g., GE, training time) reversed for consistency. Fig. 2 (b) presents results under equal-weighting scheme, where AT methods consistently rank highest, corroborating earlier findings that they generally outperform PR-targeted training methods. Moreover, the KL-PGD achieves competitive performance, indicating that generating AEs with $\mathcal{L}_{KL}$ is more effective than with the traditional $\mathcal{L}_{CE}$. For additional results see Fig. 6.

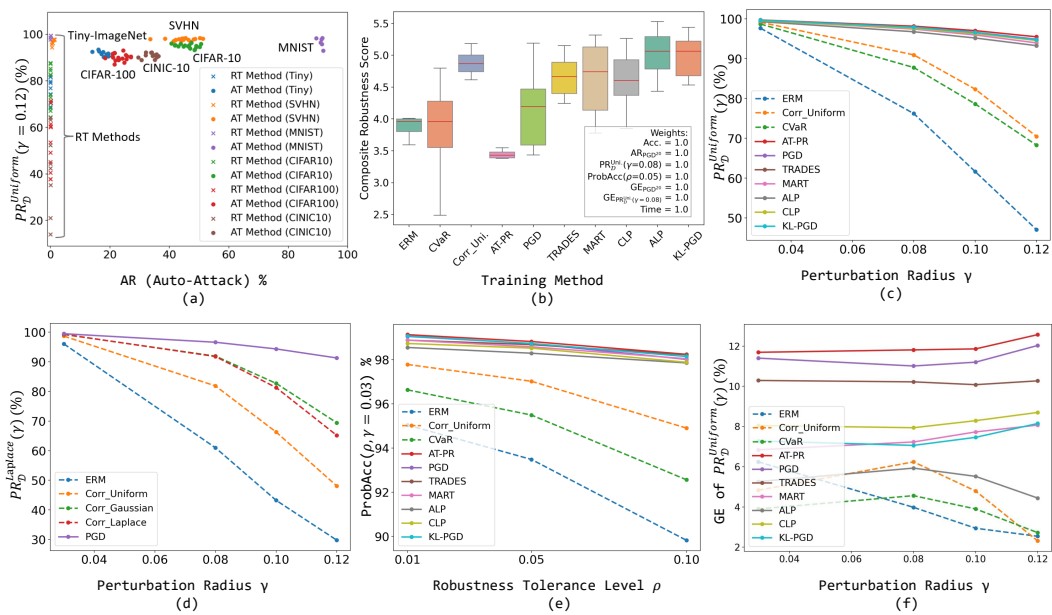

Figure 2: (a) Comparison of training methods (AT and RT) in terms of AR (AA) and PR ($PR_{\mathcal{D}}^{\text{Uniform}}(\gamma)$) performance across various datasets. (b) Composite robustness scores of different training methods, aggregated over all test datasets and model architectures. (c) $PR_{\mathcal{D}}^{\text{Uniform}}(\gamma)$ of ResNet-18 trained with different training methods on CIFAR-10 under varying $\gamma$. (d) $PR_{\mathcal{D}}^{\text{Laplace}}(\gamma)$ for ResNet-18 trained with corruption training and PGD models on CIFAR-10 across various $\gamma$. (e) $ProbAcc(\rho, \gamma = 0.03)$ for ResNet-18 trained with different training methods on CIFAR-10 with respect to different robustness tolerance level $\rho$. (f) GE of $PR_{\mathcal{D}}^{\text{Uniform}}(\gamma)$ for ResNet-18 trained with different training methods on CIFAR-10 with respect to different $\gamma$. More experimental results are deferred to Appendix D.

## 5.2 GENERALIZATION ERROR ANALYSIS

The empirical results reveal several insights into the GE of different training methods. In this section, we provide a theoretical analysis to support the empirical findings.

**RT methods consistently yield lower GE.** As shown in Table 4 and Fig. 2 (f), all RT methods empirically exhibit smaller GEs than AT methods across both AR and PR metrics. We next investigate the theoretical basis of this generalizability advantage in the Prop 1.

**Proposition 1** *Let $\mathcal{L}$ be the objective loss function for training without adversarial perturbation, and assume that $\mathcal{L}$ is $\varphi$-Lipschitz and $\psi$-smooth. Then, the objective function of the CVaR-based training scheme in Eq. 6 is also $\varphi$-Lipschitz and $\max\{\varphi, \psi\}$-smooth.*

Our Prop. 1 shows that RT learning yields a smoother training process than AT, as it excludes the GE term ($\phi$ in Thm. 1) associated with robust overfitting, resulting in a smaller overall GE. Training with corruption can be seen as a special case of CVaR and thus inherits the same Lipschitz and smooth properties. The proof of our Prop. 1 is in Appendix G.3.

As shown in Thm. 2, successful training with a properly chosen $\lambda$ can effectively reduce the $L_2$ distance between softmax outputs, denoted by $\nu$. This reduction pushes $\widetilde{\varphi} \approx \varphi$ and $\widetilde{\psi} \approx \psi$, and in cases where $\nu$ becomes sufficiently small, the training process may become even smoother, potentially satisfying $\widetilde{\phi} < \phi$. This provides a theoretical explanation for why AT methods that impose constraints on the softmax outputs $\mathbf{p}$ tend to exhibit smaller GEs. On the other hand, an improperly chosen $\lambda$ can result in highly non-smooth optimization. The detailed proof for our Thm. 2 is in Appendix G.2. We now provide theoretical insights into the following two empirical findings.

**PGD has the highest GE among AT methods:** Within AT methods (Table. 4), PGD results in the largest GE under the $GE_{AR}$ metric. This is due to the absence of an explicit penalty in its objective, unlike methods such as TRADES and MART, which incorporate additional penalties to stabilize

training and improve generalization. This empirical result corresponds to our theoretical analysis in Thm. 2. The AT with a penalty over the distance between softmax before and after perturbation effectively enhance the smoothness for the objective function. Additional results in Appendix. D.

**AT-PR exhibits similar GE to PGD.** AT-PR, though designed as a PR-targeted method, is AT-inspired: it selects AEs from a set of PGD-generated candidates corresponding to different local optima. Consequently, as shown in Table 4 & 10, its GE aligns closely with that of PGD. A detailed theoretical analysis with its corresponding algorithm is provided in Appendix G.4.

**Optimization budget explains the GE gap between ALP and CLP:** CLP shows higher GE than ALP, as ALP uses AEs in its optimization objective while CLP relies on clean inputs. This suggests that ALP's higher optimization budget leads to better generalization. According to Thm. 2, the generalization disadvantage of CLP arises from the choice of $\lambda$. To ensure robustness against adversarial attacks, CLP requires a relatively large value of $\lambda$, since it is based on $\mathcal{L}_{CE}(\mathbf{p}(\boldsymbol{x}, \boldsymbol{\theta}), y)$ instead of $\mathcal{L}_{CE}(\mathbf{p}(\boldsymbol{x} + \boldsymbol{\delta}, \boldsymbol{\theta}), y)$, cf. Table. 2. This leads to reduced smoothness and a larger GE.

## 6 CONCLUSION

We present PRBench, the first dedicated benchmark for PR, designed to standardize the evaluation of robustness training methods and advance our understanding of effective PR improvement strategies. Through a systematic evaluation of 222 models across diverse architectures, datasets, training methods, we identify two distinct trade-off frontiers. The first, represented by AT, emphasizes high AR and PR at the cost of clean accuracy. The second, represented by RT, prioritizes high clean accuracy, PR and lower GE (supported by theoretical analyses) while trading off AR performance.

Based on our findings in PRBench, we cautiously propose a *bold hypothesis*: there may be limited *practical* need for developing *separate* PR-targeted training methods, as strong AT methods already yield substantial PR improvements.

This hypothesis is not intended to claim that PR-targeted methods are obsolete. As noted in Sec. 1, PR remains an active research area with emerging studies on PR training, highlighting its practical relevance across different AI tasks. Our statement instead reflects a consistent empirical observation: strong AT methods (e.g., PGD, TRADES) can naturally deliver surprisingly strong PR performance, and AT-inspired PR methods such as AT-PR achieve an even more balanced robustness profile. These results suggest that the AT paradigm may *inherently* support PR, even without explicit PR-targeted modifications. At the same time, current AT methods still struggle to balance AR, PR, generalization, and clean accuracy. Methods such as KL-PGD and AT-PR demonstrate the potential of hybrid strategies, but they also reveal limitations, including the high computational cost of AT-PR. Therefore, we argue that future research may benefit more from improving versatile, AT-based approaches that jointly enhance AR and PR, rather than focusing *separately* on PR-targeted training techniques.

Our deliberately bold hypothesis is intended to spark constructive discussion and motivate rigorous future work, whether to validate, refine, or refute it, in line with the core motivation of PRBench, which is to support the PR research community through systematic and evidence-driven analysis.

## 7 ETHICS STATEMENT

This work does not involve human subjects, sensitive personal data, or proprietary information. It does not present foreseeable risks related to privacy, security, fairness, discrimination, or potential misuse. All datasets and methods used are publicly available and widely adopted in the research community. We therefore believe this work complies with the ICLR Code of Ethics.

## 8 REPRODUCIBILITY STATEMENT

All theoretical results are presented with clear assumptions and complete proofs in the Appendix. G. For datasets, we rely solely on publicly available benchmarks, with detailed descriptions and preprocessing steps included in the Appendix. B.4 along with proper citations. The implementation code for all experiments is included in the supplementary materials. Furthermore, the Appendix. B contains detailed descriptions of experimental settings, including hyperparameter choices, training configurations, and evaluation protocols. Together, these resources are intended to ensure faithful reproduction and enable further validation of our findings.

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

## A  LLM USAGE

In this work, Large Language Models (LLMs) were used solely as auxiliary tools for grammar correction and language polishing during the preparation of the manuscript. They did not contribute to research ideation, experimental design, implementation, analysis, or any scientific content. All technical ideas, theoretical results, and experimental findings are entirely the work of the authors.

## B  HYPERPARAMETER SELECTION AND IMPLEMENTATION DETAILS

We provide details on hyperparameter selection and computational setup. All experiments are conducted using a total of four NVIDIA H100 GPUs. The complete codebase is are included in the supplementary materials, a public repository will be released after the review process.

### B.1  EXPERIMENT SETUP

For the MNIST dataset, we employ a four-layer CNN, consisting of two convolutional layers followed by two fully connected layers, as described in Robey et al. (2022). For the CIFAR-10, CIFAR-100, CINIC-10 and SVHN datasets, we independently train VGG-19, ResNet-18, and WideResNet-28-10 on each dataset, and additionally include Vision Transformer (ViT) and Data-efficient Image Transformer (Deit) for CIFAR-10. For the TinyImageNet dataset, we train ResNet-18 and ResNet-34 using the same training configuration as for CIFAR-10. For the ImageNet-50 dataset, we train a ResNet-18 model separately. All models are trained using stochastic gradient descent (SGD) with a momentum of 0.9 and a weight decay coefficient of $3.5 \times 10^{-3}$. Training is performed for 100 epochs (200 for ImageNet-50) with an initial learning rate of 0.01, which is decayed by a factor of 10 at epochs 75 and 90, following the setup in Wang et al. (2019); Robey et al. (2022).

### B.2  TRAINING ALGORITHMS HYPERPARAMETER SETTING

- **PGD.** We set the perturbation radius to $\gamma = 8/255$ for all dataset except MNIST. During training, we performed 10 steps of projected gradient descent attack, using a step size of $\alpha = 2/255$ for CIFAR-10, CIFAR-100, CINIC-10, TinyImageNet and ImageNet-50, and a step size of $\alpha = 1.25/255$ for SVHN. For the MNIST dataset, the perturbation radius is set to $\gamma = 0.3$ with an attack step size of $\alpha = 0.1$, as MNIST is a relatively easier task that requires a larger budget, particularly for random perturbations.

- **TRADES.** We used the same step size and number of steps as described above for PGD. Additionally, we applied a weight of $\lambda = 6.0$ for all datasets, following the approach in Zhang et al. (2019); Robey et al. (2022).

- **MART.** We used the same step size and number of steps as described above for PGD. Additionally, we applied a weight of $\lambda = 5.0$ for all datasets, following the approach in Wang et al. (2019).

- **ALP.** Follow the original work Kannan et al. (2018), we set $\lambda = 1$ for all datasets, except $\lambda = 0.01$ for SVHN, MNIST, ImageNet-50 and TinyImageNet.

- **CLP.** Following the same setting as ALP, we also set $\lambda = 1$ for all datasets, except $\lambda = 0.3$ for SVHN, MNIST, ImageNet-50 and TinyImageNet.

- **CVaR.** We set the number of perturbed samples drawn from the uniform distribution to 20, following the original setup in Robey et al. (2022).

- **Corruption.** We perform standard corruption training by sampling a perturbation from a uniform distribution for each training point. Additionally, we conduct corruption training using two other perturbation distributions: Gaussian and Laplace for comparison, with all perturbations constrained within the specified norm-ball radius.

- **AT-PR.** The number of PGD-based AE candidates is set to 5 while other parameters follow the original setup in Zhang et al. (2025).

- **Chen & Lee (2024).** Following the standard efficient parameter configuration, we integrate it into TRADES with $m = 1\text{–}8$ and $k = 10$ (10-step attack).

## B.3 EVALUATION SETTING

All models are evaluated using three categories of metrics: PR, AR evaluation, and GE.

- **PR Evaluation:** We evaluate PR using two metrics: $PR_{\mathcal{D}}(\gamma)$, computed at four radii (8/255, 0.08, 0.1, and 0.12) to assess robustness under increasing perturbation levels (for MNIST, the radii are: 0.3, 0.35, 0.4, 0.45), and $ProbAcc(\rho)$, with three tolerance levels $\rho$ (0.01, 0.05, 0.1). PR evaluations are performed by sampling 100 points from the perturbation region for each test example.
- **AR Evaluation:** We evaluate AR using three main types of white-box attacks: PGD (with two different iteration steps $PGD^{10}$ and $PGD^{20}$), the C&W attack, and Auto-Attack.
- **Generalization Evaluation:** We report generalization performance by generalization error under both AR ($GE_{AR}$) and PR settings ($GE_{PR_{\mathcal{D}}(\gamma)}$).

## B.4 MODEL AND DATASET SELECTION

Table 6: Summary of model architectures and training paradigms used in `PRBench`.

| Category | Characteristics | Model | Size | Training Paradigm | |
| --- | --- | --- | --- | --- | --- |
| | | | | **Normal** | **Pre-trained** |
| Plain CNN | Sequential conv layers | 4-layer CNN | 1.1M | ✓ | |
| | | VGG-19 | 137M | ✓ | |
| Residual CNN | Residual connections | ResNet-18 | 11.7M | ✓ | |
| | | ResNet-34 | 21.8M | ✓ | |
| | | Wide-ResNet-28-10 | 36.5M | ✓ | |
| Transformer | Self-attention | ViT-Small | 22M | | ✓ |
| | | ViT-Base | 86M | | ✓ |
| | | ViT-Large | 307M | | ✓ |
| | | DeiT-Tiny | 5M | | ✓ |
| | | DeiT-Small | 22M | | ✓ |

Table 7: Summary of datasets used in `PRBench`.

| Dataset | Training Size | Test Size | Resolution | Number of Classes |
| --- | --- | --- | --- | --- |
| MNIST | 60,000 | 10,000 | 28×28 | 10 |
| SVHN | 73,257 | 26,032 | 32×32 | 10 |
| CIFAR-10 | 50,000 | 10,000 | 32×32 | 10 |
| CIFAR-100 | 50,000 | 10,000 | 32×32 | 100 |
| CINIC-10 | 90,000 | 90,000 | 32×32 | 10 |
| TinyImageNet | 100,000 | 10,000 | 64×64 | 200 |
| ImageNet-50 | 64,000 | 2,500 | 224×224 | 50 |

In `PRBench`, we consider three major model architectures: (1) **plain CNNs**, such as VGG, representing early feedforward convolutional networks; (2) **residual CNNs**, exemplified by ResNet, which introduce skip connections to enable deeper architectures; and (3) **transformer-based models**, including Vision Transformer (ViT) Dosovitskiy et al. (2021) and Data-efficient Image Transformer (Deit) Touvron et al. (2021), which leverage self-attention mechanisms for image representation learning. In our experiments, we conduct different training methods (See Table. 2) across a diverse set of models: VGG-19, SimpleCNN (4-layer CNN), ResNet-18, ResNet-34, Wide-ResNet-28-10, ViT, and DeiT, covering various scales such as ViT-Small, ViT-Base, ViT-Large, DeiT-Tiny, and DeiT-Small. See Table 6 for details.

Table 7 presents the seven widely used benchmark datasets used by `PRBench`, each representing a different level of complexity and image characteristics.

- **MNIST Deng (2012)** is a classic dataset of handwritten digits, consisting of 60,000 training images and 10,000 test images. Each image is a 28x28 pixel grayscale image, and the dataset is categorized into 10 classes (0-9).

- **SVHN Netzer et al. (2011)** (Street View House Numbers) is a real-world dataset derived from Google Street View house numbers. It contains 73,257 training and 26,032 test images, each sized 32×32 and labeled into 10 digit classes. Compared to MNIST, SVHN is more challenging due to background clutter and variation, making it suitable for evaluating robustness under real-world conditions.

- **CIFAR-10 Krizhevsky et al. (2009)** is a widely used image classification dataset with 60,000 32x32 color images across 10 classes. The dataset is split into 50,000 training images and 10,000 test images. Compared to MNIST and SVHN, CIFAR-10 presents greater variability in content and background, making it a standard benchmark for evaluating generalization across diverse object classes.

- **CIFAR-100 Krizhevsky et al. (2009)** extends CIFAR-10 by increasing the number of classes to 100, with each class containing 600 images. The dataset is also composed of 60,000 images (50,000 for training and 10,000 for testing) with 32x32 size. Its finer-grained classification task poses a greater challenge, requiring models to generalize across a broader and more diverse set of object categories.

- **CINIC-10 Darlow et al. (2018)** is derived from CIFAR-10 and downsampled ImageNet, with all images resized to 32×32 resolution. It contains 270,000 images evenly split into training, validation, and test sets (90,000 each). Like CIFAR-10, it has 10 classes, but the larger scale and more diverse image sources make it a more challenging and realistic benchmark.

- **TinyImageNet Le & Yang (2015)** is a subset of the ImageNet dataset, containing 100,000 training, 10,000 validation, and 10,000 test images, all resized to 64×64 pixels and spanning 200 object classes. With higher resolution and a larger number of categories, TinyImageNet poses a more complex classification challenge, making it well-suited for evaluating model scalability and robustness in realistic settings.

- **ImageNet-50 Deng et al. (2009)** is a 50-class subset of ImageNet for efficiency. It consists of approximately 65,000 training images and 2,500 validation images at 224×224 resolution. Despite being smaller than full ImageNet, ImageNet-50 retains considerable diversity and complexity, serving as a scalable benchmark for robustness evaluation.

### B.5    RELATED BENCHMARKS

Several robustness platforms have been developed to support AR evaluation by implementing popular attack methods, such as FoolBox Rauber et al. (2017), AdverTorch Ding et al. (2019), Cleverhan Papernot et al. (2016a), AdvBox Goodman et al. (2020), ART Nicolae et al. (2018), SecML Melis et al. (2019), DeepRobust Li et al. (2020), etc. In addition, some AR benchmarks have also been established Ling et al. (2019); Guo et al. (2023); Croce et al. (2021); Tang et al. (2021); Dong et al. (2020); Liu et al. (2025). Despite these efforts, existing work remains confined to AR evaluation, overlooking the more practical perspective of PR. Consequently, they fail to reveal the inherent relationship between AR and PR, thereby hindering the establishment of a holistic and unified understanding of robustness progress. Compared to the aforementioned benchmarks, PRBench provides a more comprehensive framework in several key aspects: (1) it is the first benchmark to emphasize the PR, providing a formalized definition of PR along with a summary of commonly used evaluation metrics and associated training methods; (2) it includes an extensive evaluation of widely used models and datasets, covering representative training methods designed for both AR and PR, enabling direct comparisons under PR metrics and revealing the performance of these methods in both adversarial and probabilistic scenarios; and (3) it presents in-depth analyses of the generalization error for PR from empirical and theoretical perspectives, which, to the best of our knowledge, has not been explored in prior benchmarks. These insights help improve DL model robustness in practical applications by offering a more complete view from both adversarial and probabilistic perspectives.

## C    RELATED WORK ON GENERALIZATION AND ROBUST OVERFITTING

Several lines of research have studied robust overfitting and generalization error in DL models. We compare our work with four relevant studies that examine generalization from different perspectives and theoretical foundations.

- **Scope and goal:**

1. Jiang et al. (2020) investigate generalization in standard deep learning models, focusing on the empirical correlation between complexity measures (e.g., VC-dimension, norm-based, PAC-Bayes) and generalization. Their goal is to uncover potential causal relationships between these measures and generalization, and to examine how reliably these measures can predict generalization behavior.

2. Kim et al. (2023) conduct a large-scale empirical study of robust generalization under AT, analyzing how margin-based, smoothness-based, flatness-based, and gradient-based measures correlate with the GE.

3. Yin et al. Yin et al. (2019) study adversarial robust generalization through Rademacher complexity, providing theoretical insights into why robust generalization is more challenging.

4. Dziugaite et al. (2020) examine how evaluation methodologies, such as those used by Jiang et al. (2020), can obscure the successes and failures of different generalization measures, proposing a distributional robustness framework for more comprehensive assessment.

5. In contrast, our study focuses on PR, aiming to evaluate improvements in PR achieved by various robustness training methods and to provide theoretical analysis of the GE across different training objectives.

- **Theoretical foundation:**

   1. Jiang et al. (2020) base their analysis on a broad set of theoretically motivated complexity measures from generalization theory (e.g., VC-dimension, norm-based, PAC-Bayes).

   2. Kim et al. (2023) extend this framework to AT by examining margin-based, smoothness-based, and flatness-based measures to study their relationship with the robust generalization gap.

   3. Yin et al. (2019) employ Rademacher complexity to theoretically analyze adversarial robust generalization.

   4. Dziugaite et al. (2020) complement Jiang et al. (2020) methodology by using a distributional robustness framework to evaluate generalization across diverse experimental settings.

   5. In our work, we leverage the concept of $\eta$-approximate $\beta$-smoothness Xiao et al. (2022b) to analyze how optimization objectives of different training methods affect GE, highlighting connections between optimization and generalization under various robustness regimes.

- **Model and dataset coverage:**

   1. Jiang et al. (2020) evaluate convolutional networks trained on CIFAR-10 and SVHN.

   2. Kim et al. (2023) examine ResNet-18, WRN28-10 and WRN34-10 on CIFAR-10.

   3. Yin et al. (2019) analyze linear classifiers and a four-layer ReLU network on MNIST.

   4. Dziugaite et al. (2020) also examine convolutional networks on CIFAR-10 and SVHN.

   5. In contrast, our work covers three families of model architectures: (1) plain CNNs, (2) residual networks, and (3) transformer-based models. We train a diverse set of models, including VGG-19, SimpleCNN, ResNet-18/34, WRN-28-10, ViT (Small/Base/Large), and DeiT (Tiny/Small), across seven datasets: MNIST, SVHN, CIFAR-10, CIFAR-100, CINIC-10, TinyImageNet, and ImageNet-50.

# D    SUPPLEMENTARY EXPERIMENTS

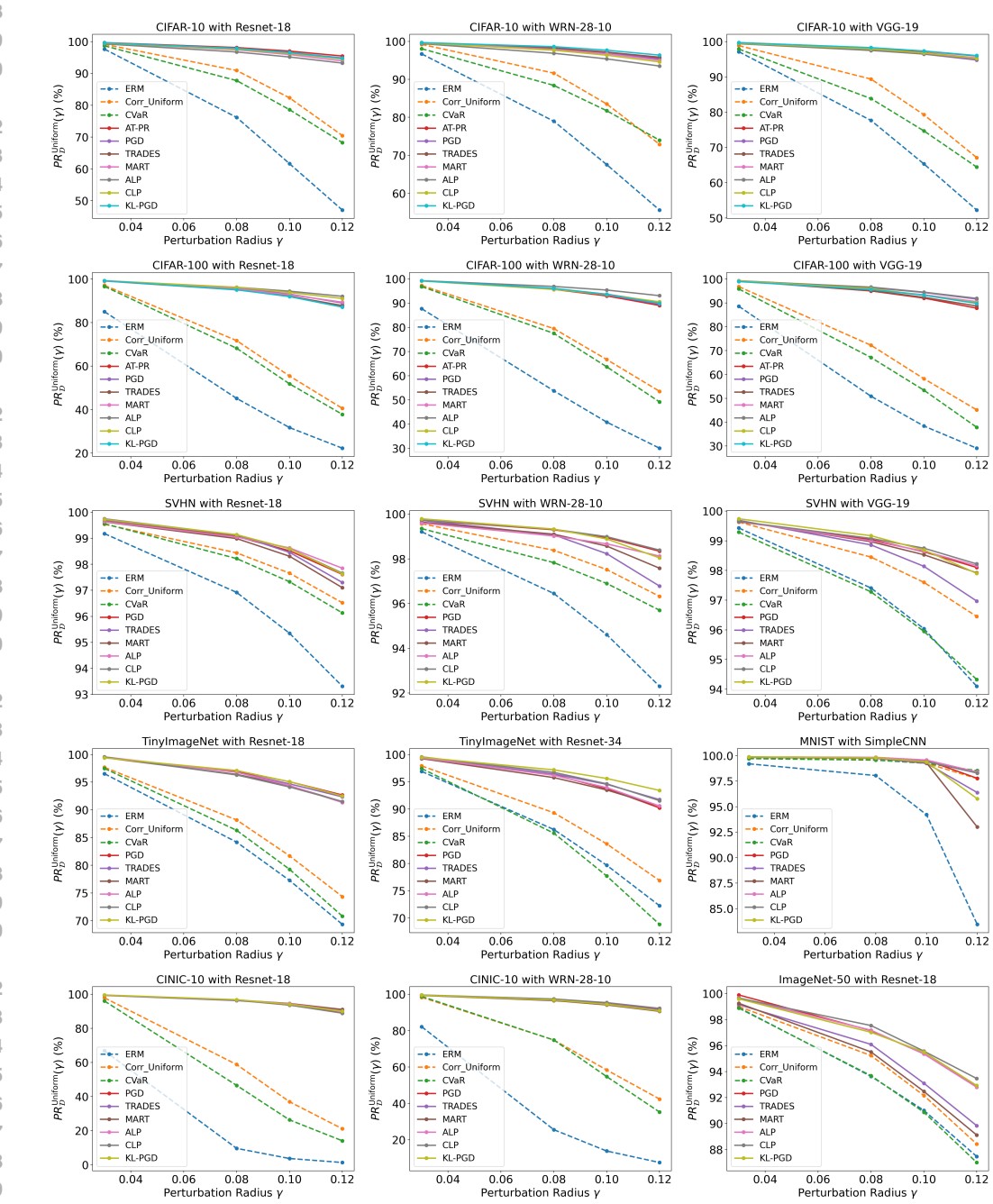

Figure 3: $PR_{\mathcal{D}}^{\text{Uniform}}(\gamma)$ for different models (ResNet-18, ResNet-34, WRN-28-10, VGG-19 and SimpleCNN) trained with various training methods both AT and PR-targeted on different datasets (CIFAR-10, CIFAR-100, CINIC-10, SVHN, MNIST, TinyImageNet, ImageNet-50), evaluated under varying perturbation radii $\gamma$.

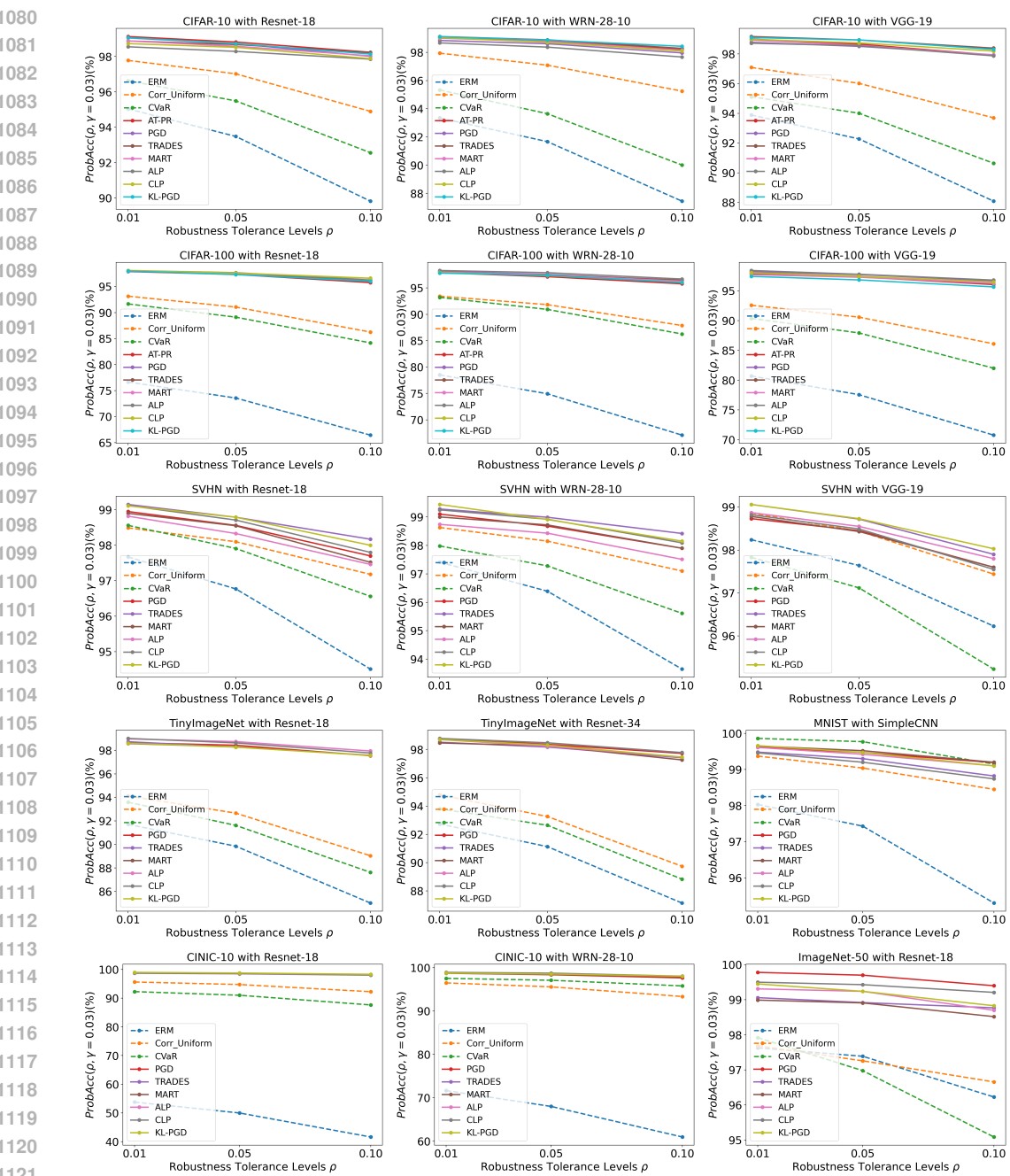

Figure 4: *ProbAcc*($\rho, \gamma = 0.03$) for different models (ResNet-18, ResNet-34, WRN-28-10, VGG-19) trained with various training methods both AT and PR-targeted on different datasets (CIFAR-10, CIFAR-100, CINIC-10, SVHN, MNIST, TinyImageNet, ImageNet-50), evaluated under varying robustness tolerance level $\rho$.

# E  PR-TARGETED TRAINING METHOD

Some recent works have focused on designing training techniques specifically to improve PR. Wang et al. (2021) proposed a simple augmentation-based approach referred to as *corruption training*, where the model is trained on randomly perturbed examples. This process is equivalent to standard AT training, except that each input $x_i$ is perturbed by a sample drawn from a uniform distribution within an $\ell_\infty$ norm ball of radius $\gamma$, rather than using worst-case AEs. They show that corruption training

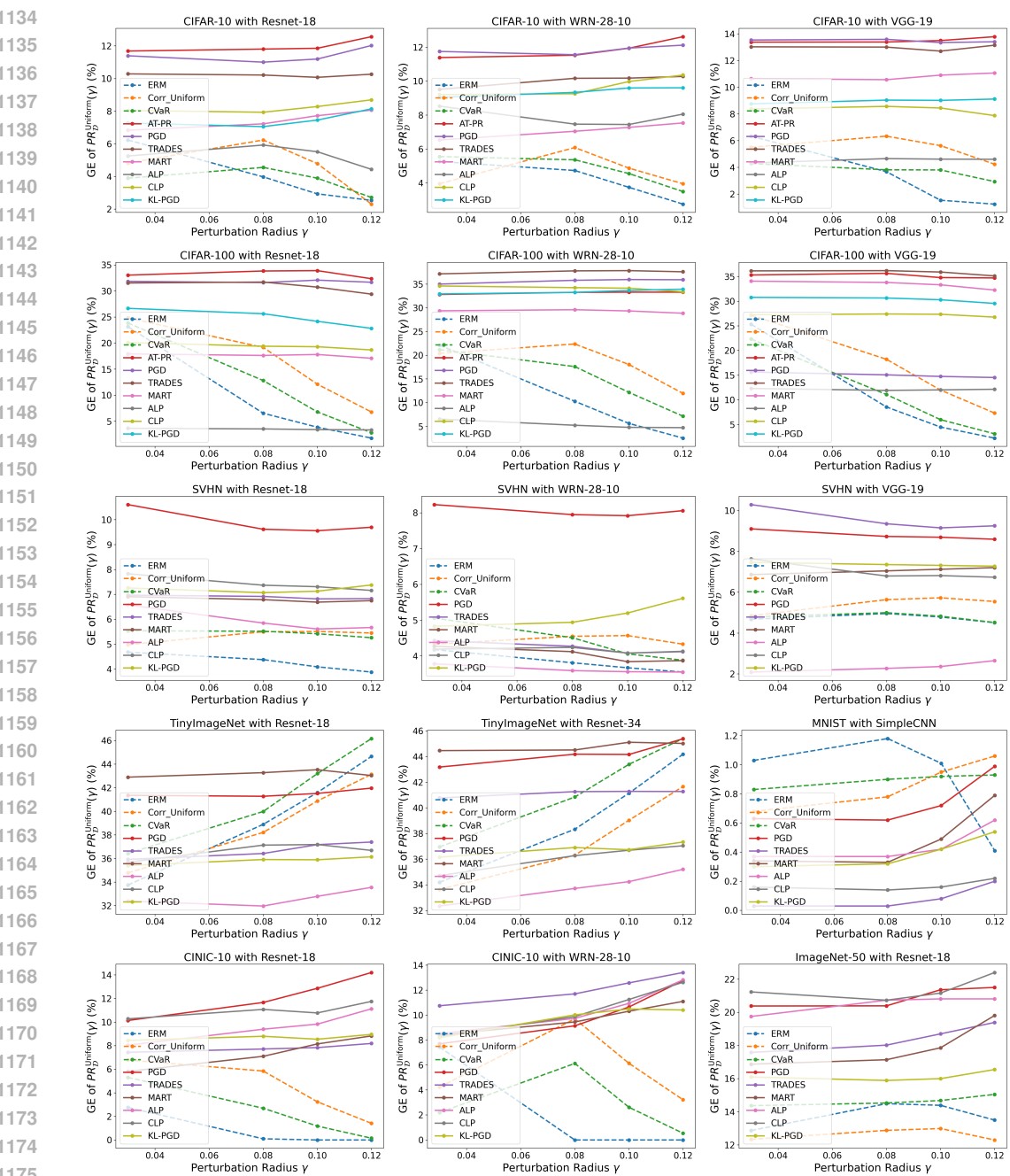

Figure 5: GE of $PR_{\mathcal{D}}^{\text{Uniform}}(\gamma)$ for different models (ResNet-18, ResNet-34, WRN-28-10, VGG-19 and SimpleCNN) trained with training methods both AT and PR-targeted on different datasets (CIFAR-10, CIFAR-100, CINIC-10, SVHN, MNIST, TinyImageNet, ImageNet-50), evaluated under varying perturbation radii $\gamma$.

improves PR and reduces generalization error under PR evaluation settings. The corresponding optimization objective is formalized as:

$$\min_{\boldsymbol{\theta}} \mathbb{E}_{(\boldsymbol{x},y)\sim\mathcal{D}} \left[ \mathcal{L}(\boldsymbol{x}+\boldsymbol{\delta}, y; \boldsymbol{\theta}) \right]; \boldsymbol{\delta} \sim Pr(\cdot \mid \boldsymbol{x}). \tag{18}$$

Later, Robey et al. (2022) introduced an optimization objective based on the *Conditional Value-at-Risk (CVaR)* to improve PR. Specifically, for a loss function $\mathcal{L}$ and a continuous distribution $Pr$, CVaR

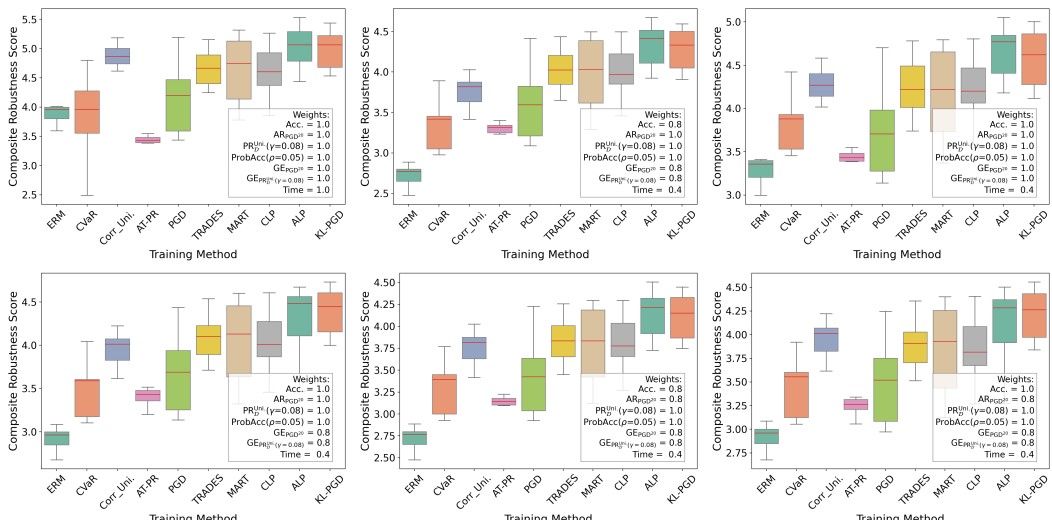

Figure 6: Composite robustness scores of different training methods, aggregated over all datasets and model architectures, with varying weight assignments for 7 metrics: clean accuracy (Acc.), $AR_{\text{PGD}^{20}}$, $PR_{\mathcal{D}}^{\text{Uniform}}(\gamma = 0.08)$, $ProbAcc(\rho = 0.05)$, $\text{GE}_{\text{PGD}^{20}}$, $\text{GE}_{PR_{\mathcal{D}}^{\text{Uniform}}(\gamma=0.08)}$, and training time (s/ep.) (In practice, task-specific priorities may vary—e.g., safety-critical applications may emphasize AR, while consumer-facing systems may prioritize clean accuracy—thus motivating unequal weightings across metrics).

Table 8: Evaluation results of ViT models trained with different training methods on CIFAR-10, reporting clean accuracy (Acc.), AR (PGD / C&W / Auto-Attack), PR ($PR_{\mathcal{D}}^{\text{Uniform}}(\gamma)/ProbAcc(\rho, \gamma = 0.03)$), $\text{GE}_{AR}$, $\text{GE}_{PR_{\mathcal{D}}^{\text{Uniform}}(\gamma)}$, and per-epoch training time.

| Model | Method | Acc.% | AR % | | | | $PR_{\mathcal{D}}^{\text{Uniform}}(\gamma)$ % | | | | $ProbAcc(\rho, \gamma = 0.03)$ % | | | $\text{GE}_{AR}$ % | $\text{GE}_{PR_{\mathcal{D}}^{\text{Uniform}}(\gamma)}$ % | | | | Time |
|---|---|---|---|---|---|---|---|---|---|---|---|---|---|---|---|---|---|---|---|
| | | | $PGD^{10}$ | $PGD^{20}$ | $CW^{20}$ | AA | 0.03 | 0.08 | 0.1 | 0.12 | 0.1 | 0.05 | 0.01 | $PGD^{20}$ | 0.03 | 0.08 | 0.1 | 0.12 | s/ep. |
| DeiT-Ti | ERM | **95.92** | 0.0 | 0.0 | 0.0 | 0.0 | 98.23 | 87.39 | 80.72 | 73.17 | 95.9 | 94.82 | 92.24 | **0.0** | 5.3 | **4.95** | **3.43** | **2.96** | 13 |
| | Corr_Uniform | 95.58 | 0.01 | 0.01 | 0.01 | 0.0 | 99.26 | 92.39 | 86.07 | 78.07 | 98.04 | 97.5 | 95.8 | 0.0 | 5.28 | 6.34 | 5.2 | 4.13 | 13 |
| | PGD | 77.8 | 47.26 | 46.57 | **44.95** | 42.71 | 99.68 | 98.33 | 97.47 | 96.43 | 99.06 | 98.8 | 98.24 | 7.31 | 6.21 | 5.64 | 5.94 | 5.95 | 120 |
| | TRADES | 78.94 | 48.15 | 47.81 | 44.3 | **43.68** | 99.54 | 98.02 | 96.99 | 95.65 | 98.83 | 98.63 | 98.08 | 7.35 | 5.82 | 5.59 | 5.74 | 5.97 | 116 |
| | MART | 72.34 | **48.28** | **47.85** | 43.4 | 41.91 | 99.63 | 98.16 | 96.31 | 94.85 | 98.93 | 98.67 | 98.22 | 6.67 | 5.8 | 5.88 | 5.77 | 5.54 | 114 |
| | CLP | 72.03 | 46.33 | 46.06 | 43.36 | 42.52 | **99.69** | **98.57** | **97.81** | **96.81** | **99.11** | **98.81** | 98.4 | 4.51 | 5.33 | 5.43 | 5.4 | 5.36 | 136 |
| | KL-PGD | 86.99 | 45.4 | 44.59 | 43.44 | 41.78 | **99.74** | 98.5 | 97.58 | 96.38 | 99.33 | 99.08 | **98.41** | 18.08 | 9.57 | 9.31 | 9.13 | 8.9 | 136 |
| DeiT-S | ERM | **97.28** | 0.23 | 0.04 | 0.06 | 0.0 | 98.91 | 92.9 | 88.76 | 83.65 | 97.5 | 96.7 | 94.75 | **0.0** | 2.65 | 3.77 | 3.43 | 3.02 | 14 |
| | Corr_Uniform | 96.69 | 0.51 | 0.06 | 0.13 | 0.0 | 99.42 | 95.8 | 92.6 | 88.06 | 98.41 | 97.93 | 96.69 | 0.0 | 2.92 | 4.39 | 4.55 | 4.16 | 14 |
| | PGD | 86.16 | 45.24 | 43.74 | 44.57 | 41.90 | 99.73 | 98.56 | 97.75 | 96.63 | 99.21 | 98.94 | 98.33 | 41.25 | 12.8 | 13.32 | 13.4 | 13.8 | 178 |
| | TRADES | 81.66 | 50.59 | 50.15 | 47.45 | 46.49 | 99.64 | 98.12 | 96.94 | 95.55 | 99.12 | 98.9 | 98.42 | 18.14 | 10.99 | 11.29 | 11.42 | 11.56 | 162 |
| | MART | 79.53 | **51.29** | 50.55 | 47.01 | 45.32 | 99.68 | 98.57 | 97.76 | 96.74 | 99.16 | 98.92 | 98.4 | 13.8 | 6.18 | 6.38 | 6.87 | 7.38 | 152 |
| | CLP | 80.63 | 51.24 | **50.65** | **48.79** | **47.25** | 99.72 | **98.66** | **97.92** | **96.91** | **99.48** | **99.32** | **99.01** | 13.98 | 7.09 | 7.68 | 7.85 | 7.9 | 208 |
| | KL-PGD | 88.23 | 46.23 | 44.88 | 44.55 | 42.73 | **99.74** | 98.6 | 97.76 | 96.54 | 99.25 | 99.01 | 98.56 | 30.0 | 10.37 | 10.8 | 10.72 | 10.87 | 188 |
| ViT-S | ERM | 95.1 | 0.04 | 0.01 | 0.01 | 0.0 | 98.83 | 93.08 | 89.73 | 85.92 | 97.07 | 96.08 | 93.68 | **0.0** | 4.72 | 6.36 | **5.84** | **4.95** | 24 |
| | Corr_Uniform | 94.76 | 0.14 | 0.02 | 0.04 | 0.0 | 99.2 | 94.94 | 92.02 | 88.4 | 97.72 | 97.04 | 95.35 | 0.0 | **4.54** | **6.29** | 6.51 | 6.21 | 24 |
| | PGD | 82.62 | 41.59 | 40.55 | 40.74 | 38.49 | **99.7** | **98.72** | **98.1** | 97.18 | **99.07** | **98.86** | 98.16 | 39.58 | 14.68 | 14.61 | 14.77 | 15.15 | 304 |
| | TRADES | 76.97 | 46.35 | 46.07 | 42.83 | 42.23 | 99.65 | 98.45 | 97.73 | 96.81 | 98.96 | 98.7 | **98.24** | 18.71 | 13.57 | 13.48 | 13.48 | 13.76 | 280 |
| | MART | 72.38 | **47.51** | **46.97** | **42.89** | 41.20 | 99.67 | 98.6 | 97.96 | 97.23 | 99.02 | 98.77 | 98.2 | 9.56 | 8.25 | 8.38 | 8.38 | 8.19 | 256 |
| | CLP | 74.54 | 47.0 | 46.75 | 42.83 | **43.63** | 99.62 | 98.62 | 98.08 | **97.35** | 99.02 | 98.83 | 98.13 | 9.75 | 7.97 | 8.38 | 8.46 | 8.45 | 360 |
| | KL-PGD | 85.22 | 41.78 | 40.42 | 40.54 | 38.77 | 99.68 | 98.69 | 98.02 | 97.19 | 99.03 | 98.77 | 98.31 | 36.3 | 13.17 | 13.92 | 14.25 | 14.57 | 324 |
| ViT-B | ERM | **98.3** | 0.6 | 0.19 | 0.12 | 0.0 | 98.92 | 92.38 | 87.54 | 81.72 | 97.51 | 96.72 | 94.57 | **0.0** | 1.36 | **0.67** | **0.87** | **1.13** | **40** |
| | Corr_Uniform | 98.04 | 0.62 | 0.15 | 0.17 | 0.0 | 99.51 | 95.88 | 92.47 | 87.96 | 98.61 | 98.03 | 96.72 | 0.0 | 1.68 | 2.13 | 2.15 | 2.1 | 40 |
| | PGD | 87.38 | 46.28 | 44.7 | 45.42 | 42.62 | 99.7 | 98.55 | 97.81 | 96.8 | 99.08 | 98.87 | 98.41 | 35.82 | 10.23 | 9.99 | 10.02 | 10.56 | 330 |
| | TRADES | 83.99 | **52.03** | **51.04** | **48.29** | **46.95** | 99.62 | 97.92 | 96.84 | 95.46 | 99.06 | 98.86 | 98.5 | 10.87 | 4.77 | 4.4 | 4.62 | 5.13 | 295 |
| | MART | 76.12 | 50.73 | 50.28 | 45.74 | 43.99 | 99.7 | 98.3 | 97.49 | 96.47 | 99.22 | 98.9 | 98.61 | 5.74 | 3.77 | 4.15 | 4.26 | 4.04 | 278 |
| | CLP | 77.45 | 49.07 | 48.69 | 46.09 | 44.97 | 99.75 | 98.62 | 97.9 | 96.89 | 99.29 | 99.1 | 98.55 | 6.17 | 4.57 | 4.67 | 4.71 | 4.78 | 390 |
| | KL-PGD | 89.64 | 47.92 | 46.04 | 44.92 | 41.37 | **99.78** | **98.78** | **98.03** | **97.1** | **99.38** | **99.12** | **98.73** | 19.96 | 7.54 | 7.82 | 7.7 | 7.71 | 324 |
| ViT-L | ERM | **98.46** | 2.12 | 0.67 | 0.42 | 0.0 | 99.3 | 95.23 | 92.13 | 88.07 | 98.43 | 97.8 | 96.34 | 0.02 | 1.65 | **2.31** | **1.92** | **1.26** | 82 |
| | Corr_Uniform | 98.04 | 3.49 | 1.36 | 1.17 | 0.0 | 99.67 | 97.11 | 94.58 | 90.92 | 99.11 | 98.81 | 97.93 | **0.0** | **1.54** | 2.57 | 2.77 | 2.67 | 82 |
| | PGD | 87.85 | 46.27 | 45.24 | 45.82 | 43.05 | 99.75 | 98.86 | 98.27 | 97.46 | 99.32 | 99.11 | 98.46 | 51.55 | 10.88 | 11.32 | 11.51 | 12.1 | 950 |
| | TRADES | 85.35 | 51.12 | 49.71 | 49.35 | 46.89 | 99.77 | 98.78 | 98.09 | 97.18 | 99.34 | **99.13** | 98.65 | 34.97 | 13.82 | 13.96 | 13.92 | 13.95 | 845 |
| | MART | 83.63 | **51.45** | **50.09** | 47.84 | 45.6 | 99.72 | 98.89 | 98.23 | 97.33 | 99.28 | 99.08 | **98.74** | 24.35 | 8.33 | 8.23 | 8.37 | 8.47 | 800 |
| | CLP | 83.55 | 51.09 | 49.87 | **49.85** | **47.54** | **99.78** | **98.97** | **98.43** | **97.69** | **99.38** | 99.1 | 98.55 | 29.15 | 11.5 | 11.29 | 11.37 | 11.44 | 1124 |
| | KL-PGD | 86.24 | 44.21 | 43.04 | 42.73 | 41.09 | 99.71 | 98.65 | 97.94 | 96.97 | 99.13 | 98.96 | 98.42 | 26.85 | 12.42 | 12.16 | 11.94 | 11.9 | 918 |

can be interpreted as the expected value of $\mathcal{L}$ over the tail of the distribution. It admits the following convex, variational characterization, which allows it to be optimized via stochastic gradient-based methods. Based on this, they formalized the training objective as:

$$\min_{\boldsymbol{\theta}} \; \mathbb{E}_{(\boldsymbol{x},y)\sim\mathcal{D}} \left[ \text{CVaR}_{1-\rho} \left( \mathcal{L}(\boldsymbol{x} + \boldsymbol{\delta}, y; \boldsymbol{\theta}); \boldsymbol{\delta} \sim Pr(\cdot \mid \boldsymbol{x}) \right) \right];$$

$$\text{CVaR}_{\rho}(\mathcal{L}; Pr) = \inf_{\alpha\in\mathbb{R}} \left[ \alpha + \frac{\mathbb{E}_{\boldsymbol{\delta}\sim Pr(\cdot|\boldsymbol{x})} \left[ (\mathcal{L}(\boldsymbol{x} + \boldsymbol{\delta}) - \alpha)_+ \right]}{1 - \rho} \right], \tag{19}$$

Table 9: Comparison of standard training (ERM), corruption training with Uniform, Gaussian, and Laplace perturbations (PR-targeted method), and PGD training (standard AT) models. Evaluation is conducted in terms of clean accuracy (Acc.), $PR_{\mathcal{D}}(\gamma)$ and $GE_{PR_{\mathcal{D}}(\gamma=0.03)}$ under various perturbation distributions (Uniform, Gaussian, Laplace).

| Data | Model | Method | Acc. % | $PR_{\mathcal{D}}^{\text{Uniform}}(\gamma)$ % | | | | $PR_{\mathcal{D}}^{\text{Gaussian}}(\gamma)$ % | | | | $PR_{\mathcal{D}}^{\text{Laplace}}(\gamma)$ % | | | | $GE_{PR_{\mathcal{D}}(\gamma=0.03)}$ % | | |
|---|---|---|---|---|---|---|---|---|---|---|---|---|---|---|---|---|---|---|
| | | | | 0.03 | 0.08 | 0.1 | 0.12 | 0.03 | 0.08 | 0.1 | 0.12 | 0.03 | 0.08 | 0.1 | 0.12 | Uni. | Gau. | Lap. |
| CIFAR-10 | VGG-19 | ERM | **93.12** | 97.15 | 77.65 | 65.33 | 52.22 | 95.5 | 65.4 | 49.26 | 35.62 | 95.38 | 64.66 | 48.32 | 34.81 | 6.34 | 6.48 | 6.52 |
| | | Corr_Uniform | 92.94 | 98.87 | 89.38 | 79.27 | 67.1 | 98.33 | 79.38 | 63.87 | 48.29 | 98.3 | 78.67 | 62.93 | 47.34 | 5.51 | 5.82 | 5.87 |
| | | Corr_Gaussian | 92.18 | 99.26 | 95.34 | 90.44 | 82.66 | 99.05 | 90.56 | 80.4 | 66.89 | 99.02 | 90.19 | 79.66 | 65.84 | 6.85 | 6.72 | 6.69 |
| | | Corr_Laplace | 91.96 | 99.35 | 95.36 | 90.42 | 82.5 | 99.09 | 90.52 | 80.2 | 66.76 | 99.08 | 90.13 | 79.52 | 65.69 | 6.59 | 6.47 | 6.47 |
| | | PGD | 80.43 | **99.51** | **97.71** | **96.47** | **94.79** | **99.35** | **96.55** | **94.52** | **92.05** | **99.33** | **96.48** | **94.35** | **91.83** | 13.54 | 13.62 | 13.58 |
| | ResNet-18 | ERM | **94.85** | 97.64 | 76.19 | 61.65 | 47.07 | 96.16 | 61.88 | 44.2 | 30.6 | 96.04 | 60.98 | 43.24 | 29.79 | 6.24 | 6.36 | 6.3 |
| | | Corr_Uniform | 94.17 | 99.12 | 90.92 | 82.32 | 70.48 | 98.69 | 82.46 | 67.4 | 49.32 | 98.67 | 81.85 | 66.29 | 48.09 | 4.83 | 5.03 | 5.05 |
| | | Corr_Gaussian | 93.32 | 99.41 | 96.22 | 92.14 | 85.32 | 99.23 | 92.23 | 83.42 | 70.46 | 99.22 | 91.91 | 82.72 | 69.42 | 5.69 | 5.67 | 5.6 |
| | | Corr_Laplace | 93.61 | 99.45 | 96.32 | 91.97 | 84.4 | 99.3 | 92.09 | 82.09 | 66.42 | 99.29 | 91.78 | 81.3 | 65.16 | 5.72 | 5.67 | 5.63 |
| | | PGD | 83.83 | **99.63** | **97.89** | **96.59** | **94.85** | **99.48** | **96.68** | **94.5** | **91.54** | **99.47** | **96.59** | **94.34** | **91.29** | 11.4 | 11.24 | 11.21 |
| | WRN | ERM | **95.55** | 96.7 | 78.99 | 67.55 | 55.55 | 95.15 | 67.93 | 53.33 | 41.02 | 95.05 | 67.12 | 52.51 | 40.35 | 5.27 | 5.91 | 5.86 |
| | | Corr_Uniform | 95.11 | 99.2 | 91.63 | 83.5 | 72.89 | 98.81 | 83.79 | 70.62 | 56.52 | 98.79 | 83.25 | 69.77 | 55.54 | 3.96 | 4.23 | 4.12 |
| | | Corr_Gaussian | 94.83 | 99.45 | 96.78 | 93.48 | 87.71 | 99.25 | 93.64 | 86.27 | 75.6 | 99.25 | 93.36 | 85.7 | 74.76 | 5.07 | 5.08 | 5.01 |
| | | Corr_Laplace | 94.69 | 99.45 | 96.76 | 93.31 | 87.38 | 99.29 | 93.45 | 85.9 | 75.1 | 99.27 | 93.2 | 85.3 | 74.28 | 4.14 | 4.31 | 4.22 |
| | | PGD | 86.66 | **99.6** | **98.14** | **96.94** | **95.34** | **99.48** | **97.03** | **95.06** | **92.36** | **99.47** | **96.94** | **94.91** | **92.11** | 11.75 | 11.74 | 11.69 |
| | Deit-T | ERM | **95.92** | 98.23 | 87.39 | 80.72 | 73.17 | 97.27 | 81.01 | 71.57 | 61.4 | 97.17 | 80.61 | 71.05 | 60.71 | 5.3 | 5.84 | 5.82 |
| | | Corr_Uniform | 95.58 | 99.26 | 92.39 | 86.07 | 78.07 | 98.92 | 86.22 | 76.18 | 64.73 | 98.9 | 85.77 | 75.51 | 63.92 | 5.28 | 5.36 | 5.33 |
| | | Corr_Gaussian | 93.86 | 99.42 | 96.57 | 93.23 | 87.71 | 99.22 | 93.35 | 86.34 | 76.78 | 99.2 | 93.11 | 85.83 | 76.06 | 6.6 | 6.54 | 6.48 |
| | | Corr_Laplace | 93.90 | 99.45 | 96.58 | 93.3 | 88.03 | 99.29 | 93.4 | 86.77 | 76.9 | 99.29 | 93.17 | 86.3 | 76.13 | 5.7 | 5.61 | 5.54 |
| | | PGD | 77.8 | **99.68** | **98.33** | **97.47** | **96.43** | **99.56** | **97.6** | **96.38** | **94.71** | **99.55** | **97.55** | **96.29** | **94.61** | 6.21 | 6.27 | 6.27 |
| | Deit-S | ERM | **97.28** | 98.91 | 92.9 | 88.76 | 83.65 | 98.39 | 88.94 | 82.5 | 74.64 | 98.34 | 88.66 | 82.09 | 74.07 | 2.65 | 2.94 | 3.0 |
| | | Corr_Uniform | 96.69 | 99.42 | 95.8 | 92.6 | 88.06 | 99.19 | 92.69 | 86.94 | 79.35 | 99.17 | 92.47 | 86.54 | 78.73 | 2.92 | 2.91 | 2.94 |
| | | Corr_Gaussian | 96.48 | 99.6 | 97.61 | 95.47 | 92.16 | 99.45 | 95.57 | 91.28 | 85.17 | 99.44 | 95.41 | 90.94 | 84.67 | 3.11 | 3.06 | 3.05 |
| | | Corr_Laplace | 96.51 | 99.53 | 97.53 | 95.52 | 92.17 | 99.39 | 95.62 | 91.31 | 85.13 | 99.38 | 95.44 | 90.98 | 84.62 | 3.35 | 3.38 | 3.35 |
| | | PGD | 86.16 | **99.73** | **98.56** | **97.75** | **96.63** | **99.63** | **97.82** | **96.42** | **94.49** | **99.62** | **97.77** | **96.33** | **94.34** | 12.8 | 12.81 | 12.75 |
| | ViT-S | ERM | 95.1 | 98.83 | 93.09 | 89.74 | 85.9 | 98.3 | 89.93 | 85.3 | 80.08 | 98.27 | 89.72 | 85.01 | 79.66 | 4.72 | 5.05 | 5.02 |
| | | Corr_Uniform | 94.76 | 99.22 | 94.98 | 91.99 | 88.44 | 98.9 | 92.18 | 87.78 | 82.6 | 98.87 | 91.96 | 87.53 | 82.27 | 4.54 | 4.72 | 4.68 |
| | | Corr_Gaussian | 94.58 | 99.44 | 96.93 | 94.75 | 91.89 | 99.26 | 94.89 | 91.32 | 86.93 | 99.26 | 94.74 | 91.1 | 86.61 | 4.8 | 4.75 | 4.76 |
| | | Corr_Laplace | 94.43 | 99.44 | 96.95 | 94.72 | 91.83 | 99.25 | 94.84 | 91.22 | 86.97 | 99.23 | 94.68 | 91.04 | 86.62 | 5.07 | 4.93 | 4.88 |
| | | PGD | 82.62 | **99.69** | **98.74** | **98.08** | **97.19** | **99.59** | **98.14** | **97.05** | **95.68** | **99.58** | **98.09** | **96.97** | **95.59** | 14.68 | 14.71 | 14.7 |
| | ViT-B | ERM | **98.3** | 98.92 | 92.38 | 87.54 | 81.72 | 98.4 | 87.68 | 80.36 | 72.34 | 98.37 | 87.38 | 79.93 | 71.68 | 1.36 | 1.34 | 1.33 |
| | | Corr_Uniform | 98.04 | 99.51 | 95.88 | 92.47 | 87.96 | 99.28 | 92.62 | 87.0 | 80.16 | 99.26 | 92.39 | 86.61 | 79.67 | 1.68 | 1.78 | 1.75 |
| | | Corr_Gaussian | 97.71 | 99.57 | 97.73 | 95.41 | 91.43 | 99.44 | 95.56 | 90.48 | 83.16 | 99.44 | 95.4 | 90.07 | 82.57 | 1.89 | 1.88 | 1.87 |
| | | Corr_Laplace | 97.55 | 99.65 | 97.85 | 95.7 | 92.15 | 99.52 | 95.84 | 91.42 | 85.2 | 99.52 | 95.66 | 91.09 | 84.66 | 1.76 | 1.75 | 1.71 |
| | | PGD | 87.38 | **99.7** | **98.55** | **97.81** | **96.8** | **99.57** | **97.89** | **96.64** | **94.99** | **99.56** | **97.85** | **96.54** | **94.84** | 10.23 | 10.2 | 10.17 |
| CIFAR-100 | VGG-19 | ERM | **73.86** | 88.61 | 50.86 | 38.37 | 29.06 | 83.48 | 38.47 | 27.5 | 20.37 | 83.16 | 37.79 | 27.02 | 20.0 | 25.33 | 22.79 | 22.62 |
| | | Corr_Uniform | 71.88 | 96.85 | 72.26 | 58.22 | 45.13 | 95.05 | 58.29 | 42.45 | 30.25 | 94.93 | 57.51 | 41.59 | 29.5 | 27.18 | 27.23 | 27.12 |
| | | Corr_Gaussian | 69.85 | 98.06 | 86.84 | 75.43 | 61.44 | 97.42 | 75.55 | 58.33 | 42.13 | 97.39 | 74.8 | 57.3 | 41.0 | 29.0 | 29.01 | 28.97 |
| | | Corr_Laplace | 70.08 | 98.08 | 86.5 | 74.21 | 60.48 | 97.38 | 74.44 | 57.34 | 41.98 | 97.33 | 73.71 | 56.32 | 41.06 | 28.45 | 28.58 | 28.62 |
| | | PGD | 48.76 | **99.34** | **96.43** | **94.4** | **91.63** | **99.07** | **94.54** | **91.04** | **86.48** | **99.04** | **94.38** | **90.81** | **86.08** | 15.64 | 15.58 | 15.56 |
| | Res-18 | ERM | **76.03** | 85.01 | 45.08 | 31.7 | 22.19 | 79.56 | 31.79 | 20.53 | 13.82 | 79.11 | 31.16 | 20.05 | 13.44 | 23.17 | 20.78 | 20.56 |
| | | Corr_Uniform | 74.93 | 97.09 | 71.69 | 55.41 | 40.53 | 95.55 | 55.57 | 37.73 | 25.17 | 95.45 | 54.65 | 36.86 | 24.47 | 24.84 | 25.0 | 24.95 |
| | | Corr_Gaussian | 73.36 | 98.23 | 87.51 | 75.4 | 60.52 | 97.62 | 75.55 | 57.21 | 40.43 | 97.59 | 74.76 | 56.05 | 39.35 | 26.63 | 26.51 | 26.45 |
| | | Corr_Laplace | 73.18 | 98.15 | 87.12 | 75.06 | 60.09 | 97.59 | 75.37 | 56.86 | 39.63 | 97.56 | 74.51 | 55.67 | 38.6 | 25.54 | 25.66 | 25.59 |
| | | PGD | 58.09 | **99.16** | **95.82** | **93.08** | **89.0** | **98.9** | **93.35** | **88.31** | **81.39** | **98.89** | **93.12** | **87.9** | **80.72** | 31.86 | 31.75 | 31.82 |
| | WRN | ERM | **79.43** | 87.68 | 53.79 | 40.87 | 30.16 | 83.1 | 41.39 | 28.55 | 19.83 | 82.82 | 40.68 | 27.89 | 19.33 | 22.13 | 21.97 | 22.04 |
| | | Corr_Uniform | 79.01 | 97.31 | 79.5 | 66.72 | 53.57 | 96.1 | 66.93 | 50.93 | 37.01 | 96.01 | 66.22 | 50.09 | 36.18 | 20.4 | 20.63 | 20.58 |
| | | Corr_Gaussian | 77.24 | 98.39 | 90.28 | 82.16 | 70.55 | 97.83 | 82.46 | 67.97 | 52.19 | 97.79 | 81.83 | 67.04 | 51.09 | 22.97 | 22.89 | 22.88 |
| | | Corr_Laplace | 76.99 | 98.53 | 90.48 | 82.49 | 71.42 | 98.04 | 82.83 | 68.96 | 54.02 | 98.02 | 82.23 | 68.06 | 53.05 | 22.7 | 22.61 | 22.63 |
| | | PGD | 61.11 | **99.2** | **96.01** | **93.45** | **89.59** | **98.93** | **93.62** | **88.85** | **81.86** | **98.92** | **93.37** | **88.47** | **81.27** | 35.03 | 35.0 | 34.98 |
| SVHN | VGG-19 | ERM | 95.78 | 99.43 | 97.41 | 96.04 | 94.1 | 99.26 | 96.04 | 93.7 | 90.66 | 99.24 | 95.96 | 93.5 | 90.4 | 4.68 | 4.7 | 4.71 |
| | | Corr_Uniform | **95.79** | 99.63 | 98.45 | 97.6 | 96.45 | 99.52 | 97.66 | 96.16 | 94.1 | 99.51 | 97.57 | 96.05 | 93.94 | 4.85 | 4.94 | 4.94 |
| | | Corr_Gaussian | 95.71 | 99.71 | 98.97 | 98.42 | 97.7 | 99.64 | 98.47 | 97.55 | 96.17 | 99.63 | 98.44 | 97.47 | 96.05 | 5.07 | 5.12 | 5.13 |
| | | Corr_Laplace | 95.84 | 99.7 | 98.89 | 98.32 | 97.53 | 99.6 | 98.36 | 97.33 | 95.89 | 99.59 | 98.32 | 97.27 | 95.76 | 4.89 | 4.95 | 4.92 |
| | | PGD | 89.76 | **99.64** | **99.04** | **98.63** | **98.1** | **99.54** | **98.68** | **98.0** | **96.81** | **99.54** | **98.65** | **97.94** | **96.67** | 9.11 | 9.09 | 9.05 |
| | Res-18 | ERM | **96.26** | 99.18 | 96.92 | 95.35 | 93.31 | 98.97 | 95.42 | 92.88 | 89.6 | 98.97 | 95.34 | 92.7 | 89.34 | 4.68 | 4.74 | 4.7 |
| | | Corr_Uniform | 96.22 | 99.54 | 98.44 | 97.66 | 96.52 | 99.42 | 97.69 | 96.25 | 94.26 | 99.41 | 97.62 | 96.16 | 94.03 | 4.98 | 5.03 | 5.04 |
| | | Corr_Gaussian | 96.23 | 99.74 | 98.98 | 98.44 | 97.76 | 99.68 | 98.5 | 97.59 | 96.33 | 99.65 | 98.45 | 97.52 | 96.15 | 4.79 | 4.86 | 4.89 |
| | | Corr_Laplace | 96.24 | 99.67 | 98.92 | 98.39 | 97.74 | 99.59 | 98.45 | 97.6 | 96.33 | 99.59 | 98.4 | 97.52 | 96.22 | 4.71 | 4.77 | 4.76 |
| | | PGD | 90.22 | **99.67** | **99.07** | **98.52** | **97.61** | **99.61** | **98.55** | **97.45** | **95.51** | **99.6** | **98.5** | **97.34** | **95.38** | 10.61 | 10.48 | 10.49 |
| | WRN | ERM | 96.16 | 99.21 | 96.45 | 94.61 | 92.3 | 98.94 | 94.68 | 91.81 | 88.27 | 98.94 | 94.54 | 91.59 | 87.99 | 4.18 | 4.16 | 4.12 |
| | | Corr_Uniform | 96.56 | 99.57 | 98.38 | 97.52 | 96.32 | 99.45 | 97.55 | 96.04 | 93.94 | 99.44 | 97.49 | 95.97 | 93.8 | 4.33 | 4.38 | 4.36 |
| | | Corr_Gaussian | 96.69 | 99.72 | 98.94 | 98.44 | 97.74 | 99.62 | 98.46 | 97.55 | 96.29 | 99.62 | 98.45 | 97.55 | 96.2 | 4.22 | 4.22 | 4.21 |
| | | Corr_Laplace | **96.72** | 99.75 | 99.01 | 98.5 | 97.8 | 99.67 | 98.51 | 97.64 | 96.28 | 99.67 | 98.49 | 97.6 | 96.17 | 4.11 | 4.13 | 4.17 |
| | | PGD | 91.91 | **99.73** | **99.31** | **98.95** | **98.33** | **99.68** | **98.99** | **98.18** | **96.74** | **99.68** | **98.95** | **98.15** | **96.59** | 8.23 | 8.21 | 8.14 |
| Tiny-ImageNet | Res-18 | ERM | **65.92** | 96.53 | 84.18 | 77.28 | 69.34 | 95.35 | 77.92 | 68.29 | 57.86 | 95.29 | 77.49 | 67.74 | 57.12 | 33.76 | 34.2 | 34.17 |
| | | Corr_Uniform | 65.53 | 97.68 | 88.18 | 81.71 | 74.34 | 97.09 | 82.39 | 73.46 | 63.37 | 97.01 | 82.01 | 72.84 | 62.67 | 34.79 | 34.8 | 34.89 |
| | | Corr_Gaussian | 65.59 | 98.13 | 91.78 | 86.28 | 79.72 | 97.79 | 86.81 | 78.79 | 69.8 | 97.74 | 86.44 | 78.3 | 69.12 | 34.89 | 34.6 | 34.56 |
| | | Corr_Laplace | 64.95 | 98.06 | 91.46 | 85.95 | 79.06 | 97.64 | 86.1 | 78.1 | 69.06 | 97.6 | 85.64 | 77.55 | 67.85 | 33.42 | 33.24 | 33.28 |
| | | PGD | 50.11 | **99.4** | **97.0** | **95.07** | **92.69** | **99.2** | **95.21** | **92.32** | **87.66** | **99.19** | **95.1** | **92.06** | **87.17** | 41.36 | 41.39 | 41.4 |
| | ResNet-34 | ERM | 66.47 | 96.9 | 86.24 | 79.68 | 72.25 | 95.83 | 80.3 | 71.37 | 61.59 | 95.76 | 79.97 | 70.89 | 60.84 | 34.2 | 34.44 | 34.39 |
| | | Corr_Uniform | **66.84** | 97.9 | 89.31 | 83.6 | 76.87 | 97.28 | 84.01 | 76.0 | 66.89 | 97.25 | 83.72 | 75.54 | 66.23 | 33.64 | 33.34 | 33.41 |
| | | Corr_Gaussian | 66.71 | 98.38 | 92.86 | 87.94 | 81.8 | 98.03 | 88.39 | 80.77 | 72.26 | 97.98 | 88.04 | 80.34 | 71.59 | 34.43 | 34.37 | 34.33 |
| | | Corr_Laplace | 66.48 | 98.52 | 92.64 | 87.82 | 81.67 | 98.12 | 88.39 | 80.71 | 72.19 | 98.07 | 88.05 | 80.24 | 71.51 | 33.25 | 33.1 | 33.12 |
| | | PGD | 51.61 | **99.46** | **96.28** | **93.73** | **90.2** | **99.24** | **94.01** | **89.74** | **84.14** | **99.25** | **93.8** | **89.45** | **83.65** | 43.19 | 43.29 | 43.29 |
| MNIST | Simple-CNN | ERM | 99.35 | 99.2 | 98.05 | 94.21 | 83.45 | 99.07 | 98.2 | 96.34 | 92.47 | 99.02 | 98.07 | 96.05 | 91.96 | **1.03** | **1.04** | **1.04** |
| | | Corr_Uniform | 99.38 | 99.8 | 99.66 | 99.25 | 97.75 | 99.76 | 99.62 | 99.34 | 98.76 | 99.76 | 99.6 | 99.31 | 98.68 | 0.68 | 0.72 | 0.72 |
| | | Corr_Gaussian | **99.41** | 99.83 | 99.77 | 99.69 | 99.4 | 99.81 | 99.75 | 99.65 | 99.48 | 99.81 | 99.75 | 99.64 | 99.45 | 0.67 | 0.68 | 0.67 |
| | | Corr_Laplace | 99.36 | 99.86 | 99.81 | **99.72** | **99.43** | 99.84 | 99.79 | 99.7 | **99.52** | 99.84 | **99.78** | 99.69 | **99.48** | 0.69 | 0.7 | 0.7 |
| | | PGD | 99.15 | **99.88** | **99.83** | 99.56 | 97.78 | **99.85** | 99.7 | 98.91 | 90.42 | **99.85** | 99.68 | 98.7 | 87.7 | 0.63 | 0.63 | 0.64 |

Table 10: Comparison of training methods, evaluated by clean accuracy, AR (PGD / C&W / Auto-Attack), PR ($PR_{\mathcal{D}}^{\text{Uniform}}(\gamma)$ / $ProbAcc(\rho, \gamma = 0.03)$), $GE_{AR}$, $GE_{PR_{\mathcal{D}}^{\text{Uniform}}(\gamma)}$ , and training time (s/epoch).

| Data | Model | Method | Acc. % | AR % $PGD^{10}$ | $PGD^{20}$ | $CW^{20}$ | AA | $PR_{\mathcal{D}}^{\text{Uniform}}(\gamma)$ % 0.03 | 0.08 | 0.1 | 0.12 | ProbAcc($\rho,\gamma$=0.03) % 0.1 | 0.05 | 0.01 | $GE_{AR}$ % $PGD^{20}$ | $GE_{PR}$ % 0.03 | 0.08 | 0.1 | 0.12 | Time s/ep. |
|---|---|---|---|---|---|---|---|---|---|---|---|---|---|---|---|---|---|---|---|---|
| CIFAR-10 | ResNet-18 | ERM | 94.85 | 0.01 | 0.0 | 0.0 | 0.0 | 97.64 | 76.19 | 61.65 | 47.07 | 95.04 | 93.48 | 89.82 | 0.0 | 6.24 | 3.98 | 2.94 | 2.54 | 3 |
| | | Corr_Uniform | 94.17 | 0.28 | 0.05 | 0.02 | 0.0 | 99.12 | 90.92 | 82.32 | 70.48 | 97.78 | 97.02 | 94.9 | 0.04 | 4.83 | 6.24 | 4.79 | 2.31 | 3 |
| | | CVaR | 89.91 | 41.79 | 33.45 | 0.0 | 0.0 | 98.67 | 87.75 | 78.62 | 68.27 | 96.63 | 95.49 | 92.56 | 1.13 | 3.9 | 4.56 | 3.9 | 2.72 | 61 |
| | | AT-PR | 86.35 | 48.22 | 46.53 | 47.39 | 44.01 | 99.68 | 98.13 | 97.01 | 95.46 | 99.13 | 98.82 | 98.24 | 27.43 | 11.69 | 11.81 | 11.86 | 12.57 | 160 |
| | | PGD | 83.83 | 50.86 | 49.46 | 49.18 | 46.34 | 99.63 | 97.89 | 96.59 | 94.85 | 99.05 | 98.73 | 98.01 | 25.02 | 11.4 | 11.01 | 11.2 | 12.03 | 30 |
| | | TRADES | 83.34 | 54.09 | 53.15 | 50.84 | 48.99 | 99.55 | 97.74 | 96.33 | 94.55 | 98.88 | 98.68 | 98.22 | 16.12 | 10.29 | 10.22 | 10.08 | 10.27 | 25 |
| | | MART | 82.26 | 54.83 | 53.84 | 49.63 | 46.94 | 99.56 | 97.4 | 95.92 | 93.93 | 98.88 | 98.58 | 98.04 | 17.4 | 6.84 | 7.23 | 7.73 | 8.07 | 22 |
| | | ALP | 73.98 | 54.41 | 54.08 | 50.72 | 48.41 | 99.27 | 96.73 | 95.16 | 93.24 | 98.55 | 98.29 | 97.85 | 8.44 | 5.26 | 5.93 | 5.52 | 4.44 | 36 |
| | | CLP | 81.47 | 54.12 | 53.34 | 51.05 | 49.05 | 99.53 | 97.61 | 96.29 | 94.54 | 98.73 | 98.52 | 97.88 | 15.32 | 8.05 | 7.94 | 8.29 | 8.7 | 36 |
| | | KL-PGD | 87.55 | 49.4 | 48.43 | 47.06 | 44.77 | 99.63 | 97.86 | 96.54 | 94.72 | 99.07 | 98.73 | 98.14 | 14.57 | 7.28 | 7.06 | 7.46 | 8.15 | 27 |
| | WRN-28-10 | ERM | 95.55 | 0.01 | 0.0 | 0.0 | 0.0 | 96.7 | 78.97 | 67.55 | 55.57 | 93.35 | 91.67 | 87.45 | 0.0 | 5.27 | 4.74 | 3.75 | 2.76 | 15 |
| | | Corr_Uniform | 95.11 | 4.16 | 0.67 | 0.65 | 0.0 | 99.2 | 91.63 | 83.5 | 72.89 | 97.94 | 97.09 | 95.25 | 0.15 | 3.96 | 6.09 | 4.88 | 3.96 | 15 |
| | | CVaR | 87.81 | 20.7 | 15.71 | 0.58 | 0.0 | 98.06 | 88.4 | 81.74 | 73.97 | 95.33 | 93.65 | 90.01 | 1.44 | 5.56 | 5.37 | 4.56 | 3.5 | 313 |
| | | AT-PR | 87.29 | 48.15 | 46.42 | 48.16 | 44.23 | 99.66 | 98.25 | 97.19 | 95.46 | 99.09 | 98.79 | 98.19 | 34.06 | 11.39 | 11.53 | 11.94 | 12.62 | 780 |
| | | PGD | 86.66 | 50.91 | 49.24 | 50.29 | 46.71 | 99.59 | 98.14 | 96.94 | 95.33 | 98.84 | 98.61 | 97.96 | 33.26 | 11.75 | 11.56 | 11.95 | 12.12 | 165 |
| | | TRADES | 86.66 | 55.34 | 54.14 | 52.71 | 50.66 | 99.67 | 98.23 | 97.16 | 95.83 | 99.12 | 98.89 | 98.28 | 26.18 | 9.52 | 10.17 | 10.18 | 10.28 | 129 |
| | | MART | 86.15 | 55.77 | 54.36 | 51.21 | 49.02 | 99.63 | 97.88 | 96.73 | 95.02 | 99.0 | 98.77 | 98.15 | 23.5 | 6.54 | 7.05 | 7.28 | 7.54 | 114 |
| | | ALP | 75.61 | 54.37 | 53.14 | 52.29 | 48.58 | 99.39 | 96.86 | 95.38 | 93.5 | 98.67 | 98.37 | 97.67 | 12.16 | 8.53 | 7.47 | 7.45 | 8.06 | 194 |
| | | CLP | 83.86 | 55.04 | 53.87 | 53.02 | 50.18 | 99.59 | 97.66 | 96.35 | 94.59 | 99.04 | 98.71 | 98.09 | 22.92 | 9.21 | 9.26 | 9.98 | 10.36 | 194 |
| | | KL-PGD | 89.13 | 51.22 | 50.24 | 49.19 | 47.38 | 99.67 | 98.6 | 97.67 | 96.37 | 99.11 | 98.87 | 98.43 | 22.35 | 9.05 | 9.34 | 9.6 | 9.61 | 135 |
| | VGG19 | ERM | 93.12 | 0.01 | 0.01 | 0.0 | 0.0 | 97.15 | 77.65 | 65.33 | 52.22 | 93.89 | 92.28 | 88.09 | 0.0 | 6.34 | 3.68 | 1.54 | 1.25 | 1 |
| | | Corr_Uniform | 92.94 | 0.17 | 0.11 | 0.18 | 0.0 | 98.87 | 89.38 | 79.27 | 67.1 | 97.09 | 96.02 | 93.7 | 0.0 | 5.51 | 6.34 | 5.63 | 4.24 | 1 |
| | | CVaR | 86.18 | 38.12 | 34.88 | 0.02 | 0.01 | 97.9 | 83.84 | 74.69 | 64.42 | 95.12 | 94.01 | 90.64 | 1.0 | 4.29 | 3.82 | 3.81 | 2.94 | 40 |
| | | AT-PR | 83.91 | 42.05 | 39.94 | 41.57 | 37.30 | 99.62 | 98.08 | 96.87 | 95.38 | 99.0 | 98.66 | 97.92 | 21.55 | 13.38 | 13.39 | 13.5 | 13.79 | 100 |
| | | PGD | 80.43 | 48.4 | 47.15 | 46.15 | 43.04 | 99.51 | 97.71 | 96.47 | 94.79 | 98.78 | 98.51 | 97.94 | 16.76 | 13.54 | 13.59 | 13.35 | 13.41 | 20 |
| | | TRADES | 80.29 | 48.43 | 47.41 | 45.62 | 43.84 | 99.73 | 98.27 | 97.3 | 95.94 | 99.17 | 98.94 | 98.39 | 20.47 | 13.03 | 13.01 | 12.71 | 13.15 | 16 |
| | | MART | 76.65 | 49.82 | 48.93 | 44.25 | 41.99 | 99.57 | 97.95 | 96.78 | 95.39 | 98.93 | 98.58 | 97.92 | 17.99 | 10.65 | 10.57 | 10.91 | 11.07 | 14 |
| | | ALP | 60.22 | 46.68 | 46.55 | 43.04 | 41.81 | 99.4 | 97.5 | 96.49 | 95.19 | 98.72 | 98.55 | 97.87 | 3.21 | 4.33 | 4.66 | 4.62 | 4.61 | 24 |
| | | CLP | 78.65 | 50.38 | 49.62 | 47.92 | 45.23 | 99.55 | 97.93 | 96.78 | 95.23 | 98.98 | 98.74 | 98.19 | 11.25 | 8.38 | 8.57 | 8.45 | 7.88 | 24 |
| | | KL-PGD | 86.01 | 46.01 | 45.08 | 42.76 | 40.66 | 99.7 | 98.3 | 97.35 | 96.04 | 99.1 | 98.94 | 98.3 | 12.41 | 8.76 | 9.04 | 9.02 | 9.12 | 18 |
| CIFAR-100 | ResNet-18 | ERM | 76.03 | 0.0 | 0.0 | 0.0 | 0.0 | 85.02 | 45.12 | 31.7 | 22.22 | 76.65 | 73.57 | 66.44 | 0.0 | 23.17 | 6.54 | 3.86 | 1.77 | 3 |
| | | Corr_Uniform | 74.93 | 0.07 | 0.01 | 0.0 | 0.0 | 97.09 | 71.69 | 55.41 | 40.53 | 93.14 | 91.05 | 86.25 | 0.0 | 24.84 | 19.18 | 12.12 | 6.78 | 3 |
| | | CVaR | 72.14 | 0.11 | 0.05 | 0.02 | 0.0 | 96.61 | 68.17 | 51.82 | 37.76 | 91.66 | 89.11 | 84.17 | 0.0 | 23.86 | 12.82 | 6.79 | 2.81 | 60 |
| | | AT-PR | 60.03 | 23.86 | 23.05 | 22.37 | 18.25 | 99.23 | 95.3 | 91.99 | 87.38 | 97.88 | 97.31 | 95.75 | 38.44 | 33.06 | 33.85 | 33.92 | 32.37 | 162 |
| | | PGD | 58.09 | 27.52 | 26.93 | 25.43 | 22.89 | 99.16 | 95.82 | 93.04 | 88.97 | 97.94 | 97.42 | 96.2 | 37.39 | 31.86 | 31.63 | 32.1 | 31.69 | 30 |
| | | TRADES | 59.65 | 30.63 | 30.15 | 26.47 | 24.84 | 99.08 | 95.02 | 92.09 | 87.8 | 97.99 | 97.36 | 96.16 | 26.6 | 31.55 | 31.73 | 30.76 | 29.41 | 25 |
| | | MART | 54.57 | 32.0 | 31.62 | 27.9 | 25.55 | 99.08 | 95.45 | 92.73 | 89.35 | 97.96 | 97.49 | 96.59 | 20.45 | 17.93 | 17.64 | 17.82 | 17.1 | 23 |
| | | ALP | 43.68 | 26.64 | 26.43 | 23.13 | 21.51 | 99.22 | 96.2 | 94.4 | 92.05 | 98.08 | 97.66 | 96.25 | 4.01 | 3.65 | 3.53 | 3.4 | 3.34 | 35 |
| | | CLP | 50.92 | 29.87 | 29.64 | 26.04 | 25.31 | 99.19 | 96.16 | 93.81 | 91.05 | 98.08 | 97.59 | 96.6 | 19.09 | 20.11 | 19.41 | 19.31 | 18.7 | 36 |
| | | KL-PGD | 64.87 | 26.18 | 25.44 | 23.19 | 21.51 | 99.21 | 95.06 | 91.9 | 87.07 | 97.97 | 97.29 | 96.01 | 19.9 | 26.68 | 25.65 | 24.19 | 22.84 | 28 |
| | WRN-28-10 | ERM | 79.43 | 0.01 | 0.01 | 0.0 | 0.0 | 87.73 | 53.8 | 40.87 | 30.08 | 78.51 | 74.93 | 67.09 | 0.0 | 22.13 | 10.28 | 5.61 | 2.51 | 15 |
| | | Corr_Uniform | 79.01 | 0.94 | 0.38 | 0.24 | 0.01 | 97.31 | 79.5 | 66.72 | 53.57 | 93.44 | 91.83 | 87.88 | 0.05 | 20.4 | 22.38 | 18.05 | 11.94 | 15 |
| | | CVaR | 71.95 | 0.37 | 0.17 | 0.03 | 0.0 | 96.85 | 77.52 | 63.71 | 49.19 | 93.21 | 90.93 | 86.27 | 0.03 | 21.18 | 17.61 | 12.17 | 7.13 | 314 |
| | | AT-PR | 61.74 | 23.32 | 22.54 | 22.56 | 20.92 | 99.2 | 95.77 | 92.99 | 89.04 | 97.89 | 97.14 | 95.79 | 42.27 | 32.84 | 33.28 | 33.33 | 33.33 | 780 |
| | | PGD | 61.11 | 28.15 | 27.32 | 26.26 | 24.16 | 99.2 | 96.01 | 93.45 | 89.59 | 98.11 | 97.57 | 96.41 | 46.11 | 35.03 | 35.83 | 35.99 | 35.95 | 165 |
| | | TRADES | 62.36 | 32.01 | 31.54 | 28.28 | 27.01 | 99.23 | 96.08 | 93.68 | 90.42 | 97.83 | 97.51 | 96.45 | 44.01 | 37.19 | 37.82 | 37.87 | 37.63 | 129 |
| | | MART | 59.38 | 32.89 | 32.37 | 28.75 | 27.03 | 99.2 | 95.86 | 93.47 | 90.14 | 97.99 | 97.46 | 96.28 | 37.31 | 29.39 | 29.62 | 29.36 | 28.87 | 114 |
| | | ALP | 42.01 | 29.12 | 29.07 | 25.41 | 24.48 | 99.28 | 96.91 | 95.41 | 93.09 | 98.31 | 97.9 | 96.69 | 11.25 | 6.53 | 5.22 | 4.78 | 4.72 | 195 |
| | | CLP | 54.87 | 30.23 | 29.94 | 26.49 | 25.81 | 99.15 | 95.72 | 93.49 | 90.56 | 97.93 | 97.33 | 96.06 | 36.84 | 34.63 | 34.28 | 34.16 | 33.42 | 195 |
| | | KL-PGD | 66.65 | 26.64 | 25.88 | 24.3 | 22.77 | 99.24 | 96.11 | 93.58 | 89.95 | 97.81 | 97.39 | 96.01 | 34.98 | 32.95 | 33.29 | 33.69 | 33.96 | 135 |
| | VGG19 | ERM | 73.86 | 2.03 | 1.09 | 0.71 | 0.0 | 88.61 | 50.86 | 38.37 | 29.06 | 80.7 | 77.57 | 70.78 | 0.3 | 25.33 | 8.56 | 4.48 | 2.24 | 1 |
| | | Corr_Uniform | 71.88 | 3.94 | 2.0 | 1.2 | 0.0 | 96.85 | 72.26 | 58.22 | 45.13 | 92.58 | 90.58 | 86.11 | 0.52 | 27.18 | 18.23 | 11.99 | 7.31 | 1 |
| | | CVaR | 64.49 | 9.45 | 6.95 | 0.13 | 0.0 | 95.88 | 67.12 | 53.4 | 37.82 | 90.37 | 87.93 | 82.02 | 1.14 | 22.34 | 11.09 | 5.98 | 3.11 | 40 |
| | | AT-PR | 61.98 | 23.59 | 22.54 | 22.2 | 19.70 | 99.2 | 95.04 | 92.04 | 87.81 | 97.88 | 97.33 | 96.05 | 31.48 | 35.33 | 35.66 | 34.8 | 34.73 | 103 |
| | | PGD | 48.76 | 24.76 | 24.18 | 23.35 | 20.91 | 99.34 | 96.43 | 94.4 | 91.63 | 98.21 | 97.74 | 96.68 | 15.65 | 15.64 | 15.07 | 14.75 | 14.53 | 20 |
| | | TRADES | 54.54 | 26.19 | 25.58 | 23.65 | 21.82 | 99.19 | 95.24 | 92.35 | 88.72 | 97.97 | 97.46 | 96.16 | 31.28 | 36.16 | 36.21 | 35.93 | 35.14 | 17 |
| | | MART | 49.3 | 25.73 | 25.38 | 22.64 | 20.88 | 98.96 | 95.64 | 93.42 | 90.56 | 97.71 | 97.28 | 96.25 | 36.3 | 34.08 | 33.81 | 33.33 | 32.27 | 14 |
| | | ALP | 45.85 | 25.61 | 25.49 | 23.51 | 21.57 | 99.26 | 96.64 | 94.53 | 91.92 | 98.41 | 97.78 | 96.8 | 13.97 | 12.31 | 11.92 | 12.02 | 12.15 | 24 |
| | | CLP | 52.16 | 24.39 | 23.92 | 22.31 | 21.03 | 99.26 | 96.02 | 93.26 | 90.07 | 98.03 | 97.44 | 96.49 | 22.42 | 27.27 | 27.42 | 27.36 | 26.78 | 24 |
| | | KL-PGD | 59.06 | 21.96 | 20.93 | 20.64 | 18.39 | 98.98 | 95.63 | 93.24 | 89.79 | 97.42 | 96.83 | 95.66 | 23.01 | 30.79 | 30.66 | 30.3 | 29.56 | 18 |
| Tiny-ImageNet | ResNet-18 | ERM | 65.92 | 0.02 | 0.01 | 0.0 | 0.0 | 96.53 | 84.18 | 77.28 | 69.34 | 91.71 | 89.85 | 85.02 | 0.0 | 33.76 | 38.9 | 41.59 | 44.66 | 21 |
| | | Corr_Uniform | 65.53 | 0.01 | 0.01 | 0.0 | 0.0 | 97.68 | 88.18 | 81.71 | 74.34 | 94.12 | 92.66 | 89.04 | 0.0 | 34.79 | 38.22 | 40.88 | 43.14 | 21 |
| | | CVaR | 62.7 | 0.0 | 0.0 | 0.0 | 0.0 | 97.48 | 86.31 | 79.28 | 70.82 | 93.79 | 91.62 | 87.63 | 0.0 | 36.44 | 39.99 | 43.2 | 46.17 | 425 |
| | | PGD | 50.11 | 20.67 | 20.03 | 18.71 | 16.33 | 99.4 | 97.0 | 95.08 | 92.68 | 98.65 | 98.43 | 97.54 | 28.18 | 41.36 | 41.28 | 41.52 | 41.97 | 215 |
| | | TRADES | 50.38 | 23.35 | 23.12 | 18.65 | 17.72 | 99.45 | 96.79 | 94.69 | 92.35 | 98.73 | 98.27 | 97.58 | 17.71 | 35.93 | 36.43 | 37.18 | 37.4 | 175 |
| | | MART | 45.51 | 25.66 | 25.49 | 21.33 | 19.23 | 99.43 | 96.4 | 94.35 | 91.47 | 98.57 | 98.29 | 97.56 | 15.28 | 42.9 | 43.28 | 43.53 | 43.04 | 152 |
| | | ALP | 47.36 | 24.38 | 24.09 | 21.04 | 18.74 | 99.56 | 96.78 | 94.39 | 91.33 | 98.97 | 98.74 | 97.94 | 17.65 | 32.4 | 31.98 | 32.8 | 33.56 | 235 |
| | | CLP | 46.37 | 22.7 | 22.48 | 18.11 | 17.10 | 99.54 | 96.3 | 94.13 | 91.51 | 99.01 | 98.63 | 97.77 | 16.15 | 35.69 | 37.14 | 37.19 | 36.69 | 235 |
| | | KL-PGD | 56.66 | 19.23 | 18.65 | 15.68 | 14.08 | 99.38 | 97.1 | 95.6 | 93.41 | 98.6 | 98.27 | 97.56 | 13.98 | 35.41 | 35.92 | 35.9 | 36.15 | 182 |
| | ResNet-34 | ERM | 66.47 | 0.0 | 0.0 | 0.0 | 0.0 | 96.9 | 86.24 | 79.68 | 72.25 | 92.73 | 91.13 | 87.14 | 0.0 | 34.2 | 38.34 | 41.15 | 44.19 | 34 |
| | | Corr_Uniform | 66.84 | 0.02 | 0.01 | 0.0 | 0.0 | 97.9 | 89.31 | 83.6 | 76.87 | 94.79 | 93.26 | 89.74 | 0.0 | 33.64 | 36.31 | 39.04 | 41.67 | 34 |
| | | CVaR | 63.67 | 0.0 | 0.0 | 0.0 | 0.0 | 97.43 | 85.57 | 77.71 | 68.83 | 93.8 | 92.64 | 88.83 | 0.0 | 36.96 | 40.85 | 43.4 | 45.41 | 732 |
| | | PGD | 51.61 | 21.57 | 21.09 | 19.81 | 17.90 | 99.46 | 96.28 | 93.73 | 90.2 | 98.76 | 98.36 | 97.72 | 39.13 | 43.19 | 44.19 | 44.17 | 45.39 | 368 |
| | | TRADES | 53.19 | 24.59 | 24.23 | 20.78 | 19.37 | 99.36 | 96.42 | 94.53 | 91.72 | 98.51 | 98.17 | 97.41 | 32.22 | 40.78 | 41.27 | 41.29 | 41.28 | 293 |
| | | MART | 47.59 | 24.74 | 24.39 | 20.68 | 18.84 | 99.24 | 95.76 | 93.49 | 90.55 | 98.46 | 98.25 | 97.26 | 29.85 | 44.47 | 44.52 | 45.12 | 45.03 | 258 |
| | | ALP | 48.87 | 24.04 | 23.72 | 21.46 | 19.44 | 99.36 | 96.18 | 93.92 | 90.55 | 98.7 | 98.23 | 97.46 | 20.86 | 32.35 | 33.72 | 34.25 | 35.21 | 385 |
| | | CLP | 49.01 | 24.71 | 24.48 | 20.83 | 19.50 | 99.56 | 96.72 | 94.62 | 91.53 | 98.78 | 98.47 | 97.78 | 21.24 | 34.72 | 36.28 | 36.7 | 37.05 | 385 |
| | | KL-PGD | 57.66 | 20.62 | 19.96 | 18.05 | 16.07 | 99.48 | 97.18 | 95.6 | 93.41 | 98.7 | 98.31 | 97.46 | 24.73 | 36.16 | 36.91 | 36.74 | 37.34 | 312 |
| SVHN | ResNet-18 | ERM | 96.26 | 2.6 | 0.79 | 0.71 | 0.18 | 99.18 | 96.92 | 95.35 | 93.31 | 97.67 | 96.77 | 94.51 | 0.56 | 4.68 | 4.38 | 4.09 | 3.88 | 4 |
| | | Corr_Uniform | 96.22 | 7.72 | 2.49 | 2.37 | 0.56 | 99.54 | 98.44 | 97.66 | 96.52 | 98.49 | 98.1 | 97.18 | 1.14 | 4.98 | 5.5 | 5.51 | 5.45 | 4 |
| | | CVaR | 95.0 | 31.81 | 13.15 | 5.22 | 0.78 | 99.56 | 98.22 | 97.33 | 96.13 | 98.56 | 97.91 | 96.56 | 3.05 | 5.54 | 5.52 | 5.42 | 5.26 | 89 |
| | | PGD | 90.22 | 51.39 | 45.88 | 46.46 | 42.13 | 99.67 | 99.07 | 98.52 | 97.61 | 99.03 | 98.56 | 97.7 | 33.06 | 10.61 | 9.62 | 9.56 | 9.7 | 44 |
| | | TRADES | 90.3 | 58.29 | 52.79 | 50.64 | 43.97 | 99.75 | 99.13 | 98.46 | 97.3 | 99.15 | 98.79 | 98.17 | 23.38 | 6.97 | 6.92 | 6.82 | 6.83 | 37 |
| | | MART | 91.82 | 55.8 | 49.13 | 46.6 | 41.42 | 99.63 | 98.99 | 98.31 | 97.1 | 98.9 | 98.55 | 97.52 | 34.4 | 6.91 | 6.79 | 6.69 | 6.75 | 33 |
| | | ALP | 88.12 | 56.37 | 54.5 | 50.98 | 48.16 | 99.64 | 99.05 | 98.63 | 97.85 | 98.82 | 98.33 | 97.46 | 15.27 | 6.58 | 5.85 | 5.61 | 5.67 | 53 |
| | | CLP | 91.15 | 59.53 | 54.72 | 52.99 | 44.66 | 99.7 | 99.12 | 98.6 | 97.63 | 99.11 | 98.71 | 98.0 | 26.63 | 7.83 | 7.37 | 7.31 | 7.16 | 53 |
| | | KL-PGD | 91.93 | 53.09 | 50.71 | 51.19 | 46.35 | 99.73 | 99.14 | 98.6 | 97.63 | 99.11 | 98.79 | 98.0 | 22.73 | 7.26 | 7.07 | 7.13 | 7.38 | 40 |
| | WRN-28-10 | ERM | 96.16 | 8.65 | 5.78 | 5.48 | 0.13 | 99.21 | 96.45 | 96.39 | 92.3 | 98.63 | 98.15 | 97.1 | 1.94 | 4.18 | 3.81 | 3.67 | 3.55 | 22 |
| | | Corr_Uniform | 96.56 | 10.92 | 3.04 | 2.52 | 0.63 | 99.57 | 98.38 | 97.52 | 96.32 | 98.63 | 98.15 | 97.1 | 1.47 | 4.33 | 4.55 | 4.57 | 4.33 | 22 |
| | | CVaR | 94.35 | 32.26 | 16.97 | 4.06 | 0.41 | 99.35 | 97.83 | 96.9 | 95.7 | 97.98 | 97.28 | 95.61 | 2.74 | 5.06 | 4.51 | 4.06 | 3.88 | 460 |
| | | PGD | 91.91 | 56.07 | 49.84 | 51.1 | 45.41 | 99.73 | 99.3 | 98.95 | 98.34 | 99.1 | 98.68 | 97.9 | 31.87 | 8.23 | 7.95 | 7.92 | 8.06 | 241 |
| | | TRADES | 93.52 | 60.52 | 54.29 | 53.01 | 47.08 | 99.72 | 99.07 | 98.2 | 96.78 | 99.08 | 98.99 | 98.42 | 20.6 | 4.43 | 4.27 | 4.08 | 4.12 | 188 |
| | | MART | 94.15 | 58.37 | 51.06 | 48.43 | 42.16 | 99.64 | 99.1 | 98.57 | 97.58 | 99.0 | 98.72 | 97.9 | 30.11 | 4.27 | 4.12 | 3.84 | 3.87 | 166 |
| | | ALP | 90.42 | 59.85 | 55.69 | 54.41 | 51.33 | 99.6 | 99.02 | 98.67 | 98.12 | 99.04 | 98.43 | 97.51 | 11.89 | 3.78 | 3.59 | 3.56 | 3.55 | 286 |
| | | CLP | 92.0 | 64.27 | 59.76 | 58.78 | 50.25 | 99.74 | 99.32 | 98.99 | 98.39 | 99.25 | 98.92 | 98.09 | 18.09 | 4.19 | 4.24 | 4.08 | 4.13 | 286 |
| | | KL-PGD | 93.25 | 59.89 | 54.3 | 54.7 | 48.68 | 99.79 | 99.33 | 98.89 | 98.04 | 99.44 | 98.91 | 98.15 | 20.6 | 4.83 | 4.94 | 5.2 | 5.61 | 197 |
| | VGG19 | ERM | 95.78 | 9.46 | 3.24 | 3.33 | 0.27 | 99.43 | 97.41 | 96.64 | 94.1 | 98.24 | 97.64 | 96.23 | 1.49 | 4.68 | 4.96 | 4.79 | 4.51 | 2 |
| | | Corr_Uniform | 95.79 | 14.28 | 5.39 | 5.46 | 0.53 | 99.63 | 98.45 | 97.6 | 96.45 | 98.86 | 98.44 | 97.44 | 2.38 | 4.85 | 5.64 | 5.73 | 5.55 | 2 |
| | | CVaR | 92.92 | 44.55 | 31.31 | 2.81 | 0.48 | 99.29 | 97.28 | 95.95 | 94.33 | 97.83 | 97.12 | 95.23 | 2.63 | 4.79 | 5.0 | 4.83 | 4.52 | 61 |
| | | PGD | 89.76 | 52.53 | 45.75 | 46.38 | 38.51 | 99.64 | 99.04 | 98.63 | 98.1 | 98.73 | 98.45 | 97.6 | 24.97 | 9.11 | 8.74 | 8.7 | 8.6 | 29 |
| | | TRADES | 88.66 | 54.43 | 48.27 | 46.8 | 41.43 | 99.69 | 98.86 | 98.14 | 96.97 | 99.06 | 98.72 | 97.9 | 30.3 | 10.3 | 9.36 | 9.16 | 9.26 | 24 |
| | | MART | 90.09 | 49.89 | 42.42 | 38.72 | 33.54 | 99.66 | 98.97 | 98.53 | 97.92 | 98.78 | 98.43 | 97.6 | 29.35 | 6.87 | 7.05 | 7.13 | 7.21 | 22 |
| | | ALP | 88.8 | 56.08 | 53.66 | 48.31 | 44.84 | 99.6 | 99.0 | 98.65 | 98.18 | 98.78 | 98.56 | 97.8 | 11.33 | 2.09 | 2.27 | 2.36 | 2.65 | 35 |
| | | CLP | 87.43 | 56.1 | 51.46 | 48.44 | 44.22 | 99.65 | 99.08 | 98.75 | 98.22 | 98.82 | 98.49 | 97.55 | 21.85 | 7.64 | 6.8 | 6.82 | 6.74 | 35 |
| | | KL-PGD | 92.03 | 55.36 | 49.07 | 48.78 | 43.20 | 99.74 | 99.17 | 98.69 | 97.9 | 99.06 | 98.73 | 98.03 | 23.24 | 7.47 | 7.36 | 7.32 | 7.28 | 27 |

Table 11: Performance comparison of AT and RT training methods across MNIST with SimpleCNN, CINIC-10 with ResNet-18 and WRN-28-10, and ImageNet-50 with ResNet-18.

| Data | Model | Method | Acc. % | AR % | | | | $PR_D^{Uniform}(\gamma)$ % | | | | $ProbAcc(\rho, \gamma=0.03)$ % | | | $GE_{AR}$ % | $GE_{PR_D^{Uniform}(\gamma)}$ % | | | | Time |
|---|---|---|---|---|---|---|---|---|---|---|---|---|---|---|---|---|---|---|---|---|
| | | | | $PGD^{10}$ | $PGD^{20}$ | $CW^{20}$ | AA | 0.03 | 0.08 | 0.1 | 0.12 | 0.1 | 0.05 | 0.01 | $PGD^{20}$ | 0.03 | 0.08 | 0.1 | 0.12 | s/ep. |
| CINIC-10 | ResNet-18 | ERM | **86.92** | 0.0 | 0.0 | 0.0 | 0.0 | 66.83 | 9.6 | 3.65 | 1.31 | 53.78 | 49.97 | 41.6 | 0.0 | 2.76 | 0.1 | 0.0 | 0.0 | 4 |
| | | CVaR | 78.73 | 24.97 | 20.11 | 0.01 | 0.0 | 96.05 | 46.53 | 26.3 | 14.04 | 92.29 | 91.02 | 87.64 | 1.9 | 5.31 | 2.69 | 1.18 | 0.15 | 60 |
| | | Corr_Uniform | 85.45 | 0.01 | 0.0 | 0.0 | 0.0 | 97.9 | 58.77 | 36.95 | 21.07 | 95.64 | 94.76 | 92.26 | 0.0 | 6.79 | 5.85 | 3.24 | 1.41 | 4 |
| | | Corr_Gaussian | 84.0 | 0.16 | 0.02 | 0.02 | 0.0 | 99.11 | 85.76 | 70.12 | 52.37 | 97.64 | 97.0 | 95.27 | 0.0 | 5.1 | 6.06 | 8.36 | 3.67 | 4 |
| | | Corr_Laplace | 84.0 | 0.21 | 0.04 | 0.04 | 0.0 | 99.08 | 84.09 | 64.85 | 44.41 | 97.61 | 96.91 | 95.33 | 0.0 | 4.35 | 9.94 | 8.02 | 3.38 | 4 |
| | | PGD | 68.3 | 39.83 | 39.02 | 37.56 | 35.06 | 99.47 | 96.51 | 93.62 | 89.07 | 98.91 | 98.68 | 98.14 | 16.04 | 10.14 | 11.67 | 12.87 | 14.21 | 30 |
| | | TRADES | 69.22 | 39.42 | 38.78 | 35.41 | 34.34 | 99.32 | 96.3 | 94.0 | 90.75 | 98.7 | 98.51 | 98.04 | 9.23 | 7.44 | 7.72 | 7.84 | 8.19 | 25 |
| | | MART | 66.91 | 42.67 | 41.97 | 37.31 | 35.18 | 99.37 | 96.65 | 94.58 | 91.06 | 98.79 | 98.55 | 98.17 | 10.62 | 5.82 | 7.1 | 8.15 | 8.83 | 22 |
| | | ALP | 67.99 | 40.52 | 39.73 | 38.05 | 35.74 | 99.41 | 96.7 | 94.4 | 90.18 | 98.81 | 98.57 | 98.09 | 13.22 | 8.05 | 9.4 | 9.83 | 11.13 | 36 |
| | | CLP | 74.61 | 34.7 | 33.36 | 32.69 | 30.65 | 99.49 | 96.57 | 93.64 | 88.9 | 98.78 | 98.61 | 98.07 | 11.81 | 10.28 | 11.08 | 10.78 | 11.76 | 36 |
| | | KL-PGD | 75.33 | 34.83 | 33.98 | 31.0 | 29.50 | 99.57 | 96.82 | 94.08 | 90.05 | 99.0 | 98.78 | 98.34 | 8.55 | 8.45 | 8.8 | 8.55 | 8.96 | 27 |
| | WRN-28-10 | ERM | 88.01 | 0.0 | 0.0 | 0.0 | 0.0 | 82.17 | 25.55 | 13.82 | 7.51 | 71.65 | 68.0 | 60.95 | 0.0 | 7.44 | 0.0 | 0.0 | 0.0 | 15 |
| | | CVaR | 74.27 | 25.53 | 19.57 | 1.56 | 0.07 | 98.81 | 74.84 | 54.73 | 35.22 | 97.53 | 97.09 | 95.79 | 1.61 | 2.2 | 6.13 | 2.62 | 0.54 | 312 |
| | | Corr_Uniform | 86.63 | 0.04 | 0.0 | 0.01 | 0.0 | 98.33 | 74.85 | 58.37 | 42.38 | 96.48 | 95.57 | 93.35 | 0.01 | 4.24 | 9.7 | 6.14 | 3.22 | 15 |
| | | Corr_Gaussian | 85.59 | 0.21 | 0.01 | 0.03 | 0.0 | 99.23 | 89.6 | 78.56 | 64.15 | 98.13 | 97.63 | 96.35 | 0.01 | 3.21 | 10.11 | 10.62 | 7.41 | 15 |
| | | Corr_Laplace | 85.52 | 0.15 | 0.01 | 0.04 | 0.0 | 99.1 | 88.8 | 77.72 | 63.33 | 97.8 | 97.23 | 95.94 | 0.0 | 3.01 | 10.76 | 9.87 | 6.92 | 15 |
| | | PGD | 72.26 | 37.77 | 36.53 | 36.58 | 33.82 | 99.52 | 97.3 | 95.26 | 92.08 | 98.71 | 98.37 | 97.65 | 29.3 | 7.7 | 9.14 | 10.65 | 12.77 | 165 |
| | | TRADES | 73.23 | 40.08 | 39.26 | 37.13 | 35.70 | 99.41 | 96.48 | 94.48 | 91.32 | 98.73 | 98.49 | 98.03 | 24.12 | 10.75 | 11.7 | 12.58 | 13.41 | 130 |
| | | MART | 70.75 | 43.56 | 42.56 | 38.37 | 36.35 | 99.34 | 96.56 | 94.13 | 90.62 | 98.72 | 98.43 | 97.96 | 17.84 | 8.41 | 9.45 | 10.31 | 11.1 | 114 |
| | | ALP | 71.97 | 38.12 | 36.9 | 36.75 | 34.07 | 99.52 | 97.39 | 95.39 | 92.25 | 98.85 | 98.6 | 98.05 | 28.46 | 8.52 | 9.74 | 10.94 | 12.82 | 194 |
| | | CLP | 76.11 | 35.52 | 33.84 | 33.99 | 32.05 | 99.6 | 97.39 | 95.35 | 92.21 | 98.91 | 98.73 | 98.03 | 26.06 | 8.32 | 9.88 | 11.25 | 12.63 | 194 |
| | | KL-PGD | 77.44 | 36.28 | 35.5 | 33.39 | 32.01 | 99.51 | 96.84 | 94.49 | 90.95 | 98.76 | 98.54 | 98.02 | 16.9 | 8.19 | 10.02 | 10.49 | 10.41 | 135 |
| ImageNet-50 | ResNet-18 | ERM | 79.94 | 0.0 | 0.0 | 0.0 | 0 | 98.92 | 93.63 | 91.01 | 87.46 | 97.63 | 97.39 | 96.23 | 0.0 | 12.87 | 14.49 | 14.39 | 13.5 | 97 |
| | | CVaR | 77.07 | 0.0 | 0.0 | 0.0 | 0 | 98.88 | 93.68 | 90.87 | 87.0 | 97.92 | 96.98 | 95.09 | 0.0 | 14.37 | 14.53 | 14.68 | 15.05 | 300 |
| | | Corr_Uniform | 79.49 | 0.0 | 0.0 | 0.0 | 0 | 99.0 | 95.23 | 92.15 | 88.42 | 97.69 | 97.26 | 96.66 | 0.0 | 12.34 | 12.88 | 12.99 | 12.3 | 97 |
| | | Corr_Gaussian | 79.1 | 0.0 | 0.0 | 0.0 | 0 | 99.24 | 96.43 | 94.21 | 91.03 | 98.11 | 97.5 | 96.76 | 0.0 | 13.03 | 13.03 | 13.52 | 14.42 | 97 |
| | | Corr_Laplace | 80.03 | 0.0 | 0.0 | 0.0 | 0 | 98.93 | 95.84 | 93.44 | 90.33 | 98.01 | 97.53 | 96.56 | 0.0 | 12.66 | 13.39 | 14.4 | 14.19 | 97 |
| | | PGD | 63.32 | 36.0 | 35.32 | 33.39 | 31.76 | 99.89 | 97.15 | 95.36 | 92.8 | 99.78 | 99.7 | 99.4 | 5.43 | 20.38 | 20.39 | 21.36 | 21.5 | 184 |
| | | TRADES | 65.95 | 35.2 | 34.59 | 30.06 | 28.9 | 99.15 | 96.09 | 93.1 | 89.83 | 99.06 | 98.92 | 98.77 | 2.64 | 17.58 | 18.02 | 18.7 | 19.39 | 167 |
| | | MART | 60.48 | 38.44 | 37.85 | 32.58 | 30.12 | 99.25 | 95.51 | 92.47 | 89.11 | 98.99 | 98.91 | 98.52 | 3.38 | 16.87 | 17.14 | 17.86 | 19.81 | 155 |
| | | ALP | 61.51 | 35.21 | 34.69 | 33.23 | 31.03 | 99.67 | 97.18 | 95.35 | 92.79 | 99.31 | 99.24 | 98.7 | 3.16 | 19.75 | 20.73 | 20.81 | 20.81 | 214 |
| | | CLP | 66.14 | 29.51 | 28.2 | 26.44 | 25.39 | 99.65 | 97.54 | 95.57 | 93.46 | 99.5 | 99.43 | 99.21 | 2.68 | 21.23 | 20.73 | 21.16 | 22.4 | 214 |
| | | KL-PGD | 69.86 | 32.0 | 31.01 | 28.35 | 27.4 | 99.59 | 97.02 | 95.5 | 92.92 | 99.45 | 99.24 | 98.83 | 2.33 | 16.1 | 15.89 | 16.0 | 16.55 | 180 |
| MNIST | SimpleCNN | ERM | 99.35 | 5.16 | 3.72 | 0.0 | 0.0 | 99.2 | 98.05 | 94.21 | 83.45 | 98.03 | 97.43 | 95.3 | 0.0 | 1.03 | 1.18 | 1.01 | 0.41 | 1 |
| | | Corr_Uniform | 99.38 | 3.41 | 1.75 | 0.01 | 0.0 | 99.8 | 99.66 | 99.25 | 97.75 | 99.37 | 99.04 | 98.45 | 0.22 | 0.68 | 0.78 | 0.95 | 1.06 | 1 |
| | | CVaR | 99.18 | 0.3 | 0.01 | 0.19 | 0.0 | 99.71 | 99.56 | 99.25 | 98.53 | 99.86 | 99.77 | 99.15 | 0.02 | 0.83 | 0.9 | 0.92 | 0.93 | 6 |
| | | PGD | 99.15 | 95.01 | 94.67 | 94.8 | 91.17 | 99.88 | 99.83 | 99.56 | 97.78 | 99.62 | 99.47 | 99.19 | 2.48 | 0.63 | 0.62 | 0.72 | 0.99 | 3 |
| | | TRADES | 98.51 | 94.22 | 93.73 | 93.93 | 90.88 | 99.83 | 99.74 | 99.25 | 96.38 | 99.48 | 99.3 | 98.82 | 1.08 | 0.03 | 0.03 | 0.08 | 0.2 | 3 |
| | | MART | 99.01 | 95.21 | 94.88 | 94.98 | 91.77 | 99.88 | 99.81 | 99.3 | 93.0 | 99.65 | 99.52 | 99.2 | 1.75 | 0.34 | 0.33 | 0.49 | 0.79 | 3 |
| | | ALP | 99.04 | 95.08 | 94.8 | 94.7 | 91.45 | 99.86 | 99.83 | 99.57 | 98.4 | 99.61 | 99.42 | 99.1 | 1.43 | 0.37 | 0.37 | 0.42 | 0.62 | 3 |
| | | CLP | 98.52 | 93.84 | 93.35 | 92.91 | 89.41 | 99.83 | 99.75 | 99.42 | 98.29 | 99.46 | 99.2 | 98.74 | 1.51 | 0.16 | 0.14 | 0.16 | 0.22 | 3 |
| | | KL-PGD | 98.94 | 94.24 | 93.97 | 94.07 | 91.33 | 99.88 | 99.8 | 99.37 | 95.78 | 99.66 | 99.47 | 99.1 | 1.21 | 0.3 | 0.32 | 0.42 | 0.54 | 3 |

Table 12: Evaluation of an efficient AT method Chen & Lee (2024) using ResNet-18 on CIFAR-10 & CIFAR-100. The method employs a simple yet effective data filtering mechanism to enhance training efficiency and robustness. Integrated into TRADES with $m = 0$–$8$ and $k = 10$ (10-step attack), it reduces computational cost, lowers AR/PR generalization error, and improves PR performance while maintaining comparable clean accuracy and AR. This experiment also illustrates the extensibility of PRBench, which is designed to incorporate and compare future efficient AT methods.

| Data | Type | Method | Acc. % | AR % | | | $PR_D^{Uniform}(\gamma)$ % | | | | $ProbAcc(\rho, \gamma=0.03)$ % | | | $GE_{AR}$ % | $GE_{PR_D^{Uniform}(\gamma)}$ % | | | | Time |
|---|---|---|---|---|---|---|---|---|---|---|---|---|---|---|---|---|---|---|---|
| | | | | $PGD^{10}$ | $PGD^{20}$ | $CW^{20}$ | 0.03 | 0.08 | 0.1 | 0.12 | 0.1 | 0.05 | 0.01 | $PGD^{20}$ | 0.03 | 0.08 | 0.1 | 0.12 | s/ep. |
| CIFAR-10 | Std. | ERM | 94.85 | 0.01 | 0.0 | 0.0 | 97.64 | 76.19 | 61.65 | 47.07 | 95.04 | 93.48 | 89.82 | 0.0 | 6.24 | 3.98 | 2.94 | 2.54 | 3 |
| | RT | Corr_Uniform | 94.17 | 0.28 | 0.05 | 0.02 | 99.12 | 90.92 | 82.32 | 70.48 | 97.78 | 97.02 | 94.9 | 0.04 | 4.83 | 6.24 | 4.79 | 2.31 | 3 |
| | AT | TRADES (k=10) | 83.34 | 53.6 | 52.71 | 50.63 | 99.57 | 97.61 | 96.28 | 94.56 | 99.05 | 98.81 | 98.11 | 16.5 | 9.62 | 9.28 | 9.24 | 9.52 | 25 |
| | | TRADES+Chen&Lee (m=1) | 83.6 | 52.82 | 52.01 | 50.5 | 99.59 | 97.69 | 96.48 | 94.97 | 99.09 | 98.76 | 98.08 | 10.37 | **4.57** | **4.87** | **4.95** | **5.19** | 16 |
| | | TRADES+Chen&Lee (m=2) | 83.71 | 53.0 | 52.07 | 50.35 | 99.5 | 97.54 | 96.32 | 94.6 | 98.72 | 98.47 | 97.83 | 10.53 | 6.14 | 6.21 | 6.34 | 6.14 | 16 |
| | | TRADES+Chen&Lee (m=3) | 83.66 | 53.19 | 52.51 | 50.65 | 99.59 | 97.76 | 96.47 | 94.76 | 98.87 | 98.71 | 98.06 | **10.08** | 5.78 | 5.76 | 5.94 | 6.1 | 17 |
| | | TRADES+Chen&Lee (m=4) | 83.49 | 52.59 | 51.8 | 50.61 | 99.53 | 97.62 | 96.28 | 94.49 | 98.85 | 98.44 | 97.75 | 10.81 | 5.92 | 6.39 | 6.45 | 6.58 | 18 |
| | | TRADES+Chen&Lee (m=5) | **83.56** | 52.77 | 52.0 | 50.43 | 99.58 | 97.8 | 96.58 | 94.94 | 98.9 | 98.68 | 98.03 | 10.62 | 5.02 | 5.64 | 5.37 | 5.84 | 18 |
| | | TRADES+Chen&Lee (m=6) | 83.26 | 52.82 | 52.01 | 50.21 | 99.6 | **97.98** | **96.84** | **95.22** | 99.04 | **98.84** | **98.25** | 10.69 | 5.81 | 5.83 | 5.47 | 5.25 | 19 |
| | | TRADES+Chen&Lee (m=7) | 83.54 | 52.91 | 52.12 | 50.42 | 99.6 | 97.79 | 96.6 | 94.97 | 98.99 | 98.7 | 98.18 | 10.55 | 5.69 | 5.96 | 6.13 | 6.4 | 19 |
| | | TRADES+Chen&Lee (m=8) | 83.22 | 53.12 | 52.3 | 50.62 | 99.65 | 98.01 | 96.77 | 95.05 | **99.05** | 98.79 | 98.22 | 10.36 | 5.93 | 6.21 | 6.25 | 6.55 | 19 |
| CIFAR-100 | Std. | ERM | 76.03 | 0.0 | 0.0 | 0.0 | 85.02 | 45.12 | 31.7 | 22.22 | 76.65 | 73.57 | 66.44 | 0.0 | 23.17 | 6.54 | 3.86 | 1.77 | 3 |
| | RT | Corr_Uniform | 74.93 | 0.07 | 0.01 | 0.0 | 97.09 | 71.69 | 55.41 | 40.53 | 93.14 | 91.05 | 86.25 | 0.0 | 24.84 | 19.18 | 12.12 | 6.78 | 3 |
| | AT | TRADES (k=10) | **59.65** | **30.63** | **30.15** | 26.47 | 99.08 | 95.02 | 92.09 | 87.8 | 97.99 | 97.36 | 96.16 | 26.6 | 31.55 | 31.73 | 30.76 | 29.41 | 25 |
| | | TRADES+Chen&Lee (m=1) | 58.57 | 30.19 | 29.83 | 26.49 | **99.22** | **95.43** | **92.29** | 88.09 | **98.2** | **97.72** | 96.63 | 13.46 | 18.33 | 17.52 | 16.85 | **15.01** | 16 |
| | | TRADES+Chen&Lee (m=2) | 59.24 | 30.26 | 29.94 | 26.64 | 99.15 | 95.14 | 92.11 | 87.93 | 98.09 | 97.61 | **96.7** | **12.96** | **16.77** | **16.83** | **16.68** | 15.08 | 16 |
| | | TRADES+Chen&Lee (m=3) | 58.8 | 30.26 | 30.01 | 26.63 | 99.19 | 95.12 | 92.35 | **88.26** | 97.96 | 97.53 | 96.42 | 13.07 | 18.05 | 17.49 | 16.96 | 15.66 | 17 |
| | | TRADES+Chen&Lee (m=4) | 58.78 | 30.09 | 29.64 | 26.46 | 99.11 | 95.08 | 91.8 | 87.59 | 97.97 | 97.39 | 96.24 | 13.5 | 18.01 | 16.67 | 16.72 | 16.48 | 18 |
| | | TRADES+Chen&Lee (m=5) | 58.67 | 30.39 | 29.98 | **26.65** | 99.09 | 95.03 | 91.92 | 87.61 | 97.87 | 97.34 | 96.27 | 13.11 | 19.43 | 19.22 | 18.23 | 15.86 | 18 |
| | | TRADES+Chen&Lee (m=6) | 59.28 | 30.0 | 29.77 | 26.48 | 99.21 | 95.05 | 91.95 | 87.95 | 98.02 | 97.49 | 96.51 | 13.36 | 17.75 | 17.84 | 16.98 | 15.93 | 19 |
| | | TRADES+Chen&Lee (m=7) | 58.78 | 29.86 | 29.52 | 26.43 | 98.96 | 94.91 | 92.05 | 87.93 | 97.86 | 97.31 | 96.24 | 13.59 | 17.62 | 17.38 | 16.34 | 15.7 | 19 |
| | | TRADES+Chen&Lee (m=8) | 59.16 | 30.33 | 29.85 | 26.61 | 99.06 | 94.71 | 91.37 | 87.23 | 97.83 | 97.22 | 96.02 | 13.17 | 18.14 | 18.26 | 17.75 | 16.09 | 19 |

where $Pr(\cdot \mid \boldsymbol{x})$ is a uniform perturbation distribution, and $\rho \in (0, 1)$ determines the focus on the tail of the loss distribution. The threshold $\alpha$ is dynamically optimized during training to ignore perturbations with loss values below it, thereby encouraging the model to focus on AEs that incur higher but not necessarily maximal loss, aligning with the CVaR principle of prioritizing the worst-performing fraction of the distribution.

Building upon CVaR, Zhang et al. (2024a) propose Probabilistic Risk-averse Robust Learning with Stochastic Search (PRASS), which trains a risk-averse model by mitigating the *Entropic Value-at-Risk* (EVaR) over the captured worst-case perturbation distribution. Unlike CVaR, which focuses solely on the tail samples, EVaR further capitalizes on the entire distribution of samples. Specifically, EVaR

with risk level $\rho \in (0, 1]$, denoted as $\text{EVaR}_\rho(\mathcal{L}(\boldsymbol{\delta}))$, is defined as the infimum over $\alpha > 0$ of the Chernoff bound for $\mathcal{L}(\boldsymbol{\delta})$ with respect to the random variable $\boldsymbol{\delta}$. They then proposed a risk-averse robust learning paradigm to optimize the upper bound, and formalized the training objective as:

$$\min_{\boldsymbol{\theta}} \; \mathbb{E}_{(\boldsymbol{x},y)\sim\mathcal{D}} \left[ \text{EVaR}_{1-\rho} \left( \mathcal{L}(\boldsymbol{x} + \boldsymbol{\delta}, y; \boldsymbol{\theta}); \boldsymbol{\delta} \sim Pr(\cdot \mid \boldsymbol{x}) \right) \right];$$

$$\text{EVaR}_\rho(\mathcal{L}; Pr) = \inf_{\alpha > 0} \left[ \frac{1}{\alpha} \log \left( \frac{\mathbb{E}_{\boldsymbol{\delta}\sim Pr(\cdot|\boldsymbol{x})} \left[ e^{\alpha \mathcal{L}(\boldsymbol{x}+\boldsymbol{\delta})} \right]}{\rho} \right) \right], \tag{20}$$

thus the inner minimization over $\alpha$ is used to compute EVaR, while the outer minimization updates the model parameters.

The latest PR-targeted work Zhang et al. (2025), though aimed at PR, is not based on the perturbation risk function in Def. 3. Instead, it is closer to the traditional AT formulation in Eq. 4, introducing a new min–max objective designed to identify AEs that lie in the largest all-AE region $k$ (i.e., $PR(\boldsymbol{x} + \boldsymbol{\delta}, k) = 0$). They achieve this by designing a numerical algorithm that first uses PGD to generate a diverse set of AEs at different local optima, i.e., $\mathcal{S} = \{\boldsymbol{x}_1', \boldsymbol{x}_2', \ldots, \boldsymbol{x}_m'\}$. For each $\boldsymbol{x}_i'$, gradient descent is then performed toward the nearest decision boundary, and the AE with the largest distance to its nearest boundary is selected, corresponding to the largest all-AE region. Formally, for a model $f_{\boldsymbol{\theta}}$ trained on dataset $\mathcal{D}$, the objective of AT-PR is:

$$\min_{\boldsymbol{\theta}} \mathbb{E}_{(\boldsymbol{x},y)\sim\mathcal{D}} \left[ \max_{\|\boldsymbol{\delta}\|\leq\gamma, \; PR(\boldsymbol{x}+\boldsymbol{\delta},k)=0} k \right], \tag{21}$$

where $k$ and $\boldsymbol{\delta}$ are variables in the inner maximization, and $PR(\cdot, \cdot)$ is defined in Eq. 5. The maximization seeks an optimal $\boldsymbol{\delta}^*$ producing an AE $\boldsymbol{x}' = \boldsymbol{x} + \boldsymbol{\delta}^*$, and maximizes the radius $k$ of a smaller norm-ball around $\boldsymbol{x}'$ where all inputs are AEs (i.e., $PR(\cdot, \cdot)$ within the region is 0).

# F EVALUATION METRICS

## F.1 AR EVALUATION METRICS

AR is a critical aspect of model robustness and has been extensively studied in previous benchmark works. To measure AR, adversarial attacks (Eq. 3) are typically adopted to generate AEs $x'$ for each input $x$ in the test dataset. In PRBench, we consider four white-box attacks: two variants of Projected Gradient Descent attack with 10 and 20 iterations ($PGD^{10}$ and $PGD^{20}$), the C&W attack Carlini & Wagner (2017), and Auto-attack Croce & Hein (2020). Formally, AR is computed as the classification accuracy of the target model on adversarially perturbed inputs across the test dataset:

$$AR = \frac{1}{M} \sum_{i=1}^{M} I_{\{f_{\boldsymbol{\theta}}(\boldsymbol{x}_i + \boldsymbol{\delta}_i) = y_i\}}(\boldsymbol{x}_i), \qquad (22)$$

where $\|\boldsymbol{\delta}_i\| \leq \gamma$ denotes the perturbation generated by the attack, $\boldsymbol{x}'_i = \boldsymbol{x}_i + \boldsymbol{\delta}_i$ is the corresponding AE, $y_i$ is the ground-truth label, and $M$ is the total number of test samples in the dataset $\mathcal{D}$. The indicator function $I_{\mathcal{S}}(\boldsymbol{x})$ returns 1 if the model prediction matches the true label, and 0 otherwise.

## F.2 PR EVALUATION METRICS

PR was first introduced by Webb et al. (2019), which proposed a formal framework to quantify PR under random perturbations (Def. 2). Computing the exact PR requires taking an expectation over the perturbation distribution, which is typically intractable. Therefore, a Monte Carlo approximation is adopted by uniformly sampling a large number of perturbations within a norm ball of radius $\gamma$. The target model then predicts the labels of these perturbed inputs to determine whether they are AEs $x'$. PR is thus defined as the proportion of non-adversarial samples among all sampled perturbations. Formally, given a test dataset $\mathcal{D}$ and a perturbation budget $\gamma$, PR is estimated as:

$$PR_{\mathcal{D}}(\gamma) = \frac{1}{M} \sum_{i=1}^{M} \frac{1}{N} \sum_{j=1}^{N} I_{\{f_{\boldsymbol{\theta}}(\boldsymbol{x}'_{i,j}) = y_i\}}(\boldsymbol{x}_{i,j}), \quad \boldsymbol{x}'_{i,j} \sim Pr(\cdot \mid \boldsymbol{x}_i), \; \|\boldsymbol{x}'_{i,j} - \boldsymbol{x}_i\| \leq \gamma, \qquad (23)$$

where $\boldsymbol{x}'_{i,j}$ denotes the $j$-th perturbed sample of $\boldsymbol{x}_i$, drawn from the conditional perturbation distribution $Pr(\cdot \mid \boldsymbol{x}_i)$, for $j \in \{1, \ldots, N\}$. Here, $M$ is the number of test samples originally classified correctly and $N$ is the number of perturbations sampled for each input.

Later, Robey et al. (2022) proposed *quantile accuracy ProbAcc($\rho$)* to evaluate PR for individual inputs using the same perturbation strategy (uniform sampling around each input). Unlike $PR_{\mathcal{D}}(\gamma)$, which estimates the mean PR over all correctly classified clean test samples, this metric focuses on a given robustness tolerance threshold $\rho$. Specifically, *ProbAcc($\rho$)* measures the proportion of inputs whose $PR(\boldsymbol{x}, \gamma)$ exceeds the threshold $1 - \rho$, formally defined as follows:

$$ProbAcc(\rho) = \frac{1}{M} \sum_{i=1}^{M} I_{\{PR(\boldsymbol{x}_i, \gamma) \geq 1 - \rho\}}(\boldsymbol{x}_i), \qquad (24)$$

In PRBench, we assess the PR of different training methods using both $PR_{\mathcal{D}}(\gamma)$ and *ProbAcc($\rho$)*. In our implementation, *ProbAcc($\rho$)* is computed only on clean test samples–that is, $M$ is the number of inputs that are originally classified correctly by the model (i.e., $f_{\boldsymbol{\theta}}(\boldsymbol{x}_i) = y_i$), reflecting the true robustness of individual examples.

## F.3 GENERALIZATION ERROR

To investigate the deeper relationship of AR and PR, and compare the effectiveness of different training methods, we also evaluate the generalization error of each training method with respect to AR and PR. Specifically, we compute AR and PR on both the training and test datasets. The difference between these values reflects the generalisability of a model under each robustness metric. Taking PR as an example, the generalization error (*GE*) of PR (e.g., $PR_{\mathcal{D}}(\gamma)$) is defined as:

$$GE_{PR_{\mathcal{D}}(\gamma)} = PR_{\mathcal{D}_{\text{train}}}(\gamma) - PR_{\mathcal{D}_{\text{test}}}(\gamma). \qquad (25)$$

# G TECHNICAL APPENDICES AND THEORETICAL ANALYSIS

## G.1 LIPSCHITZ AND SMOOTHNESS PROPERTIES OF THE SOFTMAX FUNCTION

**Lemma 1** *Given $\boldsymbol{z} \in \mathbb{R}^m$, let the softmax function be represented as*

$$
\mathbf{p}(\boldsymbol{z}) = \begin{pmatrix} \frac{e^{z_1}}{\sum e^{z_j}} \\ \frac{e^{z_2}}{\sum e^{z_j}} \\ \vdots \\ \frac{e^{z_m}}{\sum e^{z_j}} \end{pmatrix}
\tag{26}
$$

*Let $\|\cdot\|_2$ denote the $L_2$ norm for vectors and the induced norm for matrices. The Lipschitz condition for the softmax function is*

$$
\|\mathbf{p}(\boldsymbol{z}_2) - \mathbf{p}(\boldsymbol{z}_1)\|_2 \leq \|\boldsymbol{z}_2 - \boldsymbol{z}_1\|_2
\tag{27}
$$

*and we also have*

$$
\|\nabla \mathbf{p}(\boldsymbol{z}_2) - \nabla \mathbf{p}(\boldsymbol{z}_1)\| \leq 3\|\boldsymbol{z}_2 - \boldsymbol{z}_1\|_2
\tag{28}
$$

**Proof 1** *According to the Rayleigh quotient, for the $L_2$-norm we have*

$$
\|\operatorname{diag}(\mathbf{p}) - \mathbf{p}\mathbf{p}^T\|_2 = \sup_{\|\boldsymbol{v}\|_2=1} \boldsymbol{v}^T \operatorname{diag}(\mathbf{p})\boldsymbol{v} - (\boldsymbol{v}^T\mathbf{p})^2
\tag{29}
$$

$$
\leq \sup_{\|\boldsymbol{v}\|_2=1} \boldsymbol{v}^T \operatorname{diag}(\mathbf{p})\boldsymbol{v}
\tag{30}
$$

$$
= \sup_{\|\boldsymbol{v}\|_2=1} \|\operatorname{diag}(\mathbf{p})\boldsymbol{v}\|_2
\tag{31}
$$

$$
\leq \sup_{\|\boldsymbol{v}\|_2=1} \sqrt{\sum_i (p_i v_i)^2}
\tag{32}
$$

$$
\leq \sup_{\|\boldsymbol{v}\|_2=1} \sum_i |p_i v_i| \leq 1
\tag{33}
$$

*Since for all $\boldsymbol{z}_1, \boldsymbol{z}_2$ we have*

$$
\|\operatorname{diag}(\mathbf{p}(\boldsymbol{z}_2) - \mathbf{p}(\boldsymbol{z}_1))\|_2 \leq \|\mathbf{p}(\boldsymbol{z}_2) - \mathbf{p}(\boldsymbol{z}_1)\|_2
\tag{34}
$$

*and*

$$
\|\mathbf{p}(\boldsymbol{z}_2)\mathbf{p}(\boldsymbol{z}_2)^T - \mathbf{p}(\boldsymbol{z}_1)\mathbf{p}(\boldsymbol{z}_1)^T\| \leq \|\mathbf{p}(\boldsymbol{z}_2)\|_2\|\mathbf{p}(\boldsymbol{z}_2) - \mathbf{p}(\boldsymbol{z}_1)\|_2 + \|\mathbf{p}(\boldsymbol{z}_2) - \mathbf{p}(\boldsymbol{z}_1)\|_2\|\mathbf{p}(\boldsymbol{z}_1)\|_2
\tag{35}
$$

$$
\leq 2\|\mathbf{p}(\boldsymbol{z}_2) - \mathbf{p}(\boldsymbol{z}_1)\|_2,
\tag{36}
$$

*we have*

$$
\|\nabla \mathbf{p}(\boldsymbol{z}_2) - \nabla \mathbf{p}(\boldsymbol{z}_1)\|_2 \leq 3\|\boldsymbol{z}_2 - \boldsymbol{z}_1\|_2
\tag{37}
$$

## G.2 LIPSCHITZ AND SMOOTHNESS CONDITIONS FOR LOSS COMPOSITION

**Lemma 2 (Gradient of the Cross-Entropy Loss)** *Let $\mathcal{L}_{CE}(\mathbf{p}(\mathcal{M}), y)$ be the cross-entropy loss, where $\mathbf{p}$ denotes the softmax function. Then, the gradient of the cross-entropy loss — that is, the cross-entropy composed with the softmax function— is given by*

$$
\nabla_f \mathcal{L}_{CE} = \mathbf{p} - \boldsymbol{l}_y
\tag{38}
$$

.

**Proof 2** *Let $\mathbf{p}$ denote the softmax function, and let $\mathcal{L}$ be a function defined over its output. Then, we have*

$$\nabla_f \mathcal{L}^T = \nabla_{\mathbf{p}} \mathcal{L}^T \nabla_f \mathbf{p}(f) \tag{39}$$

$$= \nabla_{\mathbf{p}} \mathcal{L}^T \begin{pmatrix} \frac{e^{z_1}}{\sum e^{z_j}} - \frac{e^{z_1}e^{z_1}}{(\sum e^{z_j})^2} & -\frac{e^{z_1}e^{z_2}}{(\sum e^{z_j})^2} & \cdots & -\frac{e^{z_1}e^{z_\kappa}}{(\sum e^{z_j})^2} \\ -\frac{e^{z_2}e^{z_1}}{(\sum e^{z_j})^2} & \frac{e^{z_2}}{\sum e^{z_j}} - \frac{e^{z_2}e^{z_2}}{(\sum e^{z_j})^2} & \cdots & -\frac{e^{z_2}e^{z_\kappa}}{(\sum e^{z_j})^2} \\ \vdots & \vdots & \ddots & \vdots \\ -\frac{e^{z_\kappa}e^{z_1}}{(\sum e^{z_j})^2} & -\frac{e^{z_\kappa}e^{z_2}}{(\sum e^{z_j})^2} & \cdots & \frac{e^{z_\kappa}}{\sum e^{z_j}} - \frac{e^{z_\kappa}e^{z_\kappa}}{(\sum e^{z_j})^2} \end{pmatrix} \tag{40}$$

$$= \nabla_{\mathbf{p}} \mathcal{L}^T (\operatorname{diag}(\mathbf{p}) - \mathbf{p}\mathbf{p}^T) \tag{41}$$

*where $\nabla \mathcal{L}^T$ denotes the gradient expressed as a row vector. If $\mathcal{L}$ is the cross-entropy, we have*

$$\nabla_{\mathbf{p}} \mathcal{L}_{CE} = -\frac{\partial \sum_i^\kappa l_i \log p_i}{\partial \mathbf{p}} = -\begin{pmatrix} \frac{l_1}{p_1} \\ \frac{l_2}{p_2} \\ \vdots \\ \frac{l_\kappa}{p_\kappa} \end{pmatrix}, \tag{42}$$

*where $l_i$, for $i \in [\kappa]$, is the one-hot encoded label such that for the correct class $y$,*

$$l_i = \begin{cases} 0 & i \neq y \\ 1 & i = y. \end{cases} \tag{43}$$

*Therefore, we obtain*

$$\nabla_f \mathcal{L}_{CE}^T = -\begin{pmatrix} \frac{l_1}{p_1} & \frac{l_2}{p_2} & \cdots & \frac{l_\kappa}{p_\kappa} \end{pmatrix} (\operatorname{diag}(\mathbf{p}) - \mathbf{p}\mathbf{p}^T) \tag{44}$$

$$= -\begin{pmatrix} l_1 & l_2 & \cdots & l_\kappa \end{pmatrix} + \sum l_j \begin{pmatrix} p_1 & p_2 & \cdots & p_\kappa \end{pmatrix} \tag{45}$$

$$= \mathbf{p}^T - \boldsymbol{l}_y^T, \tag{46}$$

*where $\boldsymbol{l}_y$ denotes the one-hot encoded label vector with $1$ at the $y$-th entry, $0$ otherwise.*

**Lemma 3** *Given the assumption 1, the Lipschitz constant and smoothness of $\mathcal{L}_{CE}(\mathbf{p}(f(\boldsymbol{x}, \boldsymbol{\theta})), y)$ with respect to $\boldsymbol{\theta}$ are*

$$\|\mathcal{L}_{CE}(\mathbf{p}(f(\boldsymbol{x}, \boldsymbol{\theta}_2)), y) - \mathcal{L}_{CE}(\mathbf{p}(f(\boldsymbol{x}, \boldsymbol{\theta}_1)), y)\|_2 \leq 2L_{\boldsymbol{\theta}} \|\boldsymbol{\theta}_2 - \boldsymbol{\theta}_1\|_2 \tag{47}$$

$$\|\nabla_{\boldsymbol{\theta}} \mathcal{L}_{CE}(\mathbf{p}(f(\boldsymbol{x}, \boldsymbol{\theta}_2)), y) - \nabla_{\boldsymbol{\theta}} \mathcal{L}_{CE}(\mathbf{p}(f(\boldsymbol{x}, \boldsymbol{\theta}_1)), y)\|_2 \leq (2\beta_{\boldsymbol{\theta}} + L_{\boldsymbol{\theta}}^2) \|\boldsymbol{\theta}_2 - \boldsymbol{\theta}_1\|_2. \tag{48}$$

**Proof 3** *For the Lipschitz constant, we have*

$$\|\nabla_{\boldsymbol{\theta}} \mathcal{L}_{CE}\|_2 = \|\nabla_f \mathcal{L}_{CE}^T \nabla_{\boldsymbol{\theta}} f\| \leq 2L_{\boldsymbol{\theta}}. \tag{49}$$

*For $i = 1, 2$, let $\nabla_f \mathcal{L}_i = \nabla_f \mathcal{L}_{CE}(\mathbf{p}(f(\boldsymbol{x}, \boldsymbol{\theta}_i)), y)$, and $\nabla_{\boldsymbol{\theta}} f_i = \nabla_{\boldsymbol{\theta}} f(\boldsymbol{x}, \boldsymbol{\theta}_i)$, we have*

$$\|\nabla_{\boldsymbol{\theta}} \mathcal{L}_{CE}(\boldsymbol{x}, y, \boldsymbol{\theta}_2) - \nabla_{\boldsymbol{\theta}} \mathcal{L}_{CE}(\boldsymbol{x}, y, \boldsymbol{\theta}_1)\|_2 \tag{50}$$

$$= \|\nabla_f \mathcal{L}_2^T \nabla_{\boldsymbol{\theta}} f_2 - \nabla_f \mathcal{L}_1^T \nabla_{\boldsymbol{\theta}} f_1\|_2 \tag{51}$$

$$\leq \|\nabla_f \mathcal{L}_2^T (\nabla_{\boldsymbol{\theta}} f_2 - \nabla_{\boldsymbol{\theta}} f_1) + (\nabla_f \mathcal{L}_2 - \nabla_f \mathcal{L}_1)^T \nabla_{\boldsymbol{\theta}} f_1\|_2 \tag{52}$$

$$\leq 2\beta_{\boldsymbol{\theta}} \|\boldsymbol{\theta}_2 - \boldsymbol{\theta}_1\|_2 + L_{\boldsymbol{\theta}} \|\mathbf{p}(f(\boldsymbol{x}, \boldsymbol{\theta}_2)) - \mathbf{p}(f(\boldsymbol{x}, \boldsymbol{\theta}_1))\|_2 \tag{53}$$

$$\leq 2\beta_{\boldsymbol{\theta}} \|\boldsymbol{\theta}_2 - \boldsymbol{\theta}_1\|_2 + L_{\boldsymbol{\theta}} \|f(\boldsymbol{x}, \boldsymbol{\theta}_2) - f(\boldsymbol{x}, \boldsymbol{\theta}_1)\|_2 \tag{54}$$

$$\leq (2\beta_{\boldsymbol{\theta}} + L_{\boldsymbol{\theta}}^2) \|\boldsymbol{\theta}_2 - \boldsymbol{\theta}_1\|_2 \tag{55}$$

**Lemma 4** *Given $\boldsymbol{x} \in \mathcal{X}$, perturbation $\boldsymbol{\delta}$, we define the differences between softmax over the model such that*

$$\nu \triangleq \max_{\|\boldsymbol{\delta}\|_2 \leq \gamma, \boldsymbol{\theta} \in \Theta} \|\mathbf{p}(f(\boldsymbol{x} + \boldsymbol{\delta}, \boldsymbol{\theta})) - \mathbf{p}(f(\boldsymbol{x}, \boldsymbol{\theta}))\|_2. \tag{56}$$

*Hence, the Lipschitz and smoothness condition for the penalty function $\mathcal{C}(\boldsymbol{\delta}, \boldsymbol{x}, \boldsymbol{\theta}) = \|\mathbf{p}(f(\boldsymbol{x} + \boldsymbol{\delta}, \boldsymbol{\theta})) - \mathbf{p}(f(\boldsymbol{x}, \boldsymbol{\theta}))\|_2^2$ w.r.t. $\boldsymbol{\theta}$ is*

$$\|\mathcal{C}(\boldsymbol{\delta}, \boldsymbol{x}, \boldsymbol{\theta}_2) - \mathcal{C}(\boldsymbol{\delta}, \boldsymbol{x}, \boldsymbol{\theta}_1)\|_2 \leq (2\nu\beta\gamma + 6\nu^2 L_{\boldsymbol{\theta}}) \|\boldsymbol{\theta}_2 - \boldsymbol{\theta}_1\|_2 \tag{57}$$

$$\|\nabla_{\boldsymbol{\theta}} \mathcal{C}(\boldsymbol{\delta}, \boldsymbol{x}, \boldsymbol{\theta}_2) - \nabla_{\boldsymbol{\theta}} \mathcal{C}(\boldsymbol{\delta}, \boldsymbol{x}, \boldsymbol{\theta}_1)\|_2 \leq (6\nu^2 \beta_{\boldsymbol{\theta}} + 24\nu L_{\boldsymbol{\theta}}^2) \|\boldsymbol{\theta}_2 - \boldsymbol{\theta}_1\|_2 \tag{58}$$

**Proof 4** *We first prove the Lipschitz condition, and for simplicity, denote* $\mathbf{p} = \mathbf{p}(f(\boldsymbol{x}, \boldsymbol{\theta}))$, $f = f(\boldsymbol{x}, \boldsymbol{\theta})$ *and* $\widetilde{\mathbf{p}} = \widetilde{\mathbf{p}}(\boldsymbol{x} + \boldsymbol{\delta}, \boldsymbol{\theta})$, $\widetilde{f} = f(\boldsymbol{x} + \boldsymbol{\delta}, \boldsymbol{\theta})$. *We have*

$$\|\nabla_{\boldsymbol{\theta}} \mathcal{C}(\boldsymbol{\delta}, \boldsymbol{x}, \boldsymbol{\theta})\|_2 = 2 \left\| (\widetilde{\mathbf{p}} - \mathbf{p})^T \left( \nabla_f \widetilde{\mathbf{p}} \nabla_{\boldsymbol{\theta}} \widetilde{f} - \nabla_f \mathbf{p} \nabla_{\boldsymbol{\theta}} f \right) \right\|_2 \tag{59}$$

*where*

$$\left\| \nabla_f \widetilde{\mathbf{p}} \nabla_{\boldsymbol{\theta}} \widetilde{f} - \nabla_f \mathbf{p} \nabla_{\boldsymbol{\theta}} f \right\|_2 \leq \|\nabla_f \widetilde{\mathbf{p}}\|_2 \|\nabla_{\boldsymbol{\theta}} \widetilde{f} - \nabla_{\boldsymbol{\theta}} f\|_2 + \|\nabla_f \widetilde{\mathbf{p}} - \nabla_f \mathbf{p}\|_2 \|\nabla_{\boldsymbol{\theta}} f\|_2 \tag{60}$$

$$\leq \beta \|\boldsymbol{\delta}\|_2 + L_{\boldsymbol{\theta}} \|\nabla_f \widetilde{\mathbf{p}} - \nabla_f \mathbf{p}\|_2 \tag{61}$$

*and*

$$\|\nabla_f \widetilde{\mathbf{p}} - \nabla_f \mathbf{p}\|_2 = \| \operatorname{diag}(\widetilde{\mathbf{p}}) - \operatorname{diag}(\mathbf{p}) - \widetilde{\mathbf{p}} \widetilde{\mathbf{p}}^T + \mathbf{p} \mathbf{p}^T \|_2 \tag{62}$$

$$\leq \| \operatorname{diag}(\widetilde{\mathbf{p}}) - \operatorname{diag}(\mathbf{p}) \|_F + \|\widetilde{\mathbf{p}}\|_2 \|\widetilde{\mathbf{p}} - \mathbf{p}\|_2 + \|\widetilde{\mathbf{p}} - \mathbf{p}\|_2 \|\mathbf{p}\|_2 \tag{63}$$

$$\leq \|\widetilde{\mathbf{p}} - \mathbf{p}\|_2 + \|\widetilde{\mathbf{p}}\|_2 \|\widetilde{\mathbf{p}} - \mathbf{p}\|_2 + \|\widetilde{\mathbf{p}} - \mathbf{p}\|_2 \|\mathbf{p}\|_2 \tag{64}$$

$$\leq 3 \|\widetilde{\mathbf{p}} - \mathbf{p}\|_2. \tag{65}$$

*Hence,*

$$\left\| \nabla_f \widetilde{\mathbf{p}} \nabla_{\boldsymbol{\theta}} \widetilde{f} - \nabla_f \mathbf{p} \nabla_{\boldsymbol{\theta}} f \right\|_2 \leq \beta \|\boldsymbol{\delta}\|_2 + 3 L_{\boldsymbol{\theta}} \|\widetilde{\mathbf{p}} - \mathbf{p}\|_2 \tag{66}$$

*Then,*

$$\|\nabla_{\boldsymbol{\theta}} \mathcal{C}(\boldsymbol{\delta}, \boldsymbol{x}, \boldsymbol{\theta})\|_2 \leq 2\beta \|\boldsymbol{\delta}\|_2 \|\widetilde{\mathbf{p}} - \mathbf{p}\|_2 + 6 L_{\boldsymbol{\theta}} \|\widetilde{\mathbf{p}} - \mathbf{p}\|_2^2 \tag{67}$$

$$\leq 2\nu\beta \|\boldsymbol{\delta}\|_2 + 6\nu^2 L_{\boldsymbol{\theta}} \tag{68}$$

*In addition, for each* $i = 1, 2$, *denote* $\mathbf{p}_i = \mathbf{p}(f(\boldsymbol{x}, \boldsymbol{\theta}_i))$, $\widetilde{\mathbf{p}}_i = \widetilde{\mathbf{p}}(f(\boldsymbol{x} + \boldsymbol{\delta}, \boldsymbol{\theta}_i))$ *and* $\widetilde{f}_i = f(\boldsymbol{x} + \boldsymbol{\delta}, \boldsymbol{\theta}_i)$, $\widetilde{\mathcal{C}}_i = \mathcal{C}(\boldsymbol{x} + \boldsymbol{\delta}, \boldsymbol{\theta}_i)$ *we have*

$$\|\nabla_{\boldsymbol{\theta}} \widetilde{\mathcal{C}}_2 - \nabla_{\boldsymbol{\theta}} \widetilde{\mathcal{C}}_1\|_2 = \|\nabla_f \widetilde{\mathcal{C}}_2^T \nabla_{\boldsymbol{\theta}} \widetilde{f}_2 - \nabla_f \widetilde{\mathcal{C}}_1^T \nabla_{\boldsymbol{\theta}} \widetilde{f}_1\|_2 \tag{69}$$

$$\leq \|\nabla_f \widetilde{\mathcal{C}}_2\|_2 \|\nabla_{\boldsymbol{\theta}} \widetilde{f}_2 - \nabla_{\boldsymbol{\theta}} \widetilde{f}_1\|_2 + \|\nabla_f \widetilde{\mathcal{C}}_2 - \nabla_f \widetilde{\mathcal{C}}_1\|_2 \|\nabla_{\boldsymbol{\theta}} \widetilde{f}_1\|_2 \tag{70}$$

$$\leq \|\nabla_f \widetilde{\mathcal{C}}_2\|_2 \beta_{\boldsymbol{\theta}} \|\boldsymbol{\theta}_2 - \boldsymbol{\theta}_1\|_2 + \|\nabla_f \widetilde{\mathcal{C}}_2 - \nabla_f \widetilde{\mathcal{C}}_1\|_2 L_{\boldsymbol{\theta}} \tag{71}$$

*where*

$$\|\nabla_f \widetilde{\mathcal{C}}_2\|_2 = 2\| (\widetilde{\mathbf{p}}_2 - \mathbf{p}_2)^T (\nabla_f \widetilde{\mathbf{p}}_2 - \nabla_f \mathbf{p}_2) \|_2 \tag{72}$$

$$\leq 6 \|\widetilde{\mathbf{p}}_2 - \mathbf{p}_2\|_2^2 \tag{73}$$

$$\leq 6\nu^2 \tag{74}$$

*and*

$$\|\nabla_f \widetilde{\mathcal{C}}_2 - \nabla_f \widetilde{\mathcal{C}}_1\|_2 = 2\| (\widetilde{\mathbf{p}}_2 - \mathbf{p}_2)^T (\nabla_f \widetilde{\mathbf{p}}_2 - \nabla_f \mathbf{p}_2) - (\widetilde{\mathbf{p}}_1 - \mathbf{p}_1)^T (\nabla_f \widetilde{\mathbf{p}}_1 - \nabla_f \mathbf{p}_1) \|_2 \tag{75}$$

$$\leq 2\|\widetilde{\mathbf{p}}_2 - \mathbf{p}_2\|_2 (\|\nabla_f \widetilde{\mathbf{p}}_2 - \nabla_f \widetilde{\mathbf{p}}_1\|_2 + \|\nabla_f \mathbf{p}_2 - \nabla_f \mathbf{p}_1\|_2) \tag{76}$$

$$+ 2 (\|\widetilde{\mathbf{p}}_2 - \widetilde{\mathbf{p}}_1\| + \|\mathbf{p}_2 - \mathbf{p}_1\|_2) \|\nabla_f \widetilde{\mathbf{p}}_1 - \nabla_f \mathbf{p}_1\|_2 \tag{77}$$

$$\leq 2\|\widetilde{\mathbf{p}}_2 - \mathbf{p}_2\|_2 3 (\|\widetilde{\mathbf{p}}_2 - \widetilde{\mathbf{p}}_1\|_2 + \|\mathbf{p}_2 - \mathbf{p}_1\|_2) \tag{78}$$

$$+ 2 (\|\widetilde{\mathbf{p}}_2 - \widetilde{\mathbf{p}}_1\| + \|\mathbf{p}_2 - \mathbf{p}_1\|_2) 3 \|\widetilde{\mathbf{p}}_1 - \mathbf{p}_1\|_2 \tag{79}$$

$$= 6 (\|\widetilde{\mathbf{p}}_2 - \widetilde{\mathbf{p}}_1\|_2 + \|\mathbf{p}_2 - \mathbf{p}_1\|_2) (\|\widetilde{\mathbf{p}}_2 - \mathbf{p}_2\|_2 + \|\widetilde{\mathbf{p}}_1 - \mathbf{p}_1\|_2) \tag{80}$$

$$\leq 6 (L_{\boldsymbol{\theta}} + L_{\boldsymbol{\theta}}) (\|\widetilde{\mathbf{p}}_2 - \mathbf{p}_2\|_2 + \|\widetilde{\mathbf{p}}_1 - \mathbf{p}_1\|_2) \|\boldsymbol{\theta}_2 - \boldsymbol{\theta}_1\|_2 \tag{81}$$

$$\leq 24\nu L_{\boldsymbol{\theta}} \|\boldsymbol{\theta}_2 - \boldsymbol{\theta}_1\|_2 \tag{82}$$

*Hence,*

$$\|\nabla_{\boldsymbol{\theta}} \widetilde{\mathcal{C}}_2 - \nabla_{\boldsymbol{\theta}} \widetilde{\mathcal{C}}_1\|_2 \leq \left( 6\nu^2 \beta_{\boldsymbol{\theta}} + 24\nu L_{\boldsymbol{\theta}}^2 \right) \|\boldsymbol{\theta}_2 - \boldsymbol{\theta}_1\|_2 \tag{83}$$

**Theorem 3** *Given the Lipschitz and smoothness conditions for the model* $f$, *cross-entropy loss, and* $L_2$-*norm penalty, consider the objective function*

$$\max_{\|\boldsymbol{\delta}\|_2 \leq \gamma} \mathcal{L}_{CE}(f(\mathbf{p}(\boldsymbol{x} + \boldsymbol{\delta}, \boldsymbol{\theta})), y) + \lambda \|\mathbf{p}(f(\boldsymbol{x} + \boldsymbol{\delta}, \boldsymbol{\theta})) - \mathbf{p}(f(\boldsymbol{x}, \boldsymbol{\theta}))\|_2^2. \tag{84}$$

*We show that the objective function is* $\phi$-*approximate* $\psi$-*smooth, where*

$$\phi = (4\beta\gamma + 2\nu L_{\boldsymbol{\theta}}) + 12\lambda \left( \nu^2 \beta + 2\nu L L_{\boldsymbol{\theta}} \right) \gamma \tag{85}$$

$$\psi = \left( 2\beta_{\boldsymbol{\theta}} + L_{\boldsymbol{\theta}}^2 \right) + \lambda \left( 6\nu^2 \beta_{\boldsymbol{\theta}} + 24\nu L_{\boldsymbol{\theta}}^2 \right) \tag{86}$$

**Proof 5** *For simplicity, given $\boldsymbol{x}$ we denote*

$$\mathcal{L}_\lambda(\boldsymbol{\delta}, \boldsymbol{\theta}) = \mathcal{L}_{CE}(\boldsymbol{\delta}, \boldsymbol{\theta}) + \lambda \mathcal{C}(\boldsymbol{\delta}, \boldsymbol{\theta}). \tag{87}$$

*Consider the surrogate loss $\mathcal{L}_\lambda^{\max}(\boldsymbol{\theta}) = \max_{\|\boldsymbol{\delta}\|_2 \le \gamma} \mathcal{L}_\lambda(\boldsymbol{\delta}, \boldsymbol{\theta})$ and let*

$$\boldsymbol{\delta}_1 = \arg \max_{\|\boldsymbol{\delta}\|_2 \le \gamma} \mathcal{L}_\lambda(\boldsymbol{\delta}, \boldsymbol{\theta}_1) \tag{88}$$

$$\boldsymbol{\delta}_2 = \arg \max_{\|\boldsymbol{\delta}\|_2 \le \gamma} \mathcal{L}_\lambda(\boldsymbol{\delta}, \boldsymbol{\theta}_2). \tag{89}$$

*Without generality assume $\mathcal{L}_\lambda^{\max}(\boldsymbol{\theta}_2) \ge \mathcal{L}_\lambda^{\max}(\boldsymbol{\theta}_1)$. Hence, there exists $\boldsymbol{\delta}_1$ and $\boldsymbol{\delta}_2$ such that*

$$|\mathcal{L}_\lambda^{\max}(\boldsymbol{\theta}_2) - \mathcal{L}_\lambda^{\max}(\boldsymbol{\theta}_1)| = \mathcal{L}_\lambda(\boldsymbol{\delta}_2, \boldsymbol{\theta}_2) - \mathcal{L}_\lambda(\boldsymbol{\delta}_1, \boldsymbol{\theta}_1) \tag{90}$$

$$\le \mathcal{L}_\lambda(\boldsymbol{\delta}_2, \boldsymbol{\theta}_2) - \mathcal{L}_\lambda(\boldsymbol{\delta}_2, \boldsymbol{\theta}_1) \tag{91}$$

$$\le \mathcal{L}_{CE}(\boldsymbol{\delta}_2, \boldsymbol{\theta}_2) - \mathcal{L}_{CE}(\boldsymbol{\delta}_2, \boldsymbol{\theta}_1) + \lambda \left( \mathcal{C}(\boldsymbol{\delta}_2, \boldsymbol{\theta}_1) - \mathcal{C}(\boldsymbol{\delta}_2, \boldsymbol{\theta}_2) \right) \tag{92}$$

$$\le 2 \left[ L_{\boldsymbol{\theta}} + \lambda(\nu\beta\gamma + 3\nu^2 L_{\boldsymbol{\theta}}) \right] \|\boldsymbol{\theta}_2 - \boldsymbol{\theta}_1\|_2 \tag{93}$$

*And for smoothness, consider Frechet sub-gradient, such that $\forall \boldsymbol{g}_1 \in \partial_{\boldsymbol{\theta}} \mathcal{L}_\lambda^{\max}(\boldsymbol{\theta}_1)$ and $\forall \boldsymbol{g}_2 \in \partial_{\boldsymbol{\theta}} \mathcal{L}_\lambda^{\max}(\boldsymbol{\theta}_2)$, there exists $\boldsymbol{\delta}_1$ and $\boldsymbol{\delta}_2$ such that*

$$\|\boldsymbol{g}_2 - \boldsymbol{g}_1\|_2 \le \|\nabla_{\boldsymbol{\theta}} \mathcal{L}_\lambda(\boldsymbol{\delta}_2, \boldsymbol{\theta}_2) - \nabla_{\boldsymbol{\theta}} \mathcal{L}_\lambda(\boldsymbol{\delta}_1, \boldsymbol{\theta}_1)\|_2 \tag{94}$$

$$\le \|\nabla_{\boldsymbol{\theta}} \mathcal{L}_\lambda(\boldsymbol{\delta}_2, \boldsymbol{\theta}_2) - \nabla_{\boldsymbol{\theta}} \mathcal{L}_\lambda(\boldsymbol{\delta}_2, \boldsymbol{\theta}_1)\|_2 + \|\nabla_{\boldsymbol{\theta}} \mathcal{L}_\lambda(\boldsymbol{\delta}_2, \boldsymbol{\theta}_1) - \nabla_{\boldsymbol{\theta}} \mathcal{L}_\lambda(\boldsymbol{\delta}_1, \boldsymbol{\theta}_1)\|_2. \tag{95}$$

*For the second term of the RHS of the above inequality is*

$$\|\nabla_{\boldsymbol{\theta}} \mathcal{L}_\lambda(\boldsymbol{\delta}_2, \boldsymbol{\theta}_1) - \nabla_{\boldsymbol{\theta}} \mathcal{L}_\lambda(\boldsymbol{\delta}_1, \boldsymbol{\theta}_1)\|_2 \tag{96}$$

$$\le \|\nabla_{\boldsymbol{\theta}} \mathcal{L}_{CE}(\boldsymbol{\delta}_2, \boldsymbol{\theta}_1) - \nabla_{\boldsymbol{\theta}} \mathcal{L}_{CE}(\boldsymbol{\delta}_1, \boldsymbol{\theta}_1)\|_2 + \lambda \|\nabla_{\boldsymbol{\theta}} \mathcal{C}(\boldsymbol{\delta}_2, \boldsymbol{\theta}_1) - \nabla_{\boldsymbol{\theta}} \mathcal{C}(\boldsymbol{\delta}_1, \boldsymbol{\theta}_1)\|_2. \tag{97}$$

*And for simplicity, let $\nabla_f \mathcal{L}_i = \nabla_f \mathcal{L}_{CE}(\mathbf{p}(f(\boldsymbol{x} + \boldsymbol{\delta}_i, \boldsymbol{\theta}_1)), y), \nabla_{\boldsymbol{\theta}} f_i = \nabla_{\boldsymbol{\theta}} f(\boldsymbol{x} + \boldsymbol{\delta}_i, \boldsymbol{\theta}_1), i = 1, 2.$*

$$\|\nabla_{\boldsymbol{\theta}} \mathcal{L}_{CE}(\boldsymbol{\delta}_2, \boldsymbol{\theta}_1) - \nabla_{\boldsymbol{\theta}} \mathcal{L}_{CE}(\boldsymbol{\delta}_1, \boldsymbol{\theta}_1)\|_2 \tag{98}$$

$$\le \|\nabla_f \mathcal{L}_{CE}^T \nabla_{\boldsymbol{\theta}} f(\boldsymbol{\delta}_2, \boldsymbol{\theta}_1) - \nabla_f \mathcal{L}_{CE}^T \nabla_{\boldsymbol{\theta}} f(\boldsymbol{\delta}_1, \boldsymbol{\theta}_1)\|_2 \tag{99}$$

$$\le \|\nabla_f \mathcal{L}_2\|_2 \|\nabla_{\boldsymbol{\theta}} f_2 - \nabla_{\boldsymbol{\theta}} f_1\|_2 + \|\nabla_f \mathcal{L}_2 - \nabla_f \mathcal{L}_1\|_2 \|\nabla_{\boldsymbol{\theta}} f_1\|_2 \tag{100}$$

$$\le 2\|\nabla_{\boldsymbol{\theta}} f_2 - \nabla_{\boldsymbol{\theta}} f_1\|_2 + \|\mathbf{p}(f(\boldsymbol{x} + \boldsymbol{\delta}_2, \boldsymbol{\theta}_1)) - \mathbf{p}(f(\boldsymbol{x} + \boldsymbol{\delta}_1, \boldsymbol{\theta}_1))\|_2 L_{\boldsymbol{\theta}} \tag{101}$$

$$\le 2\beta \|\boldsymbol{\delta}_2 - \boldsymbol{\delta}_1\|_2 + 2\nu L_{\boldsymbol{\theta}} \tag{102}$$

$$\le 4\beta\gamma + 2\nu L_{\boldsymbol{\theta}} \tag{103}$$

*In addition, denote $\nabla_{\boldsymbol{\theta}} \mathcal{C}_i = \nabla_{\boldsymbol{\theta}} \mathcal{C}(\boldsymbol{\delta}_i, \boldsymbol{\theta}_1), \mathbf{p}_i = \mathbf{p}(f(\boldsymbol{x} + \boldsymbol{\delta}_i, \boldsymbol{\theta}_1)), i = 1, 2$ and $\mathbf{p} = \mathbf{p}(f(\boldsymbol{x}, \boldsymbol{\theta}_1))$, we have*

$$\|\nabla_{\boldsymbol{\theta}} \mathcal{C}_2 - \nabla_{\boldsymbol{\theta}} \mathcal{C}_1\|_2 = \|\nabla_f \mathcal{C}_2^T \nabla_{\boldsymbol{\theta}} f_2 - \nabla_f \mathcal{C}_1^T \nabla_{\boldsymbol{\theta}} f_1\|_2 \tag{104}$$

$$\le \|\nabla_f \mathcal{C}_2\|_2 \|\nabla_{\boldsymbol{\theta}} f_2 - \nabla_{\boldsymbol{\theta}} f_1\|_2 + \|\nabla_f \mathcal{C}_2 - \nabla_f \mathcal{C}_1\|_2 \|\nabla_{\boldsymbol{\theta}} f_1\|_2 \tag{105}$$

$$\le \|\nabla_f \mathcal{C}_2\|_2 \beta \|\boldsymbol{\delta}_2 - \boldsymbol{\delta}_1\|_2 + \|\nabla_f \mathcal{C}_2 - \nabla_f \mathcal{C}_1\|_2 L_{\boldsymbol{\theta}} \tag{106}$$

*where*

$$\|\nabla_f \mathcal{C}_2\|_2 = 2\|(\mathbf{p}_2 - \mathbf{p})^T (\nabla_f \mathbf{p}_2 - \nabla_f \mathbf{p})\|_2 \tag{107}$$

$$\le 6\|\mathbf{p}_2 - \mathbf{p}\|_2^2 \tag{108}$$

$$\le 6\nu^2. \tag{109}$$

*And*

$$\|\nabla_f \mathcal{C}_2 - \nabla_f \mathcal{C}_1\|_2 = 2\|(\mathbf{p}_2 - \mathbf{p})^T (\nabla_f \mathbf{p}_2 - \nabla_f \mathbf{p}) - (\mathbf{p}_1 - \mathbf{p})^T (\nabla_f \mathbf{p}_1 - \nabla_f \mathbf{p})\|_2 \tag{110}$$

$$\le 2 \left( \|\mathbf{p}_2 - \mathbf{p}\|_2 \|\nabla_f \mathbf{p}_2 - \nabla_f \mathbf{p}_1\|_2 + \|\mathbf{p}_2 - \mathbf{p}_1\|_2 \|\nabla_f \mathbf{p}_1 - \nabla_f \mathbf{p}\|_2 \right) \tag{111}$$

$$\le 6 \left( \|\mathbf{p}_2 - \mathbf{p}\|_2 + \|\mathbf{p}_1 - \mathbf{p}\|_2 \right) \|\mathbf{p}_2 - \mathbf{p}_1\|_2 \tag{112}$$

$$\le 12\nu \|f(\boldsymbol{x} + \boldsymbol{\delta}_2, \boldsymbol{\theta}_1) - f(\boldsymbol{x} + \boldsymbol{\delta}_1, \boldsymbol{\theta}_1)\|_2 \tag{113}$$

$$\le 12\nu L \|\boldsymbol{\delta}_2 - \boldsymbol{\delta}_1\|_2 \tag{114}$$

*Hence,*

$$\|\nabla_{\boldsymbol{\theta}} \mathcal{C}_2 - \nabla_{\boldsymbol{\theta}} \mathcal{C}_1\|_2 \le \left(6\nu^2\beta + 12\nu LL_{\boldsymbol{\theta}}\right) \|\boldsymbol{\delta}_2 - \boldsymbol{\delta}_1\|_2 \tag{115}$$

$$\le 12\left(\nu^2\beta + 2\nu LL_{\boldsymbol{\theta}}\right)\gamma \tag{116}$$

*In conclusion, we have*

$$\|\boldsymbol{g}_2 - \boldsymbol{g}_1\|_2 = \|\nabla_{\boldsymbol{\theta}} \mathcal{L}_\lambda(\boldsymbol{\delta}_2, \boldsymbol{\theta}_2) - \nabla_{\boldsymbol{\theta}} \mathcal{L}_\lambda(\boldsymbol{\delta}_1, \boldsymbol{\theta}_1)\|_2 \tag{117}$$

$$\le \|\nabla_{\boldsymbol{\theta}} \mathcal{L}_\lambda(\boldsymbol{\delta}_2, \boldsymbol{\theta}_2) - \nabla_{\boldsymbol{\theta}} \mathcal{L}_\lambda(\boldsymbol{\delta}_2, \boldsymbol{\theta}_1)\|_2 + \|\nabla_{\boldsymbol{\theta}} \mathcal{L}_\lambda(\boldsymbol{\delta}_2, \boldsymbol{\theta}_1) - \nabla_{\boldsymbol{\theta}} \mathcal{L}_\lambda(\boldsymbol{\delta}_1, \boldsymbol{\theta}_1)\|_2 \tag{118}$$

$$\le \left(2\beta_{\boldsymbol{\theta}} + L_{\boldsymbol{\theta}}^2 + \lambda\left(6\nu^2\beta_{\boldsymbol{\theta}} + 24\nu L_{\boldsymbol{\theta}}^2\right)\right) \|\boldsymbol{\theta}_2 - \boldsymbol{\theta}_1\|_2 + (4\beta\gamma + 2\nu L_{\boldsymbol{\theta}}) \tag{119}$$

$$+ 12\lambda\left(\nu^2\beta + 2\nu LL_{\boldsymbol{\theta}}\right)\gamma \tag{120}$$

## G.3 ANALYSIS ON RT LEARNING

Here, we provide a proof of the Lipschitz and smoothness conditions for the RT method. As shown in Wang et al. (2021), the optimization process follows gradient descent with perturbations sampled from the distribution $\Pr(\cdot \mid \boldsymbol{x})$, which can be interpreted as a form of data augmentation within standard training. Consequently, it shares the same theoretical generalization error bounds as standard training.

The pseudo-code for CVaR is shown in 1. As is shown that the $\alpha_j$ is first updated then the parameter $\boldsymbol{\theta}$. And we have that for the updated gradient is

$$\nabla_{\boldsymbol{\theta}} \left[\ell(f_{\boldsymbol{\theta}}(\boldsymbol{x}_j + \boldsymbol{\delta}_k), y_j) - \alpha_j\right]_+ \tag{121}$$

$$= \begin{cases} \nabla_{\boldsymbol{\theta}}\ell(f_{\boldsymbol{\theta}}(\boldsymbol{x}_j + \boldsymbol{\delta}_k), y_j) & \ell(f_{\boldsymbol{\theta}}(\boldsymbol{x}_j + \boldsymbol{\delta}_k), y_j) > \alpha_j \\ \left\{\boldsymbol{s} : \boldsymbol{s}^T(\boldsymbol{\theta} - \boldsymbol{\vartheta}) \le \ell \circ f_{\boldsymbol{\theta}} - \ell \circ f_{\boldsymbol{\vartheta}}, \quad \forall\boldsymbol{\vartheta}\right\} & \ell(f_{\boldsymbol{\theta}}(\boldsymbol{x}_j + \boldsymbol{\delta}_k), y_j) = \alpha_j \\ 0 & \ell(f_{\boldsymbol{\theta}}(\boldsymbol{x}_j + \boldsymbol{\delta}_k), y_j) < \alpha_j \end{cases} \tag{122}$$

Let $L$ and $\beta$ be the Lipschitz and smoothness conditions for $\ell \circ f_{\boldsymbol{\theta}}$, We show that

$$\left\|\nabla_{\boldsymbol{\theta}} \left[\ell(f_{\boldsymbol{\theta}}(\boldsymbol{x}_j + \boldsymbol{\delta}_k), y_j) - \alpha_j\right]_+\right\| \le \left\|\nabla_{\boldsymbol{\theta}}\ell(f_{\boldsymbol{\theta}}(\boldsymbol{x}_j + \boldsymbol{\delta}_k), y_j)\right\| \le L \tag{123}$$

and

$$\left\|\nabla_{\boldsymbol{\theta}} \left[\ell(f_{\boldsymbol{\theta}_2}(\boldsymbol{x}_j + \boldsymbol{\delta}_k), y_j) - \alpha_j\right]_+ - \nabla_{\boldsymbol{\theta}} \left[\ell(f_{\boldsymbol{\theta}_1}(\boldsymbol{x}_j + \boldsymbol{\delta}_k), y_j) - \alpha_j\right]_+\right\| \le \max\{L, \beta\}. \tag{124}$$

According to the training algorithm in Alg. 1, the weight updates are Lipschitz smooth for most of the training process. Even in the worst case, the smoothness remains bounded by the Lipschitz constant, which largely explains the reduced robust overfitting.

---

**Algorithm 1** Probabilistically Robust Learning (PRL)

---

**Hyperparameters:** sample size $M$, step sizes $\eta_\alpha, \eta > 0$, robustness parameter $\rho > 0$, neighborhood distribution $\tau$, num. of inner optimization steps $T$, batch size $B$
**repeat**
    **for** minibatch $\{(x_j, y_j)\}_{j=1}^B$ **do**
        **for** $T$ steps **do**
            Draw $\delta_k \sim \tau, k = 1, \dots, M$
            $g_{\alpha_n} \leftarrow 1 - \frac{1}{\rho M} \sum_{k=1}^M \mathbb{I}\left[\ell(f_\theta(x_j + \delta_k), y_j) \ge \alpha_j\right]$
            $\alpha_j \leftarrow \alpha_j - \eta_\alpha g_{\alpha_j}, \quad n = 1, \dots, B$
        **end for**
        $g \leftarrow \frac{1}{\rho MB} \sum_{j,k} \nabla_\theta \left[\ell(f_\theta(x_j + \delta_k), y_j) - \alpha_j\right]_+$
        $\theta \leftarrow \theta - \eta g$
    **end for**
**until** convergence

---

## G.4 ANALYSIS ON AT-PR

We show that the AT-PR algorithm is essentially a variant of PGD. Instead of performing a single PGD run, AT-PR executes multiple PGD runs and selects the one that yields the widest coverage over

---

**Algorithm 2** PGD and gradient-based search for $\boldsymbol{x}'_{pr}$

---

**Require:** Inputs $[X, Y], N, \alpha_{min}, \alpha_{max}, step_{min}, step_{max}$

1: Initialize $AE_s \leftarrow []$            ▷ Initialize AE candidate set
2: Apply PGD to get different AE candidates
3: **for** idx $= 0$ **to** $N$ **do**
4:      $\boldsymbol{x}_{init} \leftarrow \boldsymbol{x} + \text{Uniform}(-\gamma, \gamma)$
5:      $\alpha \leftarrow \text{Uniform}(\alpha_{min}, \alpha_{max})$
6:      $steps \leftarrow \text{Uniform}(step_{min}, step_{max})$
7:      $\boldsymbol{x}'_{idx} \leftarrow pgd(\boldsymbol{x}_{init}, y, \gamma, \alpha, steps)$
8:      Append $\boldsymbol{x}'_{idx}$ to $AE_s$
9: **end for**
**Require:** $AE_s, \boldsymbol{x}, y, C$
10: Initialize $Max\_d \leftarrow 0$ ; $\boldsymbol{x}'_{pr} \leftarrow$ None
11: **for** each $\boldsymbol{x}' \in AE_s$ **do**
12:      $\tilde{\boldsymbol{x}} = \boldsymbol{x}'$
13:      **while** $iter < C$ **do**
14:          $y' = f(\tilde{x})$
15:          **if** $y' = y$ **then**
16:              break          ▷ Exit if $\tilde{\boldsymbol{x}}$ is classified correctly
17:          **end if**
18:          $g \leftarrow \nabla_{\tilde{x}} L(\tilde{\boldsymbol{x}}, y)$          ▷ Compute gradient
19:          $\tilde{\boldsymbol{x}} = \tilde{\boldsymbol{x}} - \alpha \cdot g$          ▷ Update $\tilde{\boldsymbol{x}}$
20:      **end while**
21:      $d \leftarrow D(\tilde{\boldsymbol{x}}, \boldsymbol{x}')$          ▷ Distance between $\tilde{\boldsymbol{x}}$ & $\boldsymbol{x}'$
22:      **if** $d > Max\_d$ **then**
23:          $\boldsymbol{x}'_{pr} = \boldsymbol{x}'$
24:          $Max\_d = d$
25:      **end if**
26: **end for**
27: **Output** $\boldsymbol{x}'_{pr}$          ▷ Farthest AE from decision boundary.

---

the perturbation budget for the given inputs. Therefore, its generalization analysis still follows our framework in Thm. 1.

As illustrated in Alg. 2, AT-PR first generates a set of AE candidates. For each candidate, it computes the distance to the decision boundary and selects the one with the maximum distance. Under Assumption 1, we recall that for all $\boldsymbol{x} \in \mathcal{X}$ the curvature is globally bounded:

$$\|\nabla_{\boldsymbol{\theta}} f(\boldsymbol{x}, \boldsymbol{\theta}_2) - \nabla_{\boldsymbol{\theta}} f(\boldsymbol{x}, \boldsymbol{\theta}_1)\| \leq \beta_{\boldsymbol{\theta}} \|\boldsymbol{\theta}_2 - \boldsymbol{\theta}_1\|. \tag{125}$$

Although this bound holds globally, in practice $\boldsymbol{x}$ may lie near different local optima. Let $\Delta_i \subseteq \mathcal{X}, i \in [N]$ denote the local region around adversarial examples (local optima found by PGD). We then define local smoothness bounds for each $\Delta_i$, such that for all $i \in [N]$:

$$\forall \boldsymbol{x} \in \Delta_i \quad \|\nabla_{\boldsymbol{\theta}} f(\boldsymbol{x}, \boldsymbol{\theta}_2) - \nabla_{\boldsymbol{\theta}} f(\boldsymbol{x}, \boldsymbol{\theta}_1)\| \leq \beta_{\boldsymbol{\theta}}^{(i)} \|\boldsymbol{\theta}_2 - \boldsymbol{\theta}_1\|. \tag{126}$$

Alg. 2 essentially searches for the region $\Delta_i$ with the largest coverage along the loss surface. Two cases arise from this selection, which are also discussed in Zhang et al. (2025):

1. The region $\Delta_i$ with the largest coverage is relatively flat but not the one with the "highest peak" (short and wide peak in loss landscape). This corresponds to a smaller smoothness bound, i.e., $\beta_{\boldsymbol{\theta}}^{(i)} < \beta_{\boldsymbol{\theta}}$.

2. The region $\Delta_i$ with the largest coverage also has the "highest peak" (both tall and wide peak in loss landscape), indicating a better local optima that is closer to the global one, i.e., $\beta_{\boldsymbol{\theta}}^{(i)} \approx \beta_{\boldsymbol{\theta}}$.

In the first case, the generalization gap decreases (as shown in Thm. 1), which corresponds to the experimental results of WRN-28-10 on CIFAR-100. In the second case, the generalization gap of AT-PR is comparable to that of PGD, as shown in the CIFAR-10 experiments (See Table. 10). Our experiments, along with results from Zhang et al. (2025), demonstrate that approximately 90% of inputs correspond to the second case. This corresponds to the special case where only a single AE candidate is considered in Alg. 2.

### G.5 UNIFORM ALGORITHM STABILITY ANALYSIS ON ADVERSARIAL TRAINING

**Lemma 5** *Let $f : \mathbb{R}^d \times \mathbb{R}^m \to \mathbb{R}$, for all $z \in \mathbb{R}^d$, $f$ be $\eta$-approximate $\beta$-smooth. Given the weight update rule $G(\boldsymbol{\theta}) = \boldsymbol{\theta} - \alpha \nabla_{\boldsymbol{\theta}} f(\boldsymbol{z}, \boldsymbol{\theta})$ by SGD, we have*

$$\|G(\boldsymbol{\theta}_2) - G(\boldsymbol{\theta}_1)\|_2 \leq (1 + \alpha_t \beta)\|\boldsymbol{\theta}_2 - \boldsymbol{\theta}_1\|_2 + \eta \tag{127}$$

**Proof 6**

$$\|G(\boldsymbol{\theta}_2) - G(\boldsymbol{\theta}_1)\|_2 = \|\boldsymbol{\theta}_2 - \alpha_t \nabla_{\boldsymbol{\theta}} f(\boldsymbol{z}, \boldsymbol{\theta}_2) - (\boldsymbol{\theta}_1 - \alpha_t \nabla_{\boldsymbol{\theta}} f(\boldsymbol{z}, \boldsymbol{\theta}_1))\|_2 \tag{128}$$
$$\leq \|\boldsymbol{\theta}_2 - \boldsymbol{\theta}_1\|_2 + \alpha_t \|\nabla_{\boldsymbol{\theta}} f(\boldsymbol{z}, \boldsymbol{\theta}_2) - \nabla_{\boldsymbol{\theta}} f(\boldsymbol{z}, \boldsymbol{\theta}_1)\|_2 \tag{129}$$
$$\leq (1 + \alpha_t \beta)\|\boldsymbol{\theta}_2 - \boldsymbol{\theta}_1\|_2 + \eta \tag{130}$$

**Lemma 6** *Let $\ell : \mathcal{X} \times \mathcal{Y} \times \Theta \to [0,1]$ be a general loss function, and $\ell(z, \cdot), z \in \mathcal{X} \times \mathcal{Y}$ is nonnegative and $L$-Lipschitz for all $z$. $S$ and $S'$ are two sets of samples differing in only one example. Consider 2 trajectories of parameters $\boldsymbol{\theta}_t, \boldsymbol{\theta}'_t, t = 1, 2, \ldots T$ that generated by SGD with sets of samples $S$ and $S'$, respectively. Then, $\forall z \in \mathcal{X} \times \mathcal{Y}$ and $\forall t_0 \in \{0, 1, \ldots, n\}$, we have*

$$\mathbb{E}\left[\ell(z, \boldsymbol{\theta}_T) - \ell(z, \boldsymbol{\theta}'_T)\right] \leq \frac{t_0}{n} + L\mathbb{E}\left[\delta_T \mid \delta_{t_0} = 0\right]. \tag{131}$$

*where $\delta_t = \|\boldsymbol{\theta}_t - \boldsymbol{\theta}'_t\|$, for $t = 1, 2, \ldots T$.*

**Proof 7** *Let $z \in \mathcal{X} \times \mathcal{Y}$ be an arbitrary example. We have,*

$$\mathbb{E}[\ell(z, \boldsymbol{\theta}_T) - \ell(z, \boldsymbol{\theta}'_T)] = \mathbb{P}\{\delta_{t_0} = 0\}\mathbb{E}[\ell(z, \boldsymbol{\theta}_T) - \ell(z, \boldsymbol{\theta}'_T) \mid \delta_{t_0} = 0] \tag{132}$$
$$+ \mathbb{P}\{\delta_{t_0} \neq 0\}\,\mathbb{E}[\ell(z, \boldsymbol{\theta}_T) - \ell(z, \boldsymbol{\theta}'_T) \mid \delta_{t_0} \neq 0] \tag{133}$$
$$\leq \mathbb{E}[|\ell(z, \boldsymbol{\theta}_T) - \ell(z, \boldsymbol{\theta}'_T)| \mid \delta_{t_0} = 0] + \mathbb{P}\{\delta_{t_0} \neq 0\} \cdot \sup_{\boldsymbol{\theta}, z} \ell(z, \boldsymbol{\theta}) \tag{134}$$
$$\leq L\,\mathbb{E}[\|\boldsymbol{\theta}_T - \boldsymbol{\theta}'_T\| \mid \delta_{t_0} = 0] + \mathbb{P}\{\delta_{t_0} \neq 0\}. \tag{135}$$

*Under the random permutation rule, we have*

$$\mathbb{P}\{\delta_{t_0} \neq 0\} \leq \frac{t_0}{n}. \tag{136}$$

**Theorem 4** *Let $\ell(z, \cdot) \in [0,1]$ be $L$-Lipschitz and $\eta$-approximate $\beta$-smooth loss function for all $z \in \mathcal{X} \times \mathcal{Y}$. Given a constant $c$, we run SGD with learning rate $\alpha_t \leq c/t$. Then, the algorithm stability is bounded as*

$$|\mathcal{E}_{stab}| \leq \frac{1}{n} + \frac{2L^2 + nL\eta}{\beta(n-1)}T^{c\beta}. \tag{137}$$

**Proof 8** *For simplicity, denote $\Delta_t = \mathbb{E}[\delta_t \mid \delta_{t_0} = 0]$. Hence, follow Lem. 6, we have that there is only probability of $\frac{1}{n}$ that $z' \in S'/S$ will be selected by SGD, denoted as event $Z = z'$, hence we have*

$$\Delta_{t+1} = \mathbb{P}\{Z = z'\}\mathbb{P}\{\Delta_{t+1} \mid Z = z'\} + \mathbb{P}\{Z \neq z'\}\mathbb{P}\{\Delta_{t+1} \mid Z \neq z'\} \tag{138}$$
$$\leq \left(1 - \frac{1}{n}\right)(1 + \alpha_t\beta)\Delta_t + \frac{1}{n}\Delta_t + \alpha_t\left(\eta + \frac{2L}{n}\right) \tag{139}$$
$$= \left(1 + \left(1 - \frac{1}{n}\right)\frac{c\beta}{t}\right)\Delta_t + \frac{c}{t}\left(\eta + \frac{2L}{n}\right) \tag{140}$$
$$\leq \exp\left((1 - 1/n)\frac{c\beta}{t}\right)\Delta_t + \frac{c}{t}\left(\eta + \frac{2L}{n}\right). \tag{141}$$

*The last inequality comes from the fact that $1 + x \leq \exp(x)$ for all $x$. Thus,*

$$\Delta_T \leq \sum_{t=t_0+1}^{T} \left[ \prod_{k=t+1}^{T} \exp\left( \left(1 - \frac{1}{n}\right) \frac{c\beta}{k} \right) \frac{c}{t} \left( \eta + \frac{2L}{n} \right) \right] \tag{142}$$

$$= \sum_{t=t_0+1}^{T} \exp\left( \left(1 - \frac{1}{n}\right) c\beta \sum_{k=t+1}^{T} \frac{1}{k} \right) \frac{c}{t} \left( \eta + \frac{2L}{n} \right) \tag{143}$$

$$\leq \sum_{t=t_0+1}^{T} \exp\left( \left(1 - \frac{1}{n}\right) c\beta \log \frac{T}{t} \right) \frac{c}{t} \left( \eta + \frac{2L}{n} \right) \tag{144}$$

$$= \left( \eta + \frac{2L}{n} \right) c T^{c\beta\left(1 - \frac{1}{n}\right)} \sum_{t=t_0+1}^{T} t^{-c\beta\left(1 - \frac{1}{n}\right) - 1} \tag{145}$$

$$\leq \left( \eta + \frac{2L}{n} \right) \frac{c}{(1 - 1/n)\, c\beta} \left( \frac{T}{t_0} \right)^{c\beta(1 - 1/n)} \tag{146}$$

$$\leq \frac{2L + n\eta}{\beta(n - 1)} \left( \frac{T}{t_0} \right)^{c\beta}. \tag{147}$$

*Hence, we get*

$$\mathbb{E}\left[ |f(\boldsymbol{z}, \boldsymbol{\theta}_T) - f(\boldsymbol{z}, \boldsymbol{\theta}'_T)| \right] \leq \frac{t_0}{n} + \frac{2L^2 + nL\eta}{\beta(n - 1)} \left( \frac{T}{t_0} \right)^{c\beta}. \tag{148}$$

*Letting $q = \beta c$, the right-hand side is approximately minimized when*

$$t_0 = \left[ c \left( 2L^2 + nL\eta \right) \right]^{\frac{1}{q+1}} T^{\frac{q}{q+1}}. \tag{149}$$

*Hence,*

$$\mathbb{E}\left[ |f(\boldsymbol{z}, \boldsymbol{\theta}_T) - f(\boldsymbol{z}, \boldsymbol{\theta}'_T)| \right] \leq \frac{1 + \frac{1}{c\beta}}{n - 1} \left[ c \left( 2L^2 + nL\eta \right) \right]^{\frac{1}{c\beta+1}} T^{\frac{c\beta}{c\beta+1}}. \tag{150}$$

*Since $T$ is arbitrary, and $t_0 \in \{1, 2, \ldots, n\}$, when $T$ is large, let $t_0 = 1$ we have*

$$\mathbb{E}\left[ |f(\boldsymbol{z}, \boldsymbol{\theta}_T) - f(\boldsymbol{z}, \boldsymbol{\theta}'_T)| \right] \leq \frac{1}{n} + \frac{2L^2 + nL\eta}{\beta(n - 1)} T^{c\beta}. \tag{151}$$

**Theorem 5 (Generalization in Expectation by Hardt et al. (2016))** *If the algorithm $\mathcal{A}$ is $\epsilon$-stable, then*

$$\mathbb{E}_{\mathcal{A}, S}[R(\mathcal{A}(S)) - R_S(\mathcal{A}(S))] < \epsilon \tag{152}$$

*where the expectation is over algorithm $\mathcal{A}$ and the training set $S = \{\boldsymbol{z}_1, \ldots, \boldsymbol{z}_n\}$ where each $\boldsymbol{z}_i = (\boldsymbol{x}_i, y_i) \overset{i.i.d.}{\sim} \mathcal{D}$ for classification. And,*

$$R(\mathcal{A}(S)) = \mathbb{E}_{\boldsymbol{z} \sim \mathcal{D}} \left[ \ell\left( \boldsymbol{h}_{\boldsymbol{\theta}}, \boldsymbol{z} \right) \right] \tag{153}$$

$$R_S(\mathcal{A}(S)) = \frac{1}{n} \sum_{i=1}^{n} \ell\left( \boldsymbol{h}_{\boldsymbol{\theta}}, \boldsymbol{z}_i \right) \tag{154}$$

*where $\ell$ denotes the loss function, and $\boldsymbol{z} = (\boldsymbol{x}, \boldsymbol{y}) \sim \mathcal{D}$ represents the inputs. $\boldsymbol{h}$ is the hypothesis parameterized by $\theta \in \Theta$. Since the parameters are generated by an algorithm $\mathcal{A}$ from the data set $S$, we have*

$$\boldsymbol{\theta} = \mathcal{A}(S) \tag{155}$$

**Proof 9** *Consider the fact*

$$\mathbb{E}[Y] = \mathbb{E}[\mathbb{E}[Y|X]], \tag{156}$$

*For simplicity, denote the $\ell(\boldsymbol{h_\theta}, \boldsymbol{z}) = f(\mathcal{A}(S), \boldsymbol{z})$.*

$$\mathbb{E}\left[\mathbb{E}\left[R(\mathcal{A}(S)) - R_S(\mathcal{A}(S)) \mid S\right]\right] \tag{157}$$

$$= \mathbb{E}\left[\mathbb{E}\left[\mathbb{E}[f(\mathcal{A}(S), \boldsymbol{z}) \mid \mathcal{A}, S] - \frac{1}{n}\sum_{i=1}^{n} f(\mathcal{A}(S), \boldsymbol{z}_i) \mid S\right]\right] \tag{158}$$

$$= \mathbb{E}\left[\mathbb{E}\left[\mathbb{E}[f(\mathcal{A}(S), \boldsymbol{z}) \mid \mathcal{A}, S] - \frac{1}{n}\sum_{i=1}^{n} f(\mathcal{A}(S^{(i)}), \boldsymbol{z}'_i) \mid S\right]\right] \tag{159}$$

$$= \mathbb{E}\left[\mathbb{E}\left[\mathbb{E}\left[\frac{1}{n}\sum_{i=1}^{n} f(\mathcal{A}(S), \boldsymbol{z}'_{\boldsymbol{i}}) \mid \mathcal{A}, S\right] - \frac{1}{n}\sum_{i=1}^{n} f(\mathcal{A}(S^{(i)}), \boldsymbol{z}'_i) \mid S\right]\right] \tag{160}$$

$$= \mathbb{E}\left[\mathbb{E}\left[\mathbb{E}\left[\frac{1}{n}\sum_{i=1}^{n}\left(f(\mathcal{A}(S), \boldsymbol{z}'_{\boldsymbol{i}}) - f(\mathcal{A}(S^{(i)}), \boldsymbol{z}'_i)\right) \mid \mathcal{A}, S\right] \mid S\right]\right] \tag{161}$$

$$\leq \sup_{S,S',z} \mathbb{E}\left[f(\mathcal{A}(S), \boldsymbol{z_i}) - f(\mathcal{A}(S'), \boldsymbol{z}_i)\right] \leq \epsilon \tag{162}$$

*where $S'$ is at most one date point different from $S$ and $S^{(i)}$ is the training set that the $i$-th date point is $\boldsymbol{z}'_i$.*