# OpenReview forum: "Probabilistic Robustness for Free? Revisiting Training via a Benchmark"
_ICLR.cc/2026/Conference — Submitted to ICLR 2026_

### Official Review · Reviewer_hsym · 2025-10-26

**Soundness:** 3
**Presentation:** 2
**Contribution:** 3
**Rating:** 6
**Confidence:** 3

**Summary:**

This paper proposes the PRBench, a unified evaluation benchmark for adversarial robustness and probabilistic robustness.

The authors compared multiple robust learning methods across several model architectures, concluding that robustness generalizes from the adversarial setting to the probabilistic setting but not vice versa, while models trained for probabilistic robustness see smaller generalization gaps.

Additionally, the authors presented theoretical analyses on robust generalization and robust training objective smoothness.

**Strengths:**

This paper is valuable for understanding the relationship between adversarial and probabilistic robustness, providing important insight for building real-world dependable deep learning systems.

**Weaknesses:**

The paper presents some exciting theoretical analyses and a valuable benchmark. However, their connections are not entirely clear. Specifically, Theorem 1 seems disjoint from the rest of the paper, and Theorem 2 appears in a completely different section. I believe the paper would become clearer with the following modifications:
- Add some discussions about how Theorem 1 is related to the creation of the PRBench.
- Reorganize Theorems 1 and 2 into a theoretical analysis section before the discussion of the benchmarking results.

The description "some function is smoothness" appears repetitively in the paper, making it somewhat unnatural to read. For example, Definition 3 says "$f$ is $\\beta$-smoothness". I suggest changing such descriptions to "some function is smooth", like $f$ is $\\beta$-smooth" in places like Definition 3, Theorem 1, and Theorem 2.

**Questions:**

The authors mentioned that "For the two PR metrics, $| \mathcal{D} |$ denotes the number of test samples that are correctly classified by the model, whereas for AR, $| \mathcal{D} |$ refers to the total number of test samples. Why do we have this discrepancy?

---

> ### Author Response · Authors · 2025-11-20
> **Authors’ response**
>
> **Q1** We thank the reviewer for the suggestions.
>
> - We will reorganize Theorems 1 and 2 into a unified theoretical analysis section to make the theoretical analysis and experimental discussions more closely connected. In brief, Theorem 1 aims to establish the fundamental theoretical basis for our GE analysis: it formalizes the Lipschitzness and smoothness assumption of the adversarial surrogate loss follows from the results in Xiao et al. (2022b), which is the key prerequisite for analyzing robust overfitting. Theorem 2 builds directly on Theorem 1 by extending these properties to the AT objective with regularization, quantifying how the added regularizer changes the Lipschitz and smoothness constants and thereby affects the GE behavior observed in our empirical result. We will revise the organization to make this connection clearer in the next version.
>
> - We will also revise the phrasing of "f is $\beta$-smoothness” to “some function is $\beta$-smooth” for clarity and consistency.
>
>
> ---
> **Q2** This discrepancy arises because, in previous work (e.g., CVaR), they include misclassified original clean samples when evaluating PR, rather than focusing on correctly classified samples. We have corrected this in our experiments *"for the PR metrics, $|D|$ denotes the number of test samples that are correctly classified by the model"* and clarified this point in the original paper (lines 366-375):
>
> The main reasons for such correction is : *Including misclassified clean samples when computing PR fails to reflect the true robustness of individual examples*. This view is supported by prior studies, e.g., Li et al. (2024) [1] states that *"when considering the adversarial robustness of a wrongly classified sample $x$, the robustness should be 0."* Similarly, Chen \& Lee (2024) [2] emphasize that *"adversarial examples must satisfy the constraints: the original examples are classified correctly while the predictions of the adversarial examples are wrong."* In other words, if a sample is misclassified by the model initially, its ability to resist perturbations and maintain the wrong prediction (i.e., being *"robustly wrong"*) does not constitute meaningful robustness.
>
>
> [1] Z. Li, et al. "Great Score: Global Robustness Evaluation of Adversarial Perturbation Using Generative Models." NeurIPS. 2024
>
> [2] Chen, E, et al. "Data filtering for efficient adversarial training". Pattern Recognition. 2024

---

> > ### Comment · Reviewer_hsym · 2025-11-20
> > **Thank you for the response**
> >
> > I appreciate the clarifications from the authors.
> >
> > While the authors claimed that they will reorganize the theoretical results and revise the "smoothness" phrasing, I was not able to find these modification in the manuscript.
> >
> > Could you please highlight where the edits are? Thank you.

---

> > > ### Author Response · Authors · 2025-11-20
> > > **Authors’ response**
> > >
> > > We thank the reviewer for the response. We are currently revising the main manuscript by incorporating the comments from all reviewers. We will update the revised paper within 2-3 days, and all changes will be highlighted.

---

> > > ### Author Response · Authors · 2025-11-22
> > > **Authors’ response**
> > >
> > > We thank reviewer again for the feedback. We have now updated the manuscript by incorporating all reviewers’ comments, and the corresponding revisions are highlighted in blue.

---

> > > > ### Comment · Reviewer_hsym · 2025-11-25
> > > > **Thank you for the response.**
> > > >
> > > > I appreciate the authors for making the edits and highlighting the changes. Looks good to me!

---

> > > > > ### Author Response · Authors · 2025-11-25
> > > > > **Authors’ response**
> > > > >
> > > > > We thank the reviewer for the feedback. The comments and suggestions have been helpful in improving our manuscript. We sincerely appreciate reviewer's time and support.

---

### Official Review · Reviewer_KUcR · 2025-10-30

**Soundness:** 2
**Presentation:** 3
**Contribution:** 2
**Rating:** 4
**Confidence:** 5

**Summary:**

This paper introduces PRBench, which compares the adversarial robustness, probabilistic robustness, and generalization error of a variety of adversarial and PR-targeted training methods. While adversarial training outperforms PR-targeted training in probabilistic robustness in most cases, it also yields a higher generalization error. This paper involves upper bounds on the expected stability for both training methods, which implies that adversarial training theoretically leads to a higher generalization error.

**Strengths:**

1. The author conducts extensive experiments to demonstrate that AT outperforms PR-targeted training in adversarial and probabilistic robustness but results in a higher generalization error. The evidence derived from various model architectures and standard datasets is consistent and compelling.

2. The author comprehensively illustrates the multi-fold trade off among (PR, AR), GE, and efficiency, which may illuminate future research to appropriately handle them.

**Weaknesses:**

1. There are some confused descriptions in the relationship among p, f, and L, for instance, in Eq. 1, p is a softmax function outside the model f, but in Thm.1, the project p is an inner composition of the model f. Besides, the author seems mistakenly using the model f as the loss function L in Proofs located in appendix F. 5.

2. The author overlooks a critical assumption for Lemma 2 that p_i>0 must hold when for l_i=1. If l_i=1 and p_i=0, the gradient of the cross-entropy loss would be infinity, the loss is not Lipschitz in this case. Indeed, cross-entropy loss is deemed as a non-Lipschitz one due to such potential infinite discontinuities. I suggest the author to impose some constraint on the value of p_i, for example, if x is classified correctly, the author could obtain a lower bound 1/|Y|  of p_i in this case and can further derive a tighter Lipschitz constant since p and l are in the same direction, which could be beneficial to the subsequent proofs.

3. The author unreasonably omits the supremum of the loss function in Line 134 in Proof 7. The author should explain this operation since the supremum of the cross-entropy function is obviously infinity, so the expected stability in Lemma 6 is seemly unbounded, which contradicts to current result of Lemma 6.

4. The author adopts the standard Lipschitz and smoothness assumptions for the model f in the theoretical analysis. However, the architectures employed in the experiments do not strictly satisfy these conditions. This discrepancy raises the question of whether the empirical observations on GE can be interpreted by theoretical results.

5. Although the claim “AT outperforms PR-targeted training in adversarial and probabilistic robustness but results in a higher generalization error.” is substantially guaranteed by experiments, the author doesn’t display any convinced explanation on that. I think there may exist some intrinsic relationship between adversarial and probabilistic robustness that has not been exploited.

**Questions:**

In addition to my comments in the above "weakness" section, I also believe that the paper would have notable contribution if the author could further provide some persuasive explanations on the relationship between adversarial and probabilistic robustness (as mentioned in Weakness #5.).

---

> ### Author Response · Authors · 2025-11-20
> **Authors’ response**
>
> **Q1** We thank the reviewer for pointing out this typo. In Thm. 1, the use of $p$ and $f$ was incorrect due to a typo. We will correct this to ensure consistency.
>
> ---
> **Q2** We thank the reviewer for the comment. Let $z = f(x)$ denote the classifier logits, consider a one-hot encoding for the label $y$ in Eq.43. We agree that the cross-entropy defined as $L_{CE}(p, y) = \sum_{i=1}^{\kappa} l_i \log p_i$
> is not globally Lipschitz unless imposes appropriate lower bounds on $p\_i$. However, in Lemma2 our focus is on the composition of the softmax and the cross-entropy (we use cross-entropy loss to make this distinction, cf. line 1560), i.e.,
> $L_{CE}(p(z), y), \text{ where } p(z) = \begin{pmatrix}
>     \frac{e^{z_1}}{\sum_{j}e^{z_j}} \\\\
>     \frac{e^{z_2}}{\sum_{j}e^{z_j}} \\\\
>     \vdots \\\\
>     \frac{e^{z_{\kappa}}}{\sum_{j}e^{z_j}}
> \end{pmatrix}$
> as a function of the logits $z \in R^{\kappa}$. This composite function is globally Lipschitz, since the infinite term in the derivative of the cross-entropy is canceled by the corresponding one in the softmax derivative, as shown in Eq.44. Moreover, since the probabilities $p_i$ produced by the softmax are strictly positive, the case $p_i = 0$ is impossible.
>
> Similar proof of the Lipschitzness of $L\_{CE}(p(z), y)$ is in Lemma3 of the App. in [2], which provides an explicit Lipschitz bound for the softmax cross-entropy loss. Moreover, for binary classification of $\kappa = 2$, $L_{CE}(p(z), y)$ becomes logistic loss and it is 1-Lipschitz mentioned in [1] (Introduction p.1).
>
> [1] "Logistic regression: Tight bounds for stochastic and online optimization. Conference on Learning Theory." PMLR, 2014.
>
> [2] "On transfer of adversarial robustness from pretraining to downstream tasks." NeurIPS23
>
> ---
> **Q3** We thank the reviewer for the comment. Regarding Lemma 6, it already assumes a general loss function (Line 1854) $\ell:\mathcal{X}\times\mathcal{Y}\times\Theta\to [0,1],$ meaning $\ell$ is assumed to be uniformly upper bounded, hence its supremum is also upper bounded by 1.
>
> ---
> **Q4**
> Using GELU/SoftPlus for Res18/CIFAR-10
> |Type|Method|Acc(GELU)|PGD(GELU)|PR0.03(GELU)|PR0.08(GELU)|GE(GELU)|Acc(SP)|PGD(SP)|PR0.03(SP)|PR0.08(SP)|GE(SP)|
> |-|-|-|-|-|-|-|-|-|-|-|-|
> |Std.|ERM|94.38|0.0|96.4|72.54|0.0|94.24|0.0|96.45|70.58|0.0|
> |PR-t.|Corr_Uni.|94.08|0.01|99.13|90.53|0.0|94.25|0.0|99.08|89.67|0.0|
> |AT|PGD|80.29|51.23|99.55|98.02|5.4|81.17|51.84|99.64|98.08|5.92|
> ||TRADES|79.88|52.45|99.46|97.42|3.59|80.78|52.39|99.53|97.46|4.29|
> ||MART|76.88|53.85|99.6|97.52|3.55|77.19|54.4|99.44|97.28|4.2|
>
> We replaced ReLU in Res18 with the smooth activations: GELU/softplus(SP). The GE remained consistent with our theoretical analysis (cf. Sec.5.2), which indicates that the theory captures the essential mechanisms underlying GE and continues to explain the empirical behavior.
>
> ---
> **Q5–6** We appreciate the reviewer’s concern regarding the explanation, we would like to clarify that we provide the theoretical analysis in App. G.3–G.5 (line 1740-1965). These sections analyze CVaR, AT-PR, and AT, relate their behavior to the GE properties, and thereby help explain why AT outperforms PR-targeted training while still maintaining a lower GE. Regarding the intrinsic relationship between AR and PR, beyond the preliminaries in Sec.2, we formally discuss this as follows. Specifically, the AR and PR are defined as:
> $$
> AR(x,\gamma) =
> \sup\_{\\|\epsilon\\|\le \gamma}
> L(x+\epsilon, y),
> \quad
> PR(x,\gamma) =
> E\_{\substack{\epsilon\sim Pr(\cdot\mid x)\\\\\\|\epsilon\\|\le \gamma}}
> [ I\_{f(x+\epsilon) = y} ].
> $$
>
> **Proposition** Given the def. of AR and PR, and a perturbation distribution $Pr(\cdot\mid x) \in P_{\epsilon}$, then $AR(x,\gamma) \le PR(x,\gamma)$
>
> If allow $P_{\epsilon}$ to be unrestricted, representing any family of distributions  (including the Dirac delta measure), then
> $AR(x,\gamma) = PR(x,\gamma)$
>
> **Proof** For any input $x$, let $\epsilon^{\star}$ denote AE, $Pr^{\star}(\cdot\mid x)$ be the optimal perturbation distribution of PR. Then:
> $$
> \epsilon^{\star} \in
> \arg\sup\_{\\|\epsilon\\| \le \gamma}
> I\_{f(x+\epsilon) \neq y}.
> $$
> $$
> Pr^{\star}(\cdot\mid x) =
> \arg\inf\_{Pr\in P\_{\epsilon}}
> E\_{\epsilon\sim Pr(\cdot\mid x)}
> \big[ I\_{f(x+\epsilon) = y} \big].
> $$
> Let
> $$
> PR(x,\gamma) =
> E\_{Pr^{\star}}
> [I\_{f(x+\epsilon) = y}].
> $$
> If  $E\_{Pr^{\star}}[I_{f(x+\epsilon) = y}] < 1,$ there must exist at least one $\epsilon$ with $\\|\epsilon\\| \le \gamma$ such that  $f(x+\epsilon^{\star}) \neq y$, therefore $I\_{f(x+\epsilon^{\star}) = y} = 0,$ hence
> $$
> AR(x,\gamma)=I\_{f(x+\epsilon^{\star}) = y}
> \le
> E\_{Pr^{\star}}
> [I\_{f(x+\epsilon) = y}].
> $$
>
> Considering the Dirac delta measure $\delta_{\epsilon^{\star}}$, then:
> $$
> AR(x,\gamma)= I\_{ f(x+\epsilon^{\star}) = y }= E\_{\delta\_{\epsilon^{\star}}}[I\_{f(x+\epsilon)=y}]\ge \inf\_{Pr}E\_{Pr}[I\_{f(x+\epsilon)=y}]=
> PR(x,\gamma).
> $$
> Therefore, if $P_{\epsilon}$ is unrestricted, we have $AR(x,\gamma) = PR(x,\gamma).$

---

### Official Review · Reviewer_Umvf · 2025-10-31

**Soundness:** 3
**Presentation:** 3
**Contribution:** 3
**Rating:** 6
**Confidence:** 4

**Summary:**

This paper investigates the relationship between two key types of model robustness: Adversarial Robustness (AR), which measures resilience against worst-case perturbations, and Probabilistic Robustness (PR), which measures the statistical likelihood of correct predictions under stochastic perturbations. The most important contribution is PRBench, a comprehensive benchmark designed to evaluate and compare various robustness training methods, with a huge number of empirical baseline results involving multiple methods, datasets, architectures, and metrics.

**Strengths:**

- The distinction and relationship between AR and PR are of great importance to the community. The claim that standard AT is a superior method for achieving both AR and PR is a convincing finding.

- I appreciate the extensive workload and great efforts in building this benchmark. I believe it would have a significant influence on this field and be very useful for the following studies.

- The paper is well-written, also providing a solid theoretical analysis to explain the observed differences in generalization error, which makes the empirical findings more convincing.

**Weaknesses:**

- I'm a little confused by the title and conclusion that PR comes "for free". The results clearly show that this "free" PR is paid for with a significant drop in clean accuracy. So isn't this a complex, multi-objective trade-off, instead of a "free lunch"?

- The PR-targeted methods evaluated are relatively simple. The finding is more accurately "strong AT is better than simple RT," which is less surprising. Further including more advanced PR-targeted techniques might be an effective refinement.

**Questions:**

- Would it be more accurate to frame the paper's findings as an analysis of two different trade-off frontiers: one (AT) that trades accuracy for high AR+PR, and another (RT) that trades AR for high accuracy+PR?

- The AT-PR method seems to provide the best Pareto-optimal solution across Acc, AR, and PR, with its only drawback being computational time. Does this not motivate further research into efficient hybrid training methods, rather than supporting the claim that PR-targeted methods are of "limited practical need"?

- What is the intuition behind the success of the new KL-PGD variant? Why would using a KL-based objective to find the adversary, but a standard CE-loss to train on it, be a more effective strategy than using a consistent loss for both (like in TRADES)?

---

> ### Author Response · Authors · 2025-11-20
> **Authors’ response**
>
> **Q1:** We thank the reviewer for the comment and agree with the observation. We would like to clarify that the phrase "for free" is (intentionally in quotation marks and question mark in title) to provoke discussion and encourage critical thinking, rather than to be interpreted literally. In our discussion (Line 351–356,Table 4, remark 1 in appendix(now move into conclusion)), the reviewer’s observation is valid. We have revised the paper to explicitly clarify that this phenomenon can indeed be viewed through the lens of a multi-objective trade-off among different metrics(line 356-362). And also refined the text to make the intended meaning of "for free" clearer, in line with the reviewer’s helpful suggestion (line 356-362).
>
> ---
> **Q2:** We thank the reviewer and fully agree with the point raised. Indeed, one of the core motivations (and contributions we believe) of this work is to provide a *benchmark* that facilitates future research on developing more advanced PR-targeted training methods (Lines 75: *"PRBench is designed to be extendable for future inclusion and comparison of new methods."*). In the current version, to the best of our knowledge, we have included *'all'* existing state-of-the-art PR-training methods as well as commonly used AT methods, while acknowledging that PR training remains an active and promising research direction (Lines 46–56).
>
> ---
> **Q3:** We agree with the reviewer’s insightful comment and have incorporated a concluding statement accordingly (line 505-508). Our empirical results, discussed in Sec.5, indeed indicate the two trade-off frontiers in that section (line 351-376, 404-410). We have included a more concise dicussion (line 505-508) to emphasize these observations.
>
> ---
> **Q4:** Yes, indeed, we note that *"future research should focus on developing more versatile or hybrid AT methods that balance both AR and PR such as the simple yet effective KL-PGD or the AT-inspired PR-targeted method AT-PR"* (lines 520-522). In addition, *"To further investigate efficiency-oriented approaches, we additionally integrate the filtering-based method of Chen \& Lee (2024) into PRBench as a case study (Table 12). This aligns with ongoing work on efficient robustness, and our extensible design enables inclusion of future methods.(line 422-424)"* This direction aligns directly with the reviewer’s suggestion that exploring more efficient hybrid training strategies is a promising avenue. Regarding *limited practical need*, we note that our conclusion states: *"we cautiously propose a **bold hypothesis**: there *may be* limited practical need..."* (Line 509-511) We further provide a more careful and nuanced discussion (in Footnote 3 and the Remark 1(line 1321):**Now revised into conclusion (line 511-525)**), where we clarify different interpretations of the *hypothesis*  **may (or may not) be limited practical need** from multiple practical perspectives. We are happy to provide further clarification as suggested by reviewer.
>
> ---
> **Q5:**  KL-PGD is introduced primarily as a diagnostic combination to disentangle the effects of different loss functions and perturbation generation strategies. The configuration used in KL-PGD is currently missing from existing robustness training methods. We therefore include it to fill this gap for a more systematic evaluation.  Here, we offer an intuitive explanation to support the observed empirical improvement as follows, the AEs are generated according to different optimization objectives:
>
> $\delta_{PGD} = \arg\max_{\delta} L_{CE}(p(x+\delta), y)$
>
> $\delta_{TRADES} = \arg\max_{\delta} L_{KL}(p(x) \parallel p(x+\delta))$
>
> - Effect of KL-based AE generation
>
> For $\delta_{\text{PGD}}$, the gradient towards the direction of the most extreme misclassification, and is therefore very steep. This forces the classifier to create a tight and highly curved decision boundary around each training example, thus shrinking the local margin. While for $\delta_{\text{TRADES}}$, the gradient follows the direction that smooths the predictive distribution, without fully forcing misclassification. As a result, KL-based perturbations act as a distributional smoothing regularizer. This, in turn, leads to lower generalization error and better clean accuracy.
>
>
> - Effect of CE training on KL-driven AEs
>
> While both TRADES and KL-PGD generate AEs via maximizing $L\_{KL}(p(x)\|p(x+\delta))$,
> they differ in the training objective:
>
> $L_{TRADES}=L\_{CE}(p(x),y)+ \lambda L\_{KL}(p(x)\|p(x+\delta))$
>
> $L\_{\mathrm{KL\text{-}PGD}}=
> L\_{CE}(p(x+\delta),y)$
>
> Single-objective optimization improves training stability. TRADES introduces a multi-objective optimization problem by combining the CE loss with an additional smoothness regularizer. Balancing these objectives increases the optimization difficulty. In contrast, KL-PGD adopts a single-objective formulation by directly minimizing $\mathcal{L}_{\mathrm{CE}}(p(x+\delta),y)$ on KL-driven AEs, thereby yielding more stable and effective training.

---

> > ### Comment · Reviewer_Umvf · 2025-11-26
> >
> > I appreciate the authors' clarifications and efforts in revising the manuscript. Most of my concerns are well addressed by the response and appropriately involved in the revision. I agree this work would benefit the community, and am willing to support its acceptance. I‘ve increased my rating to 8. Good luck!

---

> > > ### Author Response · Authors · 2025-11-26
> > > **Authors’ response**
> > >
> > > We sincerely thank the reviewer for the feedback and recognition of our work. The comments and suggestions have been helpful in improving our manuscript. We appreciate the reviewer’s time and support.

---

### Official Review · Reviewer_bHfN · 2025-10-31

**Soundness:** 3
**Presentation:** 3
**Contribution:** 3
**Rating:** 6
**Confidence:** 4

**Summary:**

This paper introduces PRBench, the first comprehensive benchmark designed to evaluate probabilistic robustness across diverse robustness training methods systematically. While prior work has primarily focused on adversarial robustness under worst-case perturbations, this work shifts the focus to the evaluation of probabilistic robustness. PRBench addresses these issues through a well-structured benchmark with 222 trained models, 7 datasets, and 10 architectures. Most importantly, all models are accessible via the leaderboard.

**Strengths:**

This paper provides a comprehensive and systematic analysis of probabilistic robustness (PR) across a wide variety of datasets and model architectures. Specifically, the benchmark includes diverse datasets such as CIFAR-10, CIFAR-100, MNIST, SVHN, and Tiny-ImageNet, and evaluates a broad set of models including DeiT-S, DeiT-T, ResNet-18, ResNet-34, Simple-CNN, VGG-19, ViT-B, ViT-S, and WRN-28-10.

The paper further makes a novel contribution by introducing PRBench, the first dedicated framework for evaluating and comparing PR-focused training methods. This work addresses a significant gap in robustness research, traditionally dominated by adversarial robustness.

The empirical findings are particularly insightful, and the authors also include a theoretical analysis that provides a strong conceptual foundation that deepens the understanding of the empirical results. The release of a public leaderboard and benchmark suite further enhances its impact by enabling standardized evaluation and facilitating reproducible research. Overall, the paper is well-motivated, clearly presented, and offers both theoretical and practical contributions that are likely to influence future work on adversarial robustness.

**Weaknesses:**

(Minor issue only)

The paper defines GE as “the difference between the natural and empirical risk.” I suggest including its mathematical formulation, similar to Equation (4), for clarity and consistency.

While abbreviations such as AT and GE are widely used, the paper employs too many acronyms (e.g., AT, PR, GE, RT, …), which can hinder readability. I recommend adding a table summarizing all abbreviations for convenience.

Several prior works are closely related to this paper in terms of GE and robust overfitting, but are currently omitted. I recommend citing and discussing the following:

- Jiang, Yiding, et al. “Fantastic Generalization Measures and Where to Find Them.” ICLR, 2020.

- Kim, Hoki, et al. “Fantastic Robustness Measures: The Secrets of Robust Generalization.” NeurIPS 36 (2023): 48793–48818.

These papers are conceptually and experimentally relevant, especially in how they represent and benchmark generalization and robustness. Please include them and discuss the relationship and differences with your work.

I would be happy to recommend strong acceptance if the minor issues are properly addressed.

**Questions:**

N/A

---

> ### Author Response · Authors · 2025-11-20
> **Authors’ response**
>
> ## All suggestions are included in the updated manuscript.
>
> **Q1:** We thank the reviewer for the suggestion. Here we give the mathematical formulation of Generalization Error (GE) as:
>
> **Definition (Generalization Error).**
> For a deep learning model $f_\theta$ parameterised by $\theta \in \Theta$ and trained on a dataset
> $S = \{(x_i, y_i)\}_{i=1}^n$ drawn i.i.d. from an unknown distribution $D$ over $\mathcal{X} \times \mathcal{Y}$,
> the *Generalization error (GE)* is defined as the difference between the natural and empirical risks:
>
> $
> \mathrm{GE}(\theta) = R(\theta) - R_S(\theta),
> $
>
> where the natural and empirical risks can be represented as:
>
> $R(\theta) = \mathbb{E}_{(x,y)\sim D}\big[\mathcal{L}(p(f(x,\theta)), y)\big] ;$
>
> $R_S(\theta) = \frac{1}{n}\sum_{i=1}^{n} \mathcal{L}(p(f(x_i,\theta)), y_i).$
>
>
> ---
> **Q2:** We thank the reviewer for the suggestion. We will include a table summarizing all abbreviations and symbols for convenience. For clarity, we provide the corresponding abbreviations below:
>
> | **Acronyms** | **Meaning** |
> |--------------|-------------|
> | PR  | Probabilistic Robustness |
> | AR  | Adversarial Robustness |
> | AT  | Adversarial Training |
> | RT  | Risk-based Training |
> | GE  | Generalization Error |
> | ERM | Empirical Risk Minimization |
>
>
> ---
>
> **Q3:** We thank the reviewer for bringing these related works to our attention. We concur that integrating an analysis of these papers will strengthen the paper. In addition to the two related works suggested by the reviewer, we also surveyed two other studies [1,2] that discuss model overfitting and generalization from different perspectives/theoretical-bases. Accordingly, we will revise the related work section to provide a more comprehensive overview to cover the following points:
>
> - **Scope and goal:**  **(1)** Jiang et al. study generalization in DL models with standard training framework, focusing on the empirical correlation between complexity measures (e.g., VC-dimension, norm-based, PAC-Bayes) and Generalization Error (GE). Their goal is to uncover potential causal relationships between these measures and generalization.  **(2)** Kim et al. further conduct a large-scale empirical study of robust generalization under adversarial training (AT), analyzing how margin-based, smoothness-based, flatness-based, and gradient-based measures correlate with the GE. **(3)** Work [1] investigate adversarial robust generalization through the lens of Rademacher complexity, which aim to provide theoretical insights into why robust generalization can be harder. **(4)** Work [2] investigate how the evaluation methodology in Jiang et al. can obscure the successes and failures of different generalization measures. **(5)** In contrast, our work focuses on probabilistic robustness (PR). We aim to assess the improvements in PR achieved by different robustness training methods, while also provide theoretical analysis of the GE across different training methods.
>
> - **Theoretical foundation:** **(1)** Jiang et al. base their analysis on a broad set of theoretically motivated complexity measures from generalization theory (e.g., VC-dimension, norm-based, PAC-Bayes). **(2)** Kim et al. extend this line of work to adversarial training framework by examining a wide range of margin-based, smoothness-base, and flatness-based measures under adversarial training to study their relationship with the robust generalization gap. **(3)** Work [1] investigate adversarial robust generalization through Rademacher complexity. **(4)** Work [2] complements Jiang et al.'s evaluation methodology by using a distributional robustness framework to assess generalization across diverse experimental settings. **(5)** In our work, we leverage the concept of $\eta$-approximate $\beta$-smoothness from Xiao et al. (2022b) to analyze how the optimization objectives of different training methods affect GE, highlighting the connection between optimization objectives and generalization in different robustness settings.
>
> - **Model and Dataset Coverage:** **(1)** Jiang et al. conduct their analysis on convolutional networks trained on CIFAR-10 and SVHN. **(2)** Kim et al. conduct their study to ResNet-18, WRN28-10 and WRN34-10 on CIFAR-10. **(3)** Work [1] conduct their study on liner classifier and a four-layer ReLU network on MNIST. **(4)** Work [2] conduct their study on convolutional networks trained on CIFAR-10 and SVHN. **(5)** In contrast, our work covers 3 families of model architectures: (i) plain CNNs, (ii) residual networks, and (iii) transformer-based models; and train a diverse set of models including VGG-19, SimpleCNN, ResNet-18/34, WRN-28-10, ViT (Small/Base/Large), and DeiT (Tiny/Small). Moreover, our study spans 7 datasets (MNIST, SVHN, CIFAR-10, CIFAR-100, CINIC-10, TinyImageNet, and ImageNet-50).
>
>
> [1] Yin et al. "Rademacher Complexity for Adversarially Robust Generalization" ICML 2019
>
> [2] Dziugaite, G.K. et al. "In Search of Robust Measures of Generalization" NeurIPS 2020

---

### Comment · Area_Chair_keJM · 2025-11-25

Dear reviewers,

The authors have responded. We kindly ask you to review the authors' responses to your comments and provide your feedback. Thank you.

Best,

AC

---

### Author Response · Authors · 2025-12-01
**Author Final Remarks**

We thank the reviewers and the AC for their time, valuable feedback, and suggestions. We believe we have addressed all concerns raised by the four reviewers, as confirmed by the responses from reviewers **bHfN, Umvf, and hsym** during the rebuttal. **As a result, two reviewers, bHfN and Umvf, have raised their scores to 8 (good paper). All of these discussions and score updates occurred before 27/11.**

We appreciate that the reviewers recognized the strengths of our work:
1)  PRBench as the first thorough probabilistic robustness benchmark, filling an important gap in robustness research (bHfN, Umvf);

2) Comprehensive and systematic experiments with the released leaderboard offer insightful findings (bHfN, Umvf, KUcR, hsym);

3) Solid theoretical analysis of generalization error (bHfN, Umvf, hsym);

4) Significant value and impact to the community (bHfN, Umvf, KUcR, hsym);

5) Well-written with a clear presentation (bHfN, Umvf);

Although **Reviewer KUcR** did not have the chance to respond to our discussion before the early termination of the discussion due to the incident, we believe all of their major concerns have been fully addressed in our rebuttal.
1) First, the reviewer explicitly stated that *"**In addition to my comments in the above "weakness" section, I also believe that the paper would have notable contribution** if the author could further provide some persuasive explanations on the relationship between AR and PR."* This comment also explicitly reflects the reviewer’s recognition of the contribution of our work. We emphasize that this relationship is introduced in the preliminaries (Sec. 2), and we further provided a formal proposition together with a complete proof establishing the connection between AR and PR.
2) Second, regarding the concern that cross-entropy is not globally Lipschitz, we showed with a solid argument that our analysis only involves the softmax–cross-entropy composite, which is globally Lipschitz. This is because the infinite term in the derivative of the cross-entropy is canceled by the corresponding one in the softmax derivative (as demonstrated in Eq. 44). We further provide additional formal proofs of the Lipschitzness of $L\_{CE}(p(z), y)$ in previous published work [1,2].

3) Third, for the concern about using non-smooth activations, we conducted additional experiments replacing ReLU with smooth activations (GELU/SoftPlus). The results confirm that GE remains consistent with our theoretical analysis, indicating that the theory continues to capture the essential mechanisms underlying GE.

Overall, we believe their key concerns have been thoroughly clarified.

In case there are further questions, we would like to take this opportunity to clearly restate the scope, motivation and contributions of our study.

**Scope and motivation:**

Our study investigates PR beyond the focus of existing PR-specific training methods, many SoTA AT methods for AR are also known to impact PR. Our benchmark systematically evaluates a broad spectrum of robustness training methods under a unified framework that enables fair, reproducible, and comparable PR evaluation.

PR has become an active and growing research topic, with recent peer-reviewed works addressing its evaluation, theoretical properties, and applications across diverse AI tasks (Lines 46–56). Our benchmark thus serves not merely as a comparison of methods, but as a foundational resource for the growing research field.

**Contributions:**

1. **Timely benchmark for a growing research area.** We provide the first unified framework for PR evaluation, filling an important gap in this rapidly emerging topic.

2. **Comprehensive evaluation and practical insight.** Our extensive and systematic experiments provide insightful findings, revealing that general AT methods can deliver substantial PR improvements and offering actionable guidance. The benchmark further includes standardized training configurations and complete implementations for all evaluated methods, enabling both reproducibility and extensibility.

3. **Unified theoretical analysis.** We provide a solid theoretical analysis of the generalization error, comparing both AT and PR-targeted methods. Our analysis highlights the multifaceted trade-offs among PR, AR, GE, and efficiency, offering meaningful insights to the community and inspiring future research.

4. **Public leaderboard and benchmark release.** Our unified theoretical analysis, open-source toolbox, and public leaderboard together advance PR research and promote consistent evaluation practices in this evolving field.

[1] "Logistic regression: Tight bounds for stochastic and online optimization. Conference on Learning Theory." PMLR, 2014.

[2] "On transfer of adversarial robustness from pretraining to downstream tasks." NeurIPS23

---

### Meta-Review · Area_Chair_13dr · 2026-01-08

**Summary:**

The paper identifies probabilistic robustness as a research gap, provides a theoretic formalization (in line with existing literature) and contributes a comprehensive benchmark. As take-away message, the paper concludes that adversarial training yields higher probabilistic and adversarial robustness, and that probabilistic robustness training preserved higher clean accuracy.
The reviewers have raised several concerns regarding the formalization - most of which have been successfully addressed in the rebuttal. However, the AC is not convinced by the manuscript. The fact that models trained to be robust have a lower clean accuracy is well known. First: The observed trade-off (higher adversarial and probabilistic robustness --> lower clean accuracy is not novel and can not be seen as a novel finding).
Second: In contrast to statements in the paper, AT does not use deterministic attacks. In contrast, PGD-based training starts with a random perturbation that is subsequently optimized. The distinction "probabilistic vs deterministic" is therefore not correct.
Third: The paper identifies the lack of a benchmark for probabilistic robustness as a research gap and closes it. At the same time, the core finding is that adversarial robustness seems to be the stronger measure. The paper thereby invalidates its own premise.
Although the paper has received borderline to positive reviews, the AC therefore recommends rejection and encourages the authors to thoroughly discuss the above points in a revision.

**Reviewer Concerns:**

Reviewer bHfN: missing related work and some definitions --> all addressed
Reviewer Umvf: confusing title --> addressed through explanation
Reviewer Umvf: PR methods are weak training methods: addressed through argumentation. This is in line with the AC point made above: weaker robustness training (adversarial or purely random) is expected to lead to weaker robustness and stronger clean performance. Reviewer KUcR: confusing description and formalization --> to the best of the AC's understanding, these points are properly addressed.
Reviewer hsym: some reorganization of the theorems recommended. These are implemented by the authors in the revision and the concerns are properly addressed.

**Reviewer Scores:**

Reviewer bHfN: 6 --> 8
Reviewer Umvf: 6 --> 8
Reviewer KUcR: 4
Reviewer hsym: 6 --> 6 or 8

---

### Decision · Program_Chairs · 2026-01-26

Reject